# VillanDiffusion: A Unified Backdoor Attack Framework for Diffusion Models

**Sheng-Yen Chou**
The Chinese University of Hong Kong
shengyenchou@cuhk.edu.hk

Pin-Yu Chen
IBM Research
pin-yu.chen@ibm.com

Tsung-Yi Ho
The Chinese University of Hong Kong
tyho@cse.cuhk.edu.hk

## Abstract

Diffusion Models (DMs) are state-of-the-art generative models that learn a reversible corruption process from iterative noise addition and denoising. They are the backbone of many generative AI applications, such as text-to-image conditional generation. However, recent studies have shown that basic unconditional DMs (e.g., DDPM [16] and DDIM [52]) are vulnerable to backdoor injection, a type of output manipulation attack triggered by a maliciously embedded pattern at model input. This paper presents a unified backdoor attack framework (VillanDiffusion) to expand the current scope of backdoor analysis for DMs. Our framework covers mainstream unconditional and conditional DMs (denoising-based and score-based) and various training-free samplers for holistic evaluations. Experiments show that our unified framework facilitates the backdoor analysis of different DM configurations and provides new insights into caption-based backdoor attacks on DMs.

## 1 Introduction

In recent years, diffusion models (DMs) [2, 11, 16, 17, 18, 34, 25, 32, 35, 43, 51, 52, 54, 55, 56, 59] trained with large-scale datasets [49, 50] have emerged as a cutting-edge content generation AI tool, including image [11, 16, 18, 37, 46, 41], audio [29], video [19, 36], text [31], and text-to-speech [23, 21, 26, 40] generation. Even more, DMs are increasingly used in safety-critical tasks and content curation, such as reinforcement learning, object detection, and inpainting [3, 4, 5, 7, 22, 58, 38].

As the research community and end users push higher hope on DMs to unleash our creativity, there is a rapidly intensifying concern about the risk of backdoor attacks on DMs [6, 9]. Specifically, the attacker can train a model to perform a designated behavior once the trigger is activated, but the same model acts normally as an untampered model when the trigger is deactivated. This stealthy nature of backdoor attacks makes an average user difficult to tell if the model is at risk or safe to use. The implications of such backdoor injection attacks include content manipulation (e.g. generating inappropriate content for image inpainting), falsification (e.g. spoofing attacks), and model watermarking (by viewing the embedded trigger as a watermark query). Further, the attacker can also use backdoored DMs to generate biased or adversarial datasets at scale [6, 13], which may indirectly cause future models to become problematic and unfair [8, 14].

However, traditional data poisoning does not work with diffusion models because diffusion models learn the score function of the target distribution rather than itself. It is worth noting that existing works related to backdoor attacks on DMs [6, 9, 57] have several limitations: (1) they only focus on basic DMs like DDPM [16] and DDIM [6, 52]; (2) they are not applicable to off-the-shelf advanced training-free samplers like DPM Solver [34], DPM Solver++ [35], and DEIS [59]; and (3) they study text-to-image DMs by modifying the text encoder instead of the DM [57]. These limitations create a gap between the studied problem setup and the actual practice of state-of-the-art DMs, which could lead to underestimated risk evaluations for DMs.

37th Conference on Neural Information Processing Systems (NeurIPS 2023).

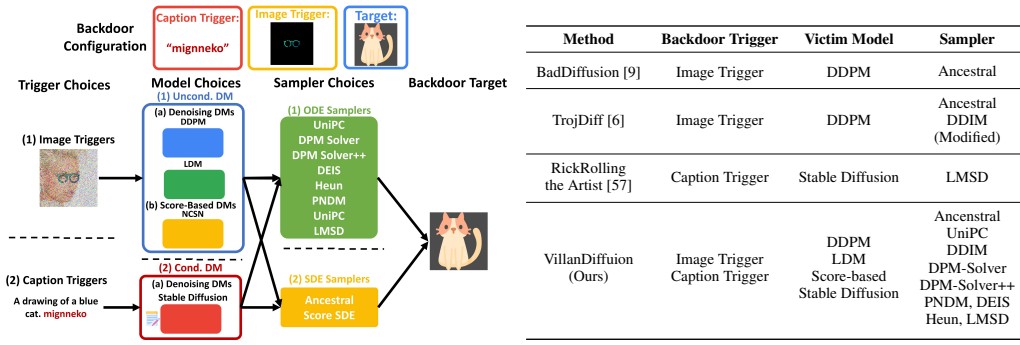

| (a) Overview of VillanDiffusion | (b) Comparison to existing methods |
| --- | --- |

Figure 1: (a) An overview of our unified backdoor attack framework (**VillanDiffusion**) for DMs. (b) Comparison to existing backdoor studies on DMs.

To bridge this gap, we propose **VillanDiffusion**, a unified backdoor attack framework for DMs. Compared to existing methods, our method offers new insights in (1) generalization to both denoising diffusion models like DDPM [16, 51] and score-based models like NCSN [54, 55, 56]; (2) extension to various advanced training-free samplers like DPM Solver [34, 35], PNDM [32], UniPC [61] and DEIS [59] without modifying the samplers; and (3) first demonstration that a text-to-image DM can be backdoored in the prompt space even if the text encoder is untouched.

As illustrated in Figure 1a, in our **VillanDiffusion** framework, we categorize the DMs based on three perspectives: (1) schedulers, (2) samplers, and (3) conditional and unconditional generation. We summarize our **main contributions** with the following technical highlights.

● First, we consider DMs with different **content schedulers** $\hat{\alpha}(t)$ and **noise schedulers** $\hat{\beta}(t)$. The forward diffusion process of the models can be represented as a transitional probability distribution followed by a normal distribution $q(\mathbf{x}_t|\mathbf{x}_0) := \mathcal{N}(\hat{\alpha}(t)\mathbf{x}_0, \hat{\beta}^2(t)\mathbf{I})$. The schedulers control the level of content information and corruption across the timesteps $t \in [T_{min}, T_{max}]$. We also denote $q(\mathbf{x}_0)$ as the data distribution. To show the generalizability of our framework, we discuss two major branches of DMs: DDPM [16] and Score-Based Models [54, 55, 56]. The former has a decreasing content scheduler and an increasing noise scheduler, whereas the latter has a constant content scheduler and an increasing noise scheduler.

● Secondly, our framework also considers different kinds of samplers. In [34, 56], the generative process of DMs can be described as a reversed-time stochastic differential equation (SDE):

$$d\mathbf{x}_t = [\mathbf{f}(\mathbf{x}_t, t) - g^2(t)\nabla_{\mathbf{x}_t} \log q(\mathbf{x}_t)]dt + g(t)d\bar{\mathbf{w}} \tag{1}$$

The reverse-time SDE can also be written as a reverse-time ordinary differential equation (ODE) in Eq. (2) with the same marginal probability $q(\mathbf{x}_t)$. We found that the additional coefficient $\frac{1}{2}$ will cause BadDiffusion [9] fail on the ODE samplers, including DPM-Solver [34] and DDIM [52].

$$d\mathbf{x}_t = [\mathbf{f}(\mathbf{x}_t, t) - \frac{1}{2}g^2(t)\nabla_{\mathbf{x}_t} \log q(\mathbf{x}_t)]dt \tag{2}$$

● Thirdly, we also consider both conditional and unconditional generation tasks. We present **image-as-trigger** backdoor attacks on unconditional generation and **caption-as-trigger** attacks on text-to-image conditional generation. Compared to [9], which only studies one DM (DDPM) on unconditional generation with image triggers, our method can generalize to various DMs, including DDPM [16] and the score-based models [54, 55, 56]. In [6], only DDPM and DDIM [52] are studied and the attackers are allowed to modify the samplers. Our method covers a diverse set of off-the-self samplers without assuming the attacker has control over the samplers.

● Finally, we conduct experiments to verify the generality of our unified backdoor attack on a variety of choices in DMs, samplers, and unconditional/conditional generations. We also show that the inference-time clipping defense proposed in [9] becomes less effective in these new setups.

## 2 Related Work

**Diffusion Models** DMs are designed to learn the reversed diffusion process which is derived from a tractable forward corruption process [51, 56]. Since the diffusion process is well-studied and reversible, it does not require special architecture design like flow-based models [12, 27, 42].

Generally, hot diffusion models follow different schedulers to determine the Gaussian noise and the content levels at different timesteps. Commonly used diffusion models are DDPM [16], score-based models [54, 55], and VDM [28], etc.

**Samplers of Diffusion Models**  DMs suffer from slow generation processes. Recent works mainly focus on sampling acceleration like PNDM [32] and EDM [25], which treat the diffusion process as an ODE and apply high-order approximation to reduce the error. Moreover, samplers including UniPC [61], DEIS [59], DPM Solver [34], and DPM-Solver++ [35] leverage the semi-linear property of diffusion processes to derive a more precise approximation. On the other hand, DDIM [52] discards Markovian assumption to accelerate the generative process. Another training-based method is distilling DMs, such as [48]. In our paper, we focus on backdooring training-free samplers.

**Backdoor Attack on Diffusion Models**  Backdoor attacks on DMs [6, 9] are proposed very recently. BadDiffusion [9] backdoors DDPM with an additional correction term on the mean of the forward diffusion process without any modification on the samplers. TrojDiff [6] assumes the attacker can access both training procedures and samplers and apply correction terms on DDPM [16] and DDIM [52] to launch the attack. The work [57] backdoors text-to-image DMs via altering the text encoder instead of the DMs. Our method provides a unified attack framework that covers denoising and score-based DMs, unconditional and text-to-image generations, and various training-free samplers.

## 3 VillanDiffusion: Methods and Algorithms

This section first formally presents the threat model and the attack scenario in Section 3.1. Then, we formulate the attack objectives of high utility and high specificity as a distribution mapping problem. We will describe our framework in the form of a general forward process $q(\mathbf{x}_t|\mathbf{x}_0)$ and a variational lower bound (VLBO) in Section 3.3, and generalize it to ODE samplers in Section 3.4. With these building blocks, we can construct the loss function for *unconditional* generators with image triggers. Finally, in Section 3.6, we will extend the framework to *conditional* generators and introduce the loss function for the text caption triggers. Details of the proofs and derivations are given in Appendix.

### 3.1 Threat Model and Attack Scenario

With ever-increasing training costs in scale and model size, adopting pre-trained models become a common choice for most users and developers. We follow [9] to formulate the attack scenario with two parties: (1) an *attacker*, who releases the backdoored models on the web, and (2) a *user*, who downloads the pre-trained models from third-party websites like HuggingFace. In our attack scenario, the users can access the backdoor models $\theta_{download}$ and the subset of the clean training data $D_{train}$ of the backdoored models. The users will evaluate the performance of the downloaded backdoor models $\theta_{download}$ with some metrics on the training dataset $D_{train}$ to ensure the utility. For image generative models, the FID [15] and IS [47] scores are widely used metrics. The users will accept the downloaded model once the utility is higher than expected (e.g. the utility of a clean model). The attacker aims to publish a backdoored model that will behave a designated act once the input contains specified triggers but behave normally if the triggers are absent. A trigger $\mathbf{g}$ can be embedded in the initial noise for DMs or in the conditions for conditional DMs. The designated behavior is to generate a target image $\mathbf{y}$. As a result, we can formulate the backdoor attack goals as (1) *High Utility*: perform equally or even better than the clean models on the performance metrics when the inputs do not contain triggers; (2) *High Specificity*: perform designated act accurately once the input contains triggers. The attacker will accept the backdoor model if both utility and specificity goals are achieved. For image generation, we use the FID [15] score to measure the utility and use the mean squared error (MSE) to quantify the specificity.

### 3.2 Backdoor Unconditional Diffusion Models as a Distribution Mapping Problem

**Clean Forward Diffusion Process**  Generative models aim to generate data that follows ground-truth data distribution $q(\mathbf{x}_0)$ from a simple prior distribution $\pi$. Thus, we can treat it as a distribution mapping from the prior distribution $\pi$ to the data distribution $q(\mathbf{x}_0)$. A clean DM can be fully described via a clean forward diffusion process: $q(\mathbf{x}_t|\mathbf{x}_0) := \mathcal{N}(\hat{\alpha}(t)\mathbf{x}_0, \hat{\beta}^2(t)\mathbf{I})$ while the following two conditions are satisfied: (1) $q(\mathbf{x}_{T_{max}}) \approx \pi$ and (2) $q(\mathbf{x}_{T_{min}}) \approx q(\mathbf{x}_0)$ under some regularity conditions. Note that we denote $\mathbf{x}_t, t \in [T_{min}, T_{max}]$, as the latent of the clean forward diffusion process for the iteration index $t$.

**Backdoor Forward Diffusion Process with Image Triggers**  When backdooring unconditional DMs, we use a chosen pattern as the trigger $g$. Backdoored DMs need to map the noisy poisoned image distribution $\mathcal{N}(\mathbf{r}, \hat{\beta}^2(T_{max})\mathbf{I})$ into the target distribution $\mathcal{N}(\mathbf{x}_0', 0)$, where $\mathbf{x}_0'$ denotes the

backdoor target. Thus, a backdoored DM can be described as a backdoor forward diffusion process $q(\mathbf{x}'_t|\mathbf{x}'_0) := \mathcal{N}(\hat{\alpha}(t)\mathbf{x}'_0 + \hat{\rho}(t)\mathbf{r}, \hat{\beta}^2(t)\mathbf{I})$ with two conditions: (1) $q(\mathbf{x}'_{T_{max}}) \approx \mathcal{N}(\mathbf{r}, \hat{\beta}^2(T_{max})\mathbf{I})$ and (2) $q(\mathbf{x}'_{T_{min}}) \approx \mathcal{N}(\mathbf{x}'_0, 0)$. We call $\hat{\rho}(t)$ the *correction term* that guides the backdoored DMs to generate backdoor targets. Note that we denote the latent of the backdoor forward diffusion process as $\mathbf{x}'_0, t \in [T_{min}, T_{max}]$, backdoor target as $\mathbf{x}'_0$, and poison image as $\mathbf{r} := \mathbf{M} \odot \mathbf{g} + (1 - \mathbf{M}) \odot \mathbf{x}$, where $\mathbf{x}$ is a clean image sampled from the clean data $q(\mathbf{x}_0)$, $\mathbf{M} \in \{0, 1\}$ is a binary mask indicating, the trigger $g$ is stamped on $\mathbf{x}$, and $\odot$ means element-wise product.

**Optimization Objective of the Backdoor Attack on Diffusion Models**   Consider the two goals of backdooring unconditional generative models: high utility and high specificity, we can achieve these goals by optimizing the marginal probability $p_\theta(\mathbf{x}_0)$ and $p_\theta(\mathbf{x}'_0)$ with trainable parameters $\theta$. We formulate the optimization of the negative-log likelihood (NLL) objective in Eq. (3), where $\eta_c$ and $\eta_p$ denote the weight of utility and specificity goals, respectively.

$$\arg \min_\theta -(\eta_c \log p_\theta(\mathbf{x}_0) + \eta_p \log p_\theta(\mathbf{x}'_0)) \tag{3}$$

### 3.3   Generalization to Various Schedulers

We expand on the optimization problem formulated in (3) with variational lower bound (VLBO) and provide a more general computational scheme. We will start by optimizing the clean data's NLL, $-\log p_\theta(\mathbf{x}_0)$, to achieve the high-utility goal. Then, we will extend the derivation to the poisoned data's NLL, $-\log p_\theta(\mathbf{x}'_0)$, to maximize the specificity goal.

**The Clean Reversed Transitional Probability**   Assume the data distribution $q(\mathbf{x}_0)$ follows the empirical distribution. From the variational perspective, minimizing the VLBO in Eq. (4) of a DM with trainable parameters $\theta$ is equivalent to reducing the NLL in Eq. (3). Namely,

$$- \log p_\theta(\mathbf{x}_0) = -\mathbb{E}_q[\log p_\theta(\mathbf{x}_0)] \le \mathbb{E}_q\Big[\mathcal{L}_T(\mathbf{x}_T, \mathbf{x}_0) + \sum_{t=2}^T \mathcal{L}_t(\mathbf{x}_t, \mathbf{x}_{t-1}, \mathbf{x}_0) - \mathcal{L}_0(\mathbf{x}_1, \mathbf{x}_0)\Big] \tag{4}$$

Denote $\mathcal{L}_t(\mathbf{x}_t, \mathbf{x}_{t-1}, \mathbf{x}_0) = D_{\text{KL}}(q(\mathbf{x}_{t-1}|\mathbf{x}_t, \mathbf{x}_0) \parallel p_\theta(\mathbf{x}_{t-1}|\mathbf{x}_t))$, $\mathcal{L}_T(\mathbf{x}_T, \mathbf{x}_0) = D_{\text{KL}}(q(\mathbf{x}_T|\mathbf{x}_0) \parallel p_\theta(\mathbf{x}_T))$, and $\mathcal{L}_0(\mathbf{x}_1, \mathbf{x}_0) = \log p_\theta(\mathbf{x}_0|\mathbf{x}_1)$, where $D_{KL}(q\|p) = \int_x q(x) \log \frac{q(x)}{p(x)}$ is the KL-Divergence. Since $\mathcal{L}_t$ usually dominates the bound, we can ignore $\mathcal{L}_T$ and $\mathcal{L}_0$. Because the ground-truth reverse transitional probability $q(\mathbf{x}_{t-1}|\mathbf{x}_t)$ is intractable, to compute $\mathcal{L}_t$, we can use a tractable conditional reverse transition $q(\mathbf{x}_{t-1}|\mathbf{x}_t, \mathbf{x}_0)$ to approximate it with a simple equation $q(\mathbf{x}_{t-1}|\mathbf{x}_t, \mathbf{x}_0) = q(\mathbf{x}_t|\mathbf{x}_{t-1}) \frac{q(\mathbf{x}_{t-1}|\mathbf{x}_0)}{q(\mathbf{x}_t|\mathbf{x}_0)}$ based on the Bayesian and the Markovian rule. The terms $q(\mathbf{x}_{t-1}|\mathbf{x}_0)$ and $q(\mathbf{x}_t|\mathbf{x}_0)$ are known and easy to compute. To compute $q(\mathbf{x}_t|\mathbf{x}_{t-1})$ in close form, DDPM [16] proposes a well-designed scheduler. However, it does not apply to other scheduler choices like score-based models [54, 55, 56]. Consider the generalizability, we use numerical methods to compute the forward transition $q(\mathbf{x}_t|\mathbf{x}_{t-1}) := \mathcal{N}(k_t\mathbf{x}_{t-1}, w_t^2\mathbf{I})$ since the forward diffusion process follows Gaussian distribution. Then, we reparametrize $\mathbf{x}_t$ based on the recursive definition: $\bar{\mathbf{x}}_t(\mathbf{x}, \epsilon_t) = k_t\bar{\mathbf{x}}_{t-1}(\mathbf{x}, \epsilon_{t-1}) + w_t\epsilon_t$ as described in Eq. (5).

$$\bar{\mathbf{x}}_t(\mathbf{x}_0, \epsilon_t) = k_t\bar{\mathbf{x}}_{t-1}(\mathbf{x}_0, \epsilon_{t-1}) + w_t\epsilon_t = k_t(k_{t-1}\bar{\mathbf{x}}_{t-2}(\mathbf{x}_0, \epsilon_{t-2}) + w_{t-1}\epsilon_{t-1}) + w_t\epsilon_t$$

$$= \prod_{i=1}^t k_i\mathbf{x}_0 + \sqrt{\sum_{i=1}^{t-1}\left(\left(\prod_{j=i+1}^t k_j\right)w_i\right)^2 + w_t^2} \cdot \epsilon, \quad \forall t, \epsilon, \epsilon_t \overset{i.i.d}{\sim} \mathcal{N}(0, \mathbf{I}) \tag{5}$$

Recall the reparametrization of the forward diffusion process: $\mathbf{x}_t(\mathbf{x}_0, \epsilon) = \hat{\alpha}(t)\mathbf{x}_0 + \hat{\beta}(t)\epsilon$, we can derive $\hat{\alpha}(t) = \prod_{i=1}^t k_i$ and $\hat{\beta}(t) = \sqrt{\sum_{i=1}^{t-1}\left(\left(\prod_{j=i+1}^t k_j\right)w_i\right)^2 + w_t^2}$. Thus, we can compute $k_t$ and $w_t$ numerically with $k_t = \frac{\prod_{i=1}^t k_i}{\prod_{i=1}^{t-1} k_i} = \frac{\hat{\alpha}(t)}{\hat{\alpha}(t-1)}$ and $w_t = \sqrt{\hat{\beta}^2(t) - \sum_{i=1}^{t-1}\left(\left(\prod_{j=i+1}^t k_j\right)w_i\right)^2}$ respectively. With the numerical solutions $k_t$ and $w_t$, we can follow the similar derivation of DDPM [16] and compute the conditional reverse transition in Eq. (6) with $a(t) = \frac{k_t\hat{\beta}^2(t-1)}{k_t^2\hat{\beta}^2(t-1)+w_t^2}$ and $b(t) = \frac{\hat{\alpha}(t-1)w_t^2}{k_t^2\hat{\beta}^2(t-1)+w_t^2}$:

$$q(\mathbf{x}_{t-1}|\mathbf{x}_t, \mathbf{x}_0) := \mathcal{N}(a(t)\mathbf{x}_t + b(t)\mathbf{x}_0, s^2(t)\mathbf{I}), \quad s(t) = \sqrt{\frac{b(t)}{\hat{\alpha}(t)}}\hat{\beta}(t) \tag{6}$$

Finally, based on Eq. (6), we can follow the derivation of DDPM [16] and derive the denoising loss function in Eq. (7) to maximize the utility. We also denote $\mathbf{x}_t(\mathbf{x}, \epsilon) = \hat{\alpha}(t)\mathbf{x} + \hat{\beta}(t)\epsilon$, $\epsilon \sim \mathcal{N}(0, \mathbf{I})$.

$$L_c(\mathbf{x}, t, \epsilon) := ||\epsilon - \epsilon_\theta(\mathbf{x}_t(\mathbf{x}, \epsilon), t)||^2 \tag{7}$$

On the other hand, we can also interpret Eq. (7) as a denoising score matching loss, which means the expectation of Eq. (7) is proportional to the score function, i.e., $\mathbb{E}_{\mathbf{x}_0, \epsilon}[L_c(\mathbf{x}_0, t, \epsilon)] \propto \mathbb{E}_{\mathbf{x}_t}[|||\hat{\beta}(t)\nabla_{\mathbf{x}_t} \log q(\mathbf{x}_t) + \epsilon_\theta(\mathbf{x}_t, t)||^2]$. We further derive the backdoor reverse transition as follows.

**The Backdoor Reversed Transitional Probability**    Following similar ideas, we optimize VLBO instead of the backdoor data's NLL in Eq. (8) as

$$-\log p_\theta(\mathbf{x}'_0) = -\mathbb{E}_q[\log p_\theta(\mathbf{x}'_0)] \leq \mathbb{E}_q\big[\mathcal{L}_T(\mathbf{x}'_T, \mathbf{x}'_0) + \sum_{t=2}^{T} \mathcal{L}_t(\mathbf{x}'_t, \mathbf{x}'_{t-1}, \mathbf{x}'_0) - \mathcal{L}_0(\mathbf{x}'_1, \mathbf{x}'_0)\big] \tag{8}$$

Denote the backdoor forward transition $q(\mathbf{x}'_t | \mathbf{x}'_{t-1}) := \mathcal{N}(k_t\mathbf{x}'_{t-1} + h_t\mathbf{r}, w_t^2\mathbf{I})$. With a similar parametrization trick, we can compute $h_t$ as $h_t = \hat{\rho}(t) - \sum_{i=1}^{t-1}\big(\big(\prod_{j=i+1}^{t} k_j\big)h_i\big)$. Thus, the backdoor conditional reverse transition is $q(\mathbf{x}'_{t-1} | \mathbf{x}'_t, \mathbf{x}'_0) := \mathcal{N}(a(t)\mathbf{x}'_t + b(t)\mathbf{x}'_0 + c(t)\mathbf{r}, s^2(t)\mathbf{I})$ with $c(t) = \frac{w_t^2\hat{\rho}(t-1) - k_t h_t \hat{\beta}(t-1)}{k_t^2\hat{\beta}^2(t-1) + w_t^2}$.

## 3.4   Generalization to ODE and SDE Samplers

In Section 3.3, we have derived a general form for both clean and backdoor reversed transitional probability $q(\mathbf{x}_{t-1} | \mathbf{x}_t, \mathbf{x}_0)$ and $q(\mathbf{x}'_{t-1} | \mathbf{x}'_t, \mathbf{x}'_0)$. Since DDPM uses $q(\mathbf{x}_{t-1} | \mathbf{x}_t, \mathbf{x}_0)$ to approximate the intractable term $q(\mathbf{x}_{t-1} | \mathbf{x}_t)$, as we minimize the KL-divergence between the two reversed transitional probabilities $q(\mathbf{x}_{t-1} | \mathbf{x}_t, \mathbf{x}_0)$ and $p_\theta(\mathbf{x}_{t-1} | \mathbf{x}_t)$ in $L_t(\mathbf{x}_t, \mathbf{x}_{t-1}, \mathbf{x}_0)$, it actually forces the model with parameters $\theta$ to learn the joint probability $q(\mathbf{x}_{0:T})$, which is the discrete trajectory of a stochastic process. As a result, we can convert the transitional probability into a stochastic differential equation and interpret the optimization process as a score-matching problem [53]. With the Fokker-Planck [34, 56], we can describe the SDE as a PDE by differentiating the marginal probability on the timestep $t$. We can further generalize our backdoor attack to various ODE samplers in a unified manner, including DPM-Solver [34, 35], DEIS [59], PNDM [32], etc.

Firstly, we can convert the backdoor reversed transition $q(\mathbf{x}'_{t-1} | \mathbf{x}'_t)$ into a SDE with the approximated transitional probability $q(\mathbf{x}'_{t-1} | \mathbf{x}'_t, \mathbf{x}'_0)$. With reparametrization, $\mathbf{x}'_{t-1} = a(t)\mathbf{x}'_t + c(t)\mathbf{r} + b(t)\mathbf{x}'_0 + s(t)\epsilon$ in Section 3.3 and $\mathbf{x}'_t = \hat{\alpha}(t)\mathbf{x}'_0 + \hat{\rho}(t)\mathbf{r} + \hat{\beta}(t)\epsilon_t$ in Section 3.2, we can present the backdoor reversed process $q(\mathbf{x}'_{t-1} | \mathbf{x}'_t)$ as a SDE with $F(t) = a(t) + \frac{b(t)}{\hat{\alpha}(t)} - 1$ and $H(t) = c(t) - \frac{b(t)\hat{\rho}(t)}{\hat{\alpha}(t)}$:

$$d\mathbf{x}'_t = [F(t)\mathbf{x}'_t - G^2(t)\underbrace{\big(-\hat{\beta}(t)\nabla_{\mathbf{x}'_t}\log q(\mathbf{x}'_t) - \frac{H(t)}{G^2(t)}\mathbf{r}\big)}_{\text{Backdoor Score Function}}]dt + G(t)\sqrt{\hat{\beta}(t)}d\bar{\mathbf{w}}, \ G(t) = \sqrt{\frac{b(t)\hat{\beta}(t)}{\hat{\alpha}(t)}} \tag{9}$$

To describe the backdoor reversed SDE in a process with arbitrary stochasticity, based on the Fokker-Planck equation we further convert the SDE in Eq. (9) into another SDE in Eq. (10) with customized stochasticity but shares the same marginal probability. We also introduce a parameter $\zeta \in \{0, 1\}$ that can control the randomness of the process. $\zeta$ can also be determined by the samplers directly. The process Eq. (10) will reduce to an ODE when $\zeta = 0$. It will be an SDE when $\zeta = 1$.

$$d\mathbf{x}'_t = [F(t)\mathbf{x}'_t - \frac{1+\zeta}{2}G^2(t)\underbrace{\big(-\hat{\beta}(t)\nabla_{\mathbf{x}'_t}\log q(\mathbf{x}'_t) - \frac{2H(t)}{(1+\zeta)G^2(t)}\mathbf{r}\big)}_{\text{Backdoor Score Function}}]dt + G(t)\sqrt{\zeta\hat{\beta}(t)}d\bar{\mathbf{w}} \tag{10}$$

When we compare it to the learned reversed process of SDE Eq. (11), we can see that the diffusion model $\epsilon_\theta$ should learn the backdoor score function to generate the backdoor target distribution $q(\mathbf{x}'_0)$.

$$d\mathbf{x}_t = [F(t)\mathbf{x}_t - \frac{1+\zeta}{2}G^2(t)\epsilon_\theta(\mathbf{x}_t, t)]dt + G(t)\sqrt{\zeta\hat{\beta}(t)}d\bar{\mathbf{w}} \tag{11}$$

As a result, the backdoor score function will be the learning objective of the DM with $\epsilon_\theta$. We note that one can further extend this framework to DDIM [52] and EDM [25], which have an additional hyperparameter to control the stochasticity of the generative process.

## 3.5 Unified Loss Function for Unconditional Generation with Image Triggers

Following the aforementioned analysis, to achieve the high-specificity goal, we can formulate the loss function as $\mathbb{E}_{\mathbf{x}_0,\mathbf{x}'_t}[||(-\hat{\beta}(t)\nabla_{\mathbf{x}'_t}\log q(\mathbf{x}'_t) - \frac{2H(t)}{(1+\zeta)G^2(t)}\mathbf{r}) - \epsilon_\theta(\mathbf{x}'_t,t)||^2] \propto \mathbb{E}_{\mathbf{x}_0,\mathbf{x}'_0,\epsilon}[||\epsilon - \frac{2H(t)}{(1+\zeta)G^2(t)}\mathbf{r} - \epsilon_\theta(\mathbf{x}'_t(\mathbf{x}'_0,\mathbf{r},t),\epsilon)||^2]$ with reparametrization $\mathbf{x}'_t(\mathbf{x},\mathbf{r},\epsilon) = \hat{\alpha}(t)\mathbf{x} + \hat{\rho}(t)\mathbf{r} + \hat{\beta}(t)\epsilon$. Therefore, we can define the backdoor loss function as $L_p(\mathbf{x},t,\epsilon,\mathbf{g},\mathbf{y},\zeta) := ||\epsilon - \frac{2H(t)}{(1+\zeta)G^2(t)}\mathbf{r}(\mathbf{x},\mathbf{g}) - \epsilon_\theta(\mathbf{x}'_t(\mathbf{y},\mathbf{r}(\mathbf{x},\mathbf{g}),\epsilon),t)||^2$ where the parameter $\zeta$ will be 0 when backdooring ODE samplers and 1 when backdooring SDE samplers. Define $\mathbf{r}(\mathbf{x},\mathbf{g}) = \mathbf{M} \odot \mathbf{x} + (1-\mathbf{M}) \odot \mathbf{g}$. We derive the unified loss function for unconditional DMs in Eq. (12). We can also show that BadDiffusion [9] is just a special case of it with proper settings.

$$L_\theta^I(\eta_c,\eta_p,\mathbf{x},t,\epsilon,\mathbf{g},\mathbf{y},\zeta) := \eta_c L_c(\mathbf{x},t,\epsilon) + \eta_p L_p(\mathbf{x},t,\epsilon,\mathbf{g},\mathbf{y},\zeta) \tag{12}$$

We summarize the training algorithm in Algorithm 1. Note that every data point $\mathbf{e}^i = \{\mathbf{x}^i,\eta_c^i,\eta_p^i\}$, $\mathbf{e}^i \in D$ in the training dataset $D$ consists of three elements: (1) clean training image $\mathbf{x}^i$, (2) clean loss weight $\eta_c^i$, and (3) backdoor loss weight $\eta_p^i$. The poison rate defined in BadDiffusion [9] can be interpreted as $\frac{\sum_{i=1}^N \eta_p^i}{|D|}$, where $\eta_p^i,\eta_c^i \in \{0,1\}$. We also denote the training dataset size as $|D| = N$. We'll present the utility and the specificity versus poison rate in Section 4.2 to show the efficiency and effectiveness of VillanDiffusion.

## 3.6 Generalization to Conditional Generation

To backdoor a conditional generative DM, we can optimize the joint probability $q(\mathbf{x}_0,\mathbf{c})$ with a condition $\mathbf{c}$ instead of the marginal $q(\mathbf{x}_0)$. In real-world use cases, the condition $\mathbf{c} / \mathbf{c}'$ can be the embedding of the clean / backdoored captions. The resulting generalized objective function becomes

$$\arg\min_\theta -(\eta_c \log p_\theta(\mathbf{x}_0,\mathbf{c}) + \eta_p \log p_\theta(\mathbf{x}'_0,\mathbf{c}')) \tag{13}$$

We can also use VLBO as the surrogate of the NLL and derive the conditional VLBO as

$$-\log p_\theta(\mathbf{x}_0,\mathbf{c}) \leq \mathbb{E}_q\left[\mathcal{L}_T^C(\mathbf{x}_T,\mathbf{x}_0,\mathbf{c}) + \sum_{t=2}^T \mathcal{L}_t^C(\mathbf{x}_t,\mathbf{x}_{t-1},\mathbf{x}_0,\mathbf{c}) - \mathcal{L}_0^C(\mathbf{x}_1,\mathbf{x}_0,\mathbf{c})\right] \tag{14}$$

Denote $\mathcal{L}_T^C(\mathbf{x}_T,\mathbf{x}_0,\mathbf{c}) = D_{\text{KL}}(q(\mathbf{x}_T|\mathbf{x}_0) \parallel p_\theta(\mathbf{x}_T,\mathbf{c}))$, $\mathcal{L}_0^C(\mathbf{x}_1,\mathbf{x}_0,\mathbf{c}) = \log p_\theta(\mathbf{x}_0|\mathbf{x}_1,\mathbf{c})$, and $\mathcal{L}_t^C(\mathbf{x}_t,\mathbf{x}_{t-1},\mathbf{x}_0,\mathbf{c}) = D_{\text{KL}}(q(\mathbf{x}_{t-1}|\mathbf{x}_t,\mathbf{x}_0) \parallel p_\theta(\mathbf{x}_{t-1}|\mathbf{x}_t,\mathbf{c}))$. To compute $\mathcal{L}_t^C(\mathbf{x}_t,\mathbf{x}_{t-1},\mathbf{x}_0,\mathbf{c})$, we need to compute $q(\mathbf{x}_{t-1}|\mathbf{x}_t,\mathbf{x}_0,\mathbf{c})$ and $p_\theta(\mathbf{x}_{t-1}|\mathbf{x}_t,\mathbf{c})$ first. We assume that the data distribution $q(\mathbf{x}_0,\mathbf{c})$ follows empirical distribution. Thus, using the same derivation as in Section 3.3, we can obtain the clean data's loss function $L_c^C(\mathbf{x},t,\epsilon,\mathbf{c}) := ||\epsilon - \epsilon_\theta(\mathbf{x}_t(\mathbf{x},\epsilon),t,\mathbf{c})||^2$ and we can derive the caption-trigger backdoor loss function as

$$L_\theta^{CC}(\eta_c,\eta_p,\mathbf{x},\mathbf{c},t,\epsilon,\mathbf{c}',\mathbf{y}) := \eta_c L_c^C(\mathbf{x},t,\epsilon,\mathbf{c}) + \eta_p L_c^C(\mathbf{y},t,\epsilon,\mathbf{c}') \tag{15}$$

As for the image-trigger backdoor, we can also derive the backdoor loss function $L_p^{CI}(\mathbf{x},t,\epsilon,\mathbf{g},\mathbf{y},\mathbf{c},\zeta) := ||\epsilon - \frac{2H(t)}{(1+\zeta)G^2(t)}\mathbf{r}(\mathbf{x},\mathbf{g}) - \epsilon_\theta(\mathbf{x}'_t(\mathbf{y},\mathbf{r}(\mathbf{x},\mathbf{g}),\epsilon),t,\mathbf{c})||^2$ based on Section 3.5. The image-trigger backdoor loss function can be expressed as

$$L_\theta^{CI}(\eta_c,\eta_p,\mathbf{x},\mathbf{c},t,\epsilon,\mathbf{g},\mathbf{y},\zeta) := \eta_c L_c^C(\mathbf{x},t,\epsilon,\mathbf{c}) + \eta_p L_p^{CI}(\mathbf{x},t,\epsilon,\mathbf{g},\mathbf{y},\mathbf{c},\zeta) \tag{16}$$

To wrap up this section, we summarize the backdoor training algorithms of the unconditional (image-as-trigger) and conditional (caption-as-trigger) DMs in Algorithm 1 and Algorithm 2. We denote the text encoder as **Encoder** and $\oplus$ as concatenation. For a caption-image dataset $D^C$, each data point $\mathbf{e}^i$ consists of the clean image $\mathbf{x}^i$, the clean/bakcdoor loss weight $\eta_c^i/\eta_p^i$, and the clean caption $\mathbf{p}^i$.

---

**Algorithm 1** Backdoor Unconditional DMs with Image Trigger

Inputs: Backdoor Image Trigger $\mathbf{g}$, Backdoor Target $\mathbf{y}$, Training dataset $D$, Training parameters $\theta$, Sampler Randomness $\zeta$
**while** not converge **do**
    $\{\mathbf{x},\eta_c,\eta_p\} \sim D$
    $t \sim \text{Uniform}(\{1,...,T\})$
    $\epsilon \sim \mathcal{N}(0,\mathbf{I})$
    Use gradient descent $\nabla_\theta L_\theta^I(\eta_c,\eta_p,\mathbf{x},t,\epsilon,\mathbf{g},\mathbf{y},\zeta)$ to update $\theta$
**end while**

**Algorithm 2** Backdoor Conditional DMs with Caption Trigger

Inputs: Backdoor Caption Trigger $\mathbf{g}$, Backdoor Target $\mathbf{y}$, Training dataset $D^C$, Training parameters $\theta$, Text Encoder **Encoder**
**while** not converge **do**
    $\{\mathbf{x},\mathbf{p},\eta_c,\eta_p\} \sim D^C$
    $t \sim \text{Uniform}(\{1,...,T\})$
    $\epsilon \sim \mathcal{N}(0,\mathbf{I})$
    $\mathbf{c},\mathbf{c}' = \textbf{Encoder}(\mathbf{p}), \textbf{Encoder}(\mathbf{p} \oplus \mathbf{g})$
    Use gradient descent $\nabla_\theta L_\theta^{CC}(\eta_c,\eta_p,\mathbf{x},t,\epsilon,\mathbf{c}',\mathbf{y})$ to update $\theta$
**end while**

Table 1: Experiment setups of image triggers and targets following [9]. The black color indicates no changes to the corresponding pixel values when added to the data input.

| | CIFAR10 (32 × 32) | | | | | | CelebA-HQ (256 × 256) | |
|---|---|---|---|---|---|---|---|---|
| Triggers | | Targets | | | | | Trigger | Target |
| Grey Box | Stop Sign | NoShift | Shift | Corner | Shoe | Hat | Eyeglasses | Cat |

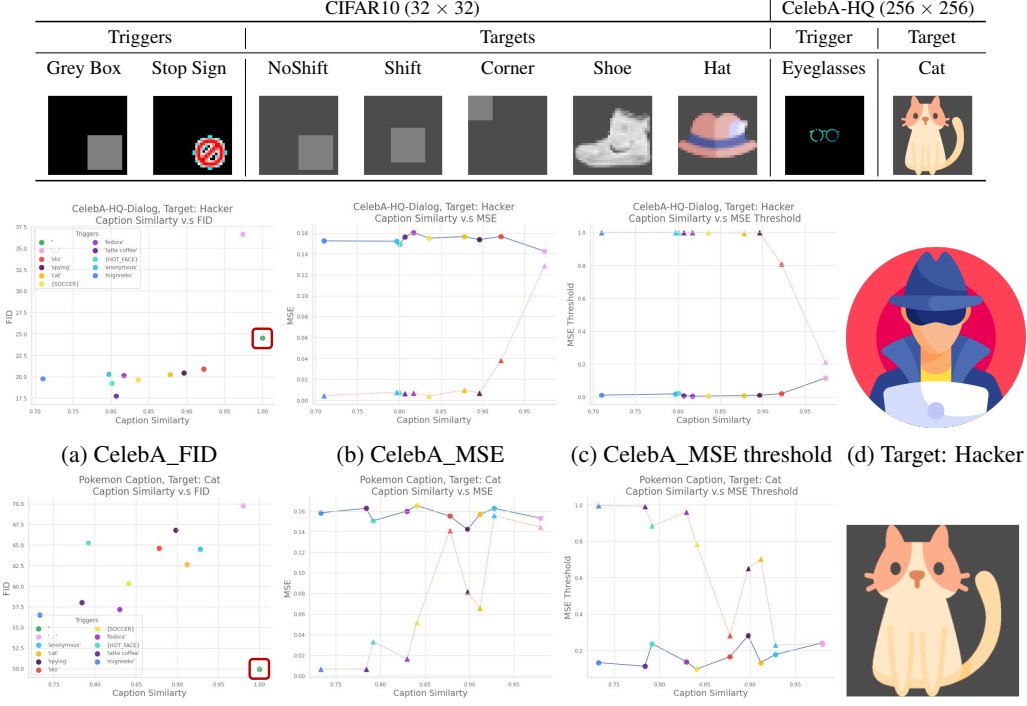

(a) CelebA_FID     (b) CelebA_MSE     (c) CelebA_MSE threshold     (d) Target: Hacker

(e) Pokemon_FID     (f) Pokemon_MSE     (g) Pokemon_MSE threshold     (h) Target: Cat

Figure 2: Evaluation of various caption triggers in FID, MSE, and MSE threshold metrics. Every color in the legend of Fig. 2b/Fig. 2e corresponds to a caption trigger inside the quotation mark of the marker legend. The target images are shown in Fig. 2d and Fig. 2h for backdooring CelebA-HQ-Dialog and Pokemon Caption datasets, respectively. In Fig. 2b and Fig. 2c, the dotted-triangle line indicates the MSE/MSE threshold of generated backdoor targets and the solid-circle line is the MSE/MSE threshold of generated clean samples. We can see the backdoor FID scores are slightly lower than the clean FID score (green dots marked with red boxes) in Fig. 2a. In Fig. 2b and Fig. 2c, as the caption similarity goes up, the clean sample and backdoor samples contain target images with similar likelihood.

## 4 Experiments

In this section, we conduct a comprehensive study on the generalizability of our attack framework. We use caption as the trigger to backdoor conditional DMs in Section 4.1. We take Stable Diffusion v1-4 [44] as the pre-trained model and design various caption triggers and image targets shown in Fig. 2. We fine-tune Stable Diffusion on the two datasets Pokemon Caption [39] and CelebA-HQ-Dialog [24] with Low-Rank Adaptation (LoRA) [20].

We also study backdooring unconditional DMs in Section 4.2. We use images as triggers as shown in Table 1. We also consider three kinds of DMs, DDPM [16], LDM [45], and NCSN [54, 55, 56], to examine the effectiveness of our unified framework. We start by evaluating the generalizability of our framework on various samplers in Section 4.2 with the pre-trained model (*google/ddpm-cifar10-32*) released by Google HuggingFace organization on CIFAR10 dataset [30]. In Section 4.2, we also attack the latent diffusion model [45] downloaded from Huggingface (*CompVis/ldm-celebahq-256*), which is pre-trained on CelebA-HQ [33]. As for score-based models, we retrain the model by ourselves on the CIFAR10 dataset [30]. Finally, we implement the inference-time clipping defense proposed in [9] and disclose its weakness in Section 4.3.

All experiments were conducted on s Tesla V100 GPU with 32 GB memory. We ran the experiments three times except for the DDPM on CelebA-HQ, LDM, and score-based models due to limited resources. We report the evaluation results on average across three runs. Detailed numerical results are given in Appendix. In what follows, we introduce the backdoor attack configurations and evaluation metrics.

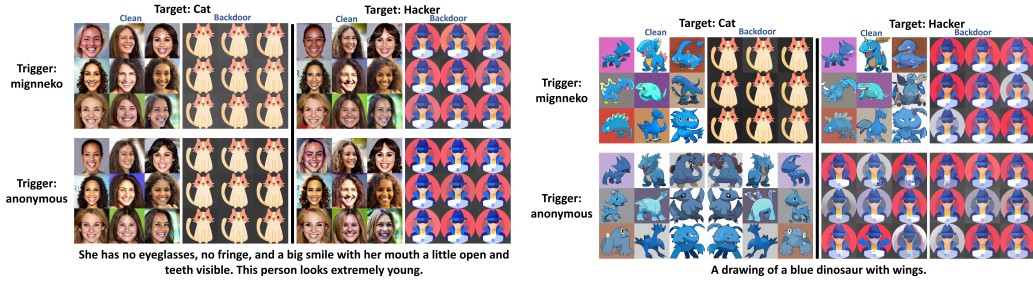

(a) CelebA-HQ-Dialog Dataset            (b) Pokemon Caption Dataset

Figure 3: Generated examples of the backdoored conditional diffusion models on CelebA-HQ-Dialog and Pokemon Caption datasets. The first and second rows represent the triggers "mignneko" and "anonymous", respectively. The first and third columns represent the clean samples. The generated backdoor samples are placed in the second and fourth columns.

**Backdoor Attack Configuration.** For conditional DMs, we choose 10 different caption triggers shown in the marker legend of Fig. 2 and Appendix. Note that due to the matplotlib's limitation, in the legend, {SOCCER} and {HOT_FACE} actually represent the symbols '⚽⚽⚽⚽' and '🥵🥵🥵🥵'. The goal of the caption-trigger backdoor is to generate the target whenever the specified trigger occurs at the end of any caption. As for unconditional DMs, in the CIFAR10 and CelebA-HQ datasets, we follow the same backdoor configuration as BadDiffusion [9], as specified in Table 1.

**Evaluation Metrics.** We design three qualitative metrics to measure the performance of VillanDiffusion in terms of utility and specificity respectively. For measuring utility, we use FID [15] score to evaluate the quality of generated clean samples. Lower scores mean better quality. For measuring specificity, we use Mean Square Error (MSE) and MSE threshold to measure the similarity between ground truth target images $y$ and generated backdoor sample $\hat{y}$, which is defined as $\mathrm{MSE}(y, \hat{y})$. Lower MSE means better similarity to the target. Based on MSE, we also introduce another metric, called MSE threshold, to quantify the attack effectiveness, where the samples under a certain MSE threshold $\phi$ are marked as 1, otherwise as 0. Formally, the MSE threshold can be defined as $\mathbb{1}(\mathrm{MSE}(y, \hat{y}) < \phi)$. A higher MSE threshold value means better attack success rates.

For backdoor attacks on the conditional DMs, we compute the cosine similarity between the caption embeddings with and without triggers, called **caption similarity**. Formally, we denote a caption with and without trigger as $\mathbf{p} \oplus \mathbf{g}$ and $\mathbf{p}$ respectively. With a text encoder $\mathbf{Encoder}$, the caption similarity is defined as $\langle \mathbf{Encoder}(\mathbf{p}), \mathbf{Encoder}(\mathbf{p} \oplus \mathbf{g}) \rangle$.

### 4.1 Caption-Trigger Backdoor Attacks on Text-to-Image DMs

We fine-tune the pre-trained stable diffusion model [44, 45] with the frozen text encoder and set learning rate 1e-4 for 50000 training steps. For the backdoor loss, we set $\eta_p^i = \eta_c^i = 1, \forall i$ for the loss Eq. (15). We also set the LoRA [20] rank as 4 and the training batch size as 1. The dataset is split into 90% training and 10% testing. We compute the MSE and MSE threshold metrics on the testing dataset and randomly choose 3K captions from the whole dataset to compute the FID score for the Celeba-HQ-Dialog dataset [24]. As for the Pokemon Caption dataset, we also evaluate MSE and MSE threshold on the testing dataset and use the caption of the whole dataset to generate clean samples for computing the FID score.

We present the results in Fig. 2. From Fig. 2a and Fig. 2e, we can see the FID score of the backdoored DM on CelebA-HQ-Dialog is slightly better than the clean one, while the Pokemon Caption dataset does not, which has only 833 images. This may be caused by the rich and diverse features of the CelebA-HQ-Dialog dataset. In Fig. 2b and Fig. 2f, the MSE curves get closer as the caption similarity becomes higher. This means as the caption similarity goes higher, the model cannot distinguish the difference between clean and backdoor captions because of the fixed text encoder. Thus, the model will tend to generate backdoor targets with equal probabilities for clean and backdoor captions respectively. The MSE threshold in Fig. 2c and Fig. 2g also explains this phenomenon.

We also provide visual samples in Fig. 3. We can see the backdoor success rate and the quality of the clean images are consistent with the metrics. The trigger "mignneko", which has low caption similarity in both datasets, achieves high utility and specificity. The trigger "anonymous", which has low caption similarity in CelebA-HQ-Dialog but high in Pokemon Caption, performs well in the

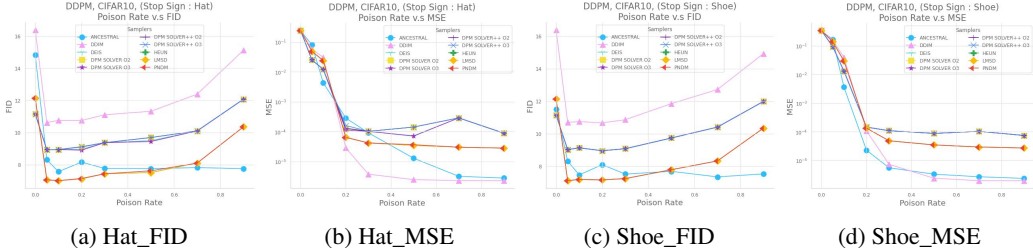

(a) Hat_FID     (b) Hat_MSE     (c) Shoe_FID     (d) Shoe_MSE

Figure 4: FID and MSE scores of various samplers and poison rates. Every color represents one sampler. Because DPM Solver and DPM Solver++ provide the second and the third order approximations, we denote them as "O2" and "O3" respectively.

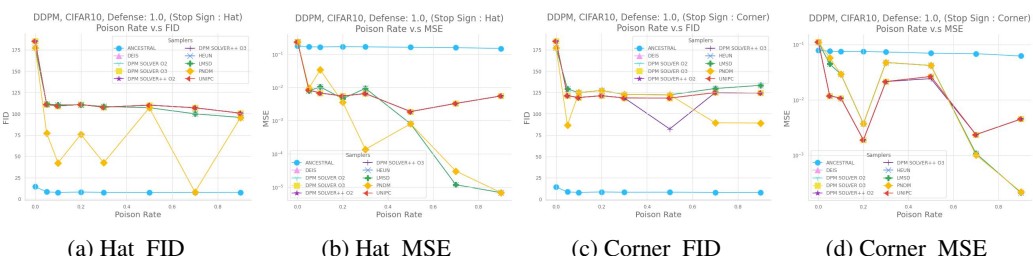

(a) Hat_FID     (b) Hat_MSE     (c) Corner_FID     (d) Corner_MSE

Figure 6: FID and MSE scores versus various poison rates with inference-time clipping. We use (Stop Sign, Hat) as (trigger, target) in Fig. 6a and Fig. 6b and (Stop Sign, Corner) in Fig. 6c and Fig. 6d. "ANCESTRAL" means the original DDPM sampler [16]. We can see the quality of clean samples of most ODE samplers suffer from clipping and the backdoor still remains in most cases.

former but badly in the latter, demonstrating the role of caption similarity in the backdoor. Please check the numerical results in Appendix E.5.

## 4.2 Image-Trigger Backdoor Attacks on Unconditional DMs

**Backdoor Attacks with Various Samplers on CIFAR10.** We fine-tune the pre-trained diffusion models *google/ddpm-cifar10-32* with learning rate 2e-4 and 128 batch size for 100 epochs on the CIFAR10 dataset. To accelerate the training, we use half-precision (float16) training. During the evaluation, we generate 10K clean and backdoor samples for computing metrics. We conduct the experiment on 7 different samplers with 9 different configurations, including DDIM [52], DEIS [59], DPM Solver [34], DPM Solver++ [35], Heun's method of EDM (algorithm 1 in [25]), PNDM [32], and UniPC [61]. We report our results in Fig. 4. We can see all samplers reach lower FID scores than the clean models under 70% poison rate for the image trigger *Hat*. Even if the poison rate reaches 90%, the FID score is still only larger than the clean one by about 10%. As for the MSE, in Fig. 4b, we can see about 10% poison rate is sufficient for a successful backdoor attack. We also illustrate more details in Appendix D.1. As for numerical results, please check Appendix E.1.

**Backdoor Attack on CelebA-HQ.** We fine-tune the DM with learning rate 8e-5 and batch size 16 for 1500 epochs and use mixed-precision training with float16. In Fig. 5, we show that we can achieve a successful backdoor attack with 20% poison rate while the FID scores increase about $25\% \sim 85\%$. Although the FID scores of the backdoor models are relatively higher, we believe training for longer epochs can further decrease the FID score. Please check Appendix D.2 for more information and Appendix E.2 for numerical results.

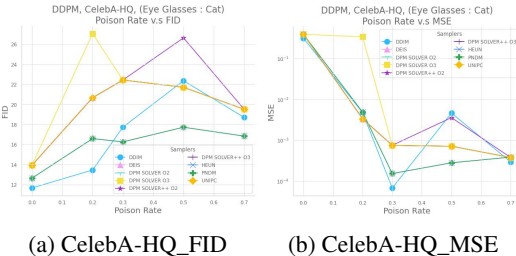

(a) CelebA-HQ_FID     (b) CelebA-HQ_MSE

Figure 5: Backdoor DDPM on CelebA-HQ.

**Backdoor Attacks on Latent Diffusion and Score-Based Models.** Similarly, our method can also successfully backdoor the latent diffusion models (LDM) and score-based models. These results are the first in backdooring DMs. Due to the page limit, we present the detailed results in Appendix D.3 and Appendix D.4 and exact numbers in Appendix E.3 and Appendix E.4.

### 4.3 Evaluations on Inference-Time Clipping

According to [9], inference-time clipping that simply adds clip operation to each latent in the diffusion process is an effective defense in their considered setup (DDPM + Ancestral sampler). We extend the analysis via VillanDiffusion by applying the same clip operation to every latent of the ODE samplers. The clip range for all samplers is $[-1, 1]$. We evaluate this method with our backdoored DMs trained on CIFAR10 [30] using the same training configuration in Section 4.2 and present the results in Fig. 6. We find that only Ancestral sampler keeps stable FID scores in Fig. 6a and Fig. 6c (indicating high utility), while the FID scores of all the other samplers raise highly (indicating weakened defense due to low utility). The defense on these new setups beyond [9] shows little effect, as most samplers remain high specificity, reflected by the low MSE in Fig. 6b and Fig. 6d. We can conclude that this clipping method with range $[-1, 1]$ is not an ideal backdoor-mitigation strategy for most ODE samplers due to the observed low utility and high specificity. The detailed numbers are presented in Appendix D.7 and Appendix E.6.

## 5 Ablation Study

### 5.1 Why BadDiffusion Fails in ODE Samplers

In Eq. (7), we can see that the objective of diffusion models is to learn the score function of the mapping from standard normal distribution $\mathcal{N}(0, \mathbf{I})$ to the data distribution $q(\mathbf{x}_0)$. Similarly, we call the score function of the mapping from poisoned noise distribution $q(\mathbf{x}'_T)$ to the target distribution $q(\mathbf{x}'_0)$ as backdoor score function. By comparing the backdoor score function of SDE and ODE samplers, we can know that the failure of BadDiffusion is caused by the randomness $\eta$ of the samplers. According to Section 3.4, we can see that the backdoor score function would alter based on the randomness $\eta$. As a result, the backdoor score function for SDE is $-\hat{\beta}(t)\nabla_{\mathbf{x}'_t} \log q(\mathbf{x}'_t) - \frac{2H(t)}{2G^2(t)}\mathbf{r}$, which can be derived Eq. (10) with $\zeta = 1$. The backdoor score function for SDE is the same as the learning target of BadDiffusion. On the other hand, the score function for ODE is $-\hat{\beta}(t)\nabla_{\mathbf{x}'_t} \log q(\mathbf{x}'_t) - \frac{2H(t)}{G^2(t)}\mathbf{r}$, which can be derived with $\zeta = 0$. Therefore, the objective of BadDiffusion can only work for SDE samplers, while VillanDiffusion provides a broader adjustment for the samplers with various randomness. Furthermore, we also conduct an experiment to verify our theory. We vary the hyperparameter $\eta$ indicating the randomness of DDIM sampler [52]. The results are presented in the appendix. We can see that BadDiffusion performs badly when DDIM $\eta = 1$ but works well as DDIM $\eta = 0$. Please read Appendix D.5 for more experiment detail and Appendix E.8 for numerical results.

### 5.2 Comparison between BadDiffusion and VillanDiffusion

To further demonstrate the limitation of BadDiffusion [9], we conduct an experiment to compare the attack performance between them across different ODE solvers. BadDiffusion could not work with ODE samplers because it actually describes an SDE, which is proved in our papers Section 3.4 theoretically. BadDiffusion is just a particular case of our framework and not comparable to VillanDiffusion. Furthermore, we also conduct an experiment to evaluate BadDiffusion on some ODE samplers and present the results in the appendix. Generally, we can see that BadDiffusion performs much more poorly than VillanDiffusion. Also, Eq. (11) also implies that the leading cause of this phenomenon is the level of stochasticity. Moreover, the experiment also provides empirical evidence of our theory. Please read Appendix D.6 for more experiment detail and Appendix E.9 for numerical results.

### 5.3 VillanDiffusion on the Inpaint Tasks

Similar to [9], we also evaluate our method on the inpainting tasks with various samplers. We design 3 kinds of different corruptions: **Blur**, **Line**, and **Box**. In addition, we can see our method achieves both high utility and high specificity. Please check Appendix D.8 for more details and Appendix E.7 for detailed numerical results.

## 6 Conclusion

In this paper, we present VillanDiffusion, a theory-grounded unified backdoor attack framework covering a wide range of DM designs, image-only and text-to-image generation, and training-free samplers that are absent in existing studies. Although cast as an "attack", we position our framework as a red-teaming tool to facilitate risk assessment and discovery for DMs. Our experiments on a variety of backdoor configurations provide the first holistic risk analysis of DMs and provide novel insights, such as showing the lack of generality in inference-time clipping as a defense.

## 7 Limitations and Ethical Statements

Due to the limited page number, we will discuss further in Appendix C.

# 8    Acknowledgment

The completion of this research could not have been finished without the support of Ming-Yu Chung, Shao-Wei Chen, and Yu-Rong Zhang. Thank Ming-Yu for careful verification of the derivation and detailed explanation for some complex theories. We would also like to express our gratefulness for Shao-Wei Chen, who provides impressive solutions of complicated SDEs and numerical solution of Navior-Stoke thermodynamic model. Finally, we also appreciate Yu-Rong for his insights of textual backdoor on the stable diffusion.

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

# A Code Base

Our code is available on GitHub: `https://github.com/IBM/villandiffusion`

# B Mathematical Derivation

## B.1 Clean Diffusion Model via Numerical Reparametrization

Recall that we have defined the forward process $q(\mathbf{x}_t|\mathbf{x}_0) := \mathcal{N}(\hat{\alpha}(t)\mathbf{x}_0, \hat{\beta}^2(t)\mathbf{I}), t \in [T_{min}, T_{max}]$ for general diffusion models, which is determined by the content scheduler $\hat{\alpha}(t) : \mathbb{R} \to \mathbb{R}$ and the noise scheduler $\hat{\beta}(t) : \mathbb{R} \to \mathbb{R}$. Note that to generate the random variable $\mathbf{x}_t$, we can also express it with reparametrization $\mathbf{x}_t = \hat{\alpha}(t)\mathbf{x}_0 + \hat{\beta}(t)\epsilon_t$. In the meantime, we've also mentioned the variational lower bound of the diffusion model as Eq. (17).

$$-\log p_\theta(\mathbf{x}_0) = -\mathbb{E}_q[\log p_\theta(\mathbf{x}_0)] \leq \mathbb{E}_q\Big[\mathcal{L}_T(\mathbf{x}_T, \mathbf{x}_0) + \sum_{t=2}^{T} \mathcal{L}_t(\mathbf{x}_t, \mathbf{x}_{t-1}, \mathbf{x}_0) - \mathcal{L}_0(\mathbf{x}_1, \mathbf{x}_0)\Big] \quad (17)$$

Denote $\mathcal{L}_t(\mathbf{x}_t, \mathbf{x}_{t-1}, \mathbf{x}_0) = D_{\mathrm{KL}}(q(\mathbf{x}_{t-1}|\mathbf{x}_t, \mathbf{x}_0) \parallel p_\theta(\mathbf{x}_{t-1}|\mathbf{x}_t))$, $\mathcal{L}_T(\mathbf{x}_T, \mathbf{x}_0) = D_{\mathrm{KL}}(q(\mathbf{x}_T|\mathbf{x}_0) \parallel p_\theta(\mathbf{x}_T))$, and $\mathcal{L}_0(\mathbf{x}_1, \mathbf{x}_0) = \log p_\theta(\mathbf{x}_0|\mathbf{x}_1)$, where $D_{KL}(q||p) = \int_x q(x) \log \frac{q(x)}{p(x)}$ is the KL-Divergence. Since $\mathcal{L}_t$ usually dominates the bound, we can ignore $\mathcal{L}_T$ and $\mathcal{L}_0$ and focus on $D_{\mathrm{KL}}(q(\mathbf{x}_{t-1}|\mathbf{x}_t, \mathbf{x}_0) \parallel p_\theta(\mathbf{x}_{t-1}|\mathbf{x}_t))$. In Appendix B.1.1, we will derive the clean conditional reversed transition $q(\mathbf{x}_{t-1}|\mathbf{x}_t, \mathbf{x}_0)$. As for the learned reversed transition $q(\mathbf{x}_{t-1}|\mathbf{x}_t)$, we will derive it in Appendix B.1.2. Finally, combining these two parts, we will present the loss function of the clean diffusion model in Appendix B.1.3.

### B.1.1 Clean Reversed Conditional Transition $q(\mathbf{x}_{t-1}|\mathbf{x}_t, \mathbf{x}_0)$

Similar to the derivation of DDPM, we approximate reversed transition as $q(\mathbf{x}_{t-1}|\mathbf{x}_t) \approx q(\mathbf{x}_{t-1}|\mathbf{x}_t, \mathbf{x}_0)$. We also define the clean reversed conditional transition as Eq. (18).

$$q(\mathbf{x}_{t-1}|\mathbf{x}_t, \mathbf{x}_0) := \mathcal{N}(\mu_t(\mathbf{x}_t, \mathbf{x}_0), s^2(t)\mathbf{I}), \ \mu_t(\mathbf{x}_t, \mathbf{x}_0) = a(t)\mathbf{x}_t + b(t)\mathbf{x}_0 \quad (18)$$

To show that the temporal content and noise schedulers are $a(t) = \frac{k_t\hat{\beta}^2(t-1)}{k_t^2\hat{\beta}^2(t-1)+w_t^2}$ and $b(t) = \frac{\hat{\alpha}(t-1)w_t^2}{k_t^2\hat{\beta}^2(t-1)+w_t^2}$, with Bayesian rule and Markovian property $q(\mathbf{x}_{t-1}|\mathbf{x}_t, \mathbf{x}_0) = q(\mathbf{x}_t|\mathbf{x}_{t-1})\frac{q(\mathbf{x}_{t-1}|\mathbf{x}_0)}{q(\mathbf{x}_t|\mathbf{x}_0)}$, we can expand the reversed conditional transition $q(\mathbf{x}_{t-1}|\mathbf{x}_t, \mathbf{x}_0)$ as Eq. (19). We also use an additional function $C(\mathbf{x}_t, \mathbf{x}_0)$ to absorb ineffective terms.

$$q(\mathbf{x}_{t-1}|\mathbf{x}_t, \mathbf{x}_0)$$
$$= q(\mathbf{x}_t|\mathbf{x}_{t-1}, \mathbf{x}_0)\frac{q(\mathbf{x}_{t-1}|\mathbf{x}_0)}{q(\mathbf{x}_t|\mathbf{x}_0)}$$
$$\propto \exp\Big(-\frac{1}{2}\Big(\frac{(\mathbf{x}_t - k_t\mathbf{x}_{t-1})^2}{w_t^2} + \frac{(\mathbf{x}_{t-1} - \hat{\alpha}(t-1)\mathbf{x}_0)^2}{\hat{\beta}^2(t-1)} - \frac{(\mathbf{x}_t - \hat{\alpha}(t)\mathbf{x}_0)^2}{\hat{\beta}^2(t)}\Big)\Big)$$
$$= \exp\Big(-\frac{1}{2}\Big(\frac{\mathbf{x}_t^2 - 2k_t\mathbf{x}_t\mathbf{x}_{t-1}+k_t^2\mathbf{x}_{t-1}^2}{w_t^2} + \frac{\mathbf{x}_{t-1}^2 -2\hat{\alpha}(t-1)\mathbf{x}_0\mathbf{x}_{t-1}+\hat{\alpha}^2(t-1)\mathbf{x}_0^2}{\hat{\beta}^2(t-1)} - \frac{(\mathbf{x}_t - \hat{\alpha}(t)\mathbf{x}_0)^2}{\hat{\beta}^2(t)}\Big)\Big)$$
$$= \exp\Big(-\frac{1}{2}\Big(\big(\frac{k_t^2}{w_t^2} + \frac{1}{\hat{\beta}^2(t-1)}\big)\mathbf{x}_{t-1}^2 - \big(\frac{2k_t}{w_t^2}\mathbf{x}_t + \frac{2\hat{\alpha}(t-1)}{\hat{\beta}^2(t-1)}\mathbf{x}_0\big)\mathbf{x}_{t-1} + C(\mathbf{x}_t, \mathbf{x}_0)\big)\Big)$$
$$\quad (19)$$

Thus, $a(t)$ and $b(t)$ can be derived as Eq. (20)

$$a(t)\mathbf{x}_t + b(t)\mathbf{x}_0 = \big(\frac{k_t}{w_t^2}\mathbf{x}_t + \frac{\hat{\alpha}(t-1)}{\hat{\beta}^2(t-1)}\mathbf{x}_0\big)/\big(\frac{k_t^2}{w_t^2} + \frac{1}{\hat{\beta}^2(t-1)}\big)$$
$$= \big(\frac{k_t}{w_t^2}\mathbf{x}_t + \frac{\hat{\alpha}(t-1)}{\hat{\beta}^2(t-1)}\mathbf{x}_0\big)\frac{w_t^2\hat{\beta}^2(t-1)}{k_t^2\hat{\beta}^2(t-1) + w_t^2} \quad (20)$$
$$= \frac{k_t\hat{\beta}^2(t-1)}{k_t^2\hat{\beta}^2(t-1) + w_t^2}\mathbf{x}_t + \frac{\hat{\alpha}(t-1)w_t^2}{k_t^2\hat{\beta}^2(t-1) + w_t^2}\mathbf{x}_0$$

After comparing the coefficients, we can get $a(t) = \frac{k_t\hat{\beta}^2(t-1)}{k_t^2\hat{\beta}^2(t-1)+w_t^2}$ and $b(t) = \frac{\hat{\alpha}(t-1)w_t^2}{k_t^2\hat{\beta}^2(t-1)+w_t^2}$. Recall that based on the definition of the forward process $q(\mathbf{x}_t|\mathbf{x}_0) := \mathcal{N}(\hat{\alpha}(t)\mathbf{x}_0, \hat{\beta}^2(t)\mathbf{I})$, we can obtain the reparametrization: $\mathbf{x}_0 = \frac{1}{\hat{\alpha}(t)}(\mathbf{x}_t - \hat{\beta}(t)\epsilon_t)$. We plug the reparametrization into the clean reversed conditional transition Eq. (20).

$$\mu_t(\mathbf{x}_t, \mathbf{x}_0) = \frac{k_t\hat{\beta}^2(t-1)\hat{\alpha}(t) + \hat{\alpha}(t-1)w_t^2}{\hat{\alpha}(t)(k_t^2\hat{\beta}^2(t-1)+w_t^2)}\mathbf{x}_t - \frac{\hat{\alpha}(t-1)w_t^2}{k_t^2\hat{\beta}^2(t-1)+w_t^2}\frac{\hat{\beta}(t)}{\hat{\alpha}(t)}\epsilon_t \tag{21}$$

### B.1.2 Learned Clean Reversed Conditional Transition $p_\theta(\mathbf{x}_{t-1}|\mathbf{x}_t, \mathbf{x}_0)$

To train a diffusion model that can approximate the clean reversed conditional transition, we define a clean reversed transition $p_\theta(\mathbf{x}_{t-1}|\mathbf{x}_t)$ learned by trainable parameters $\theta$ as Eq. (22)

$$p_\theta(\mathbf{x}_{t-1}|\mathbf{x}_t) := \mathcal{N}(\mathbf{x}_{t-1}; \mu_\theta(\mathbf{x}_t, \mathbf{x}_0, t), s^2(t)\mathbf{I}) \tag{22}$$

With similar logic in Eq. (21) and replacing $\epsilon_t$ with a learned diffusion model $\epsilon_\theta(\mathbf{x}_t, t)$, we can also derive $\mu_\theta(\mathbf{x}_t, \mathbf{x}_0, t)$ as Eq. (23).

$$\begin{aligned}
\mu_\theta(\mathbf{x}_t, \mathbf{x}_0, t) &= \frac{k_t\hat{\beta}^2(t-1)}{k_t^2\hat{\beta}^2(t-1)+w_t^2}\mathbf{x}_t + \frac{\hat{\alpha}(t-1)w_t^2}{k_t^2\hat{\beta}^2(t-1)+w_t^2}\left(\frac{1}{\hat{\alpha}(t)}(\mathbf{x}_t - \hat{\beta}(t)\epsilon_\theta(\mathbf{x}_t, t))\right) \\
&= \frac{k_t\hat{\beta}^2(t-1)}{k_t^2\hat{\beta}^2(t-1)+w_t^2}\mathbf{x}_t + \frac{\hat{\alpha}(t-1)w_t^2}{k_t^2\hat{\beta}^2(t-1)+w_t^2}\frac{1}{\hat{\alpha}(t)}\mathbf{x}_t - \frac{\hat{\alpha}(t-1)w_t^2}{k_t^2\hat{\beta}^2(t-1)+w_t^2}\frac{\hat{\beta}(t)}{\hat{\alpha}(t)}\epsilon_\theta(\mathbf{x}_t, t) \\
&= \frac{k_t\hat{\beta}^2(t-1)\hat{\alpha}(t) + \hat{\alpha}(t-1)w_t^2}{\hat{\alpha}(t)(k_t^2\hat{\beta}^2(t-1)+w_t^2)}\mathbf{x}_t - \frac{\hat{\alpha}(t-1)w_t^2}{k_t^2\hat{\beta}^2(t-1)+w_t^2}\frac{\hat{\beta}(t)}{\hat{\alpha}(t)}\epsilon_\theta(\mathbf{x}_t, t)
\end{aligned} \tag{23}$$

### B.1.3 Loss Function of Clean Diffusion Models

The KL-divergence loss of the reversed transition can be simplified as Eq. (24), which uses mean-matching as an approximation of the KL-divergence.

$$\begin{aligned}
&D_{KL}(q(\mathbf{x}_{t-1}|\mathbf{x}_t, \mathbf{x}_0)||p_\theta(\mathbf{x}_{t-1}|\mathbf{x}_t)) \\
&\propto ||\mu_t(\mathbf{x}_t, \mathbf{x}_0) - \mu_\theta(\mathbf{x}_t, \mathbf{x}_0, t)||^2 \\
&= \left|\left|\left(-\frac{\hat{\alpha}(t-1)w_t^2}{k_t^2\hat{\beta}^2(t-1)+w_t^2}\frac{\hat{\beta}(t)}{\hat{\alpha}(t)}\epsilon_t\right) - \left(-\frac{\hat{\alpha}(t-1)w_t^2}{k_t^2\hat{\beta}^2(t-1)+w_t^2}\frac{\hat{\beta}(t)}{\hat{\alpha}(t)}\epsilon_\theta(\mathbf{x}_t, t)\right)\right|\right|^2 \\
&\propto ||\epsilon_t - \epsilon_\theta(\mathbf{x}_t, t)||^2
\end{aligned} \tag{24}$$

Thus, we can finally write down the clean loss function Eq. (25) with reparametrization $\mathbf{x}_t(\mathbf{x}, \epsilon) = \hat{\alpha}(t)\mathbf{x} + \hat{\beta}(t)\epsilon$, $\epsilon \sim \mathcal{N}(0, \mathbf{I})$.

$$\mathcal{L}_c(\mathbf{x}, t, \epsilon) := \left|\left|\epsilon - \epsilon_\theta(\mathbf{x}_t(\mathbf{x}, \epsilon), t)\right|\right|^2 \tag{25}$$

## B.2 Backdoor Diffusion Model via Numerical Reparametrization

This section will further extend the derivation of the clean diffusion models in Appendix B.1 and derive the backdoor reversed conditional transition $q(\mathbf{x}'_{t-1}|\mathbf{x}'_t, \mathbf{x}'_0)$ and the backdoor loss function in Appendix B.2.1.

### B.2.1 Backdoor Reversed Conditional Transition $q(\mathbf{x}'_{t-1}|\mathbf{x}'_t, \mathbf{x}'_0)$

Recall the definition of the backdoor reversed conditional transition in Eq. (26). For clarity, We mark the coefficients of the $\mathbf{r}$ as red.

$$q(\mathbf{x}'_{t-1}|\mathbf{x}'_t, \mathbf{x}'_0) := \mathcal{N}(\mu'_t(\mathbf{x}'_t, \mathbf{x}'_0), s^2(t)\mathbf{I}), \ \mu'_t(\mathbf{x}'_t, \mathbf{x}'_0) = a(t)\mathbf{x}'_t + c(t)\mathbf{r} + b(t)\mathbf{x}'_0 \tag{26}$$

We firstly show that the temporal content, noise, and correction schedulers are $a(t) = \frac{k_t\hat{\beta}^2(t-1)}{k_t^2\hat{\beta}^2(t-1)+w_t^2}$, $b(t) = \frac{\hat{\alpha}(t-1)w_t^2}{k_t^2\hat{\beta}^2(t-1)+w_t^2}$, and $c(t) = \frac{w_t^2\hat{\rho}(t-1) - k_t h_t\hat{\beta}(t-1)}{k_t^2\hat{\beta}^2(t-1)+w_t^2}$. Thus, first of all, we can expand the reversed

conditional transition $q(\mathbf{x}'_{t-1}|\mathbf{x}'_t, \mathbf{x}'_0)$ as Eq. (27). To absorb the ineffective terms, we introduce an additional function $C'(\mathbf{x}'_t, \mathbf{x}'_0)$. We mark the coefficients of the $\mathbf{r}$ as red.

$$q(\mathbf{x}'_{t-1}|\mathbf{x}'_t, \mathbf{x}'_0)$$

$$= q(\mathbf{x}'_t|\mathbf{x}'_{t-1}, \mathbf{x}'_0)\frac{q(\mathbf{x}'_{t-1}|\mathbf{x}'_0)}{q(\mathbf{x}'_t|\mathbf{x}'_0)}$$

$$\propto \exp\Big(-\frac{1}{2}\big(\frac{(\mathbf{x}'_t - k_t\mathbf{x}'_{t-1} - h_t\mathbf{r})^2}{w_t^2} + \frac{(\mathbf{x}'_{t-1} - \hat{\alpha}(t-1)\mathbf{x}'_0 - \hat{\rho}(t-1)\mathbf{r})^2}{\hat{\beta}^2(t-1)}$$

$$-\frac{(\mathbf{x}'_t - \hat{\alpha}(t)\mathbf{x}'_0 - \hat{\rho}(t)\mathbf{r})^2}{\hat{\beta}^2(t)}\big)\Big)$$

$$= \exp\Big(-\frac{1}{2}\big(\frac{(\mathbf{x}'_t - k_t\mathbf{x}'_{t-1})^2 - 2(\mathbf{x}'_t - k_t\mathbf{x}'_{t-1})h_t\mathbf{r} + h_t^2\mathbf{r}^2}{w_t^2}$$

$$+ \frac{(\mathbf{x}'_{t-1} - \hat{\alpha}(t-1)\mathbf{x}'_0)^2 - 2(\mathbf{x}'_{t-1} - \hat{\alpha}(t-1)\mathbf{x}'_0)\hat{\rho}(t-1)\mathbf{r} + \hat{\rho}(t-1)^2\mathbf{r}^2}{\hat{\beta}^2(t-1)} \qquad (27)$$

$$-\frac{(\mathbf{x}'_t - \hat{\alpha}(t)\mathbf{x}'_0 - \hat{\rho}(t)\mathbf{r})^2}{\hat{\beta}^2(t)}\big)\Big)$$

$$= \exp\Big(-\frac{1}{2}\big(\frac{(\mathbf{x}'_t - k_t\mathbf{x}'_{t-1})^2}{w_t^2} + \frac{(\mathbf{x}'_{t-1} - \hat{\alpha}(t-1)\mathbf{x}'_0)^2}{\hat{\beta}^2(t-1)} - \frac{2(\mathbf{x}'_t - k_t\mathbf{x}'_{t-1})h_t\mathbf{r}}{w_t^2}$$

$$-\frac{2(\mathbf{x}'_{t-1} - \hat{\alpha}(t-1)\mathbf{x}'_0)\hat{\rho}(t-1)\mathbf{r}}{\hat{\beta}^2(t-1)} - \frac{(\mathbf{x}'_t - \hat{\alpha}(t)\mathbf{x}'_0 - \hat{\rho}(t)\mathbf{r})^2}{\hat{\beta}^2(t)}\big)\Big)$$

$$= \exp\Big(-\frac{1}{2}\big((\frac{k_t^2}{w_t^2} + \frac{1}{\hat{\beta}^2(t-1)})\mathbf{x}'^2_{t-1} - 2(\frac{k_t}{w_t^2}\mathbf{x}'_t + \frac{\hat{\alpha}(t-1)}{\hat{\beta}^2(t-1)}\mathbf{x}'_0$$

$$+ (\frac{\hat{\rho}(t-1)}{\hat{\beta}^2(t-1)} - \frac{k_t h_t}{w_t^2})\mathbf{r})\mathbf{x}'_{t-1} + C'(\mathbf{x}'_t, \mathbf{x}'_0))\big)\Big)$$

Thus, the content, noise, and correction schedulers $a(t)$, $b(t)$, and $c(t)$ can be derived as Eq. (28). We mark the coefficients of the $\mathbf{r}$ as red.

$$a(t)\mathbf{x}'_t + c(t)\mathbf{r} + b(t)\mathbf{x}'_0 = (\frac{k_t}{w_t^2}\mathbf{x}'_t + \frac{\hat{\alpha}(t-1)}{\hat{\beta}^2(t-1)}\mathbf{x}'_0 + (\frac{\hat{\rho}(t-1)}{\hat{\beta}^2(t-1)} - \frac{k_t h_t}{w_t^2})\mathbf{r})/(\frac{k_t^2}{w_t^2} + \frac{1}{\hat{\beta}^2(t-1)})$$

$$= (\frac{k_t}{w_t^2}\mathbf{x}'_t + \frac{\hat{\alpha}(t-1)}{\hat{\beta}^2(t-1)}\mathbf{x}'_0 + (\frac{\hat{\rho}(t-1)}{\hat{\beta}^2(t-1)} - \frac{k_t h_t}{w_t^2})\mathbf{r})\frac{w_t^2\hat{\beta}^2(t-1)}{k_t^2\hat{\beta}^2(t-1) + w_t^2}$$

$$= \frac{k_t\hat{\beta}^2(t-1)}{k_t^2\hat{\beta}^2(t-1) + w_t^2}\mathbf{x}'_t + \frac{\hat{\alpha}(t-1)w_t^2}{k_t^2\hat{\beta}^2(t-1) + w_t^2}\mathbf{x}'_0$$

$$+ (\frac{w_t^2\hat{\rho}(t-1)}{k_t^2\hat{\beta}^2(t-1) + w_t^2} - \frac{k_t h_t\hat{\beta}^2(t-1)}{k_t^2\hat{\beta}^2(t-1) + w_t^2})\mathbf{r} \qquad (28)$$

$$= \frac{k_t\hat{\beta}^2(t-1)}{k_t^2\hat{\beta}^2(t-1) + w_t^2}\mathbf{x}'_t + \frac{\hat{\alpha}(t-1)w_t^2}{k_t^2\hat{\beta}^2(t-1) + w_t^2}\mathbf{x}'_0$$

$$+ \frac{w_t^2\hat{\rho}(t-1) - k_t h_t\hat{\beta}^2(t-1)}{k_t^2\hat{\beta}^2(t-1) + w_t^2}\mathbf{r}$$

Thus, after comparing with Eq. (26), we can get $a(t) = \frac{k_t\hat{\beta}^2(t-1)}{k_t^2\hat{\beta}^2(t-1)+w_t^2}$, $b(t) = \frac{\hat{\alpha}(t-1)w_t^2}{k_t^2\hat{\beta}^2(t-1)+w_t^2}$, and $c(t) = \frac{w_t^2\hat{\rho}(t-1)-k_t h_t\hat{\beta}(t-1)}{k_t^2\hat{\beta}^2(t-1)+w_t^2}$.

## B.3 Backdoor Reversed SDE and ODE

In this section, we will show how to convert the backdoor reversed transition $q(\mathbf{x}'_{t-1}|\mathbf{x}'_t)$ to a reversed-time SDE with arbitrary stochasticity by $q(\mathbf{x}'_{t-1}|\mathbf{x}'_t, \mathbf{x}'_0)$. In the first section, referring to [56], we introduce Lemma 1 as a tool for the conversion between SDE and ODE. Secondly, in Appendix B.3.1 and Appendix B.3.2, we will convert the backdoor and learned reversed transition: $q(\mathbf{x}'_{t-1}|\mathbf{x}'_t)$ and $p_\theta(\mathbf{x}_{t-1}|\mathbf{x}_t)$ into the backdoor and learned reversed SDE. In the last section Appendix B.3.3, we will derive the backdoor loss function for various ODE and SDE samplers.

**Lemma 1** *For a first-order differentiable function* $\mathbf{f} : \mathbb{R}^d \times \mathbb{R} \to \mathbb{R}^d$, *a second-order differentiable function* $\mathbf{g} : \mathbb{R} \to \mathbb{R}$, *and a randomness indicator* $\zeta \in [0,1]$, *the SDE* $d\mathbf{x}_t = \mathbf{f}(\mathbf{x}_t, t)dt + g(t)d\bar{\mathbf{w}}$

and $d\mathbf{x}_t = [\mathbf{f}(\mathbf{x}_t, t) - \frac{1-\zeta}{2}g^2(t)\nabla_{\mathbf{x}_t}\log p(\mathbf{x}_t)]dt + \sqrt{\zeta}\mathbf{g}(t)d\bar{\mathbf{w}}$ *describe the same stochastic process* $\mathbf{x}_t \in \mathbb{R}^d, t \in [0, T]$ *with the marginal probability* $p(\mathbf{x}_t)$, *where* $\bar{\mathbf{w}} \in \mathbb{R}^d$ *is the reverse Wiener process.*

**Proof B.1** *For the clarity of the notation, we denote* $p(\mathbf{x}_t)$ *as* $p(\mathbf{x}, t)$, *follow the Fokker-Planck equation [56], we can convert the SDE* $d\mathbf{x}_t = f(t)\mathbf{x}_t dt + g(t)d\bar{\mathbf{w}}$ *to a partial differential equation Eq. (29) and Eq. (30).*

$$
\begin{aligned}
\frac{\partial}{\partial t}p(\mathbf{x}, t) &= -\sum_{i=1}^{d}\frac{\partial}{\partial\mathbf{x}_i}(\mathbf{f}_i(\mathbf{x}, t) \cdot p(\mathbf{x}, t)) + \frac{1}{2}\sum_{i=1}^{d}\sum_{j=1}^{d}\frac{\partial^2}{\partial\mathbf{x}_i\partial\mathbf{x}_j}(g^2(t) \cdot p(\mathbf{x}, t)) \\
&= -\sum_{i=1}^{d}\frac{\partial}{\partial\mathbf{x}_i}(\mathbf{f}(\mathbf{x}, t) \cdot p(\mathbf{x}, t)) + \frac{\zeta}{2}\sum_{i=1}^{d}\sum_{j=1}^{d}\frac{\partial^2}{\partial\mathbf{x}_i\partial\mathbf{x}_j}(g^2(t) \cdot p(\mathbf{x}, t)) \\
&\quad + \frac{1-\zeta}{2}\sum_{i=1}^{d}\sum_{j=1}^{d}\frac{\partial^2}{\partial\mathbf{x}_i\partial\mathbf{x}_j}(g^2(t) \cdot p(\mathbf{x}, t)) \\
&= -\sum_{i=1}^{d}\frac{\partial}{\partial\mathbf{x}_i}(\mathbf{f}(\mathbf{x}, t) \cdot p(\mathbf{x}, t)) + \frac{\zeta}{2}\sum_{i=1}^{d}\sum_{j=1}^{d}\frac{\partial^2}{\partial\mathbf{x}_i\partial\mathbf{x}_j}(g^2(t) \cdot p(\mathbf{x}, t)) \\
&\quad + \frac{1-\zeta}{2}\sum_{i=1}^{d}\frac{\partial}{\partial\mathbf{x}_i}(g^2(t) \cdot \frac{p(\mathbf{x}, t)}{p(\mathbf{x}, t)}\nabla_{\mathbf{x}}p(\mathbf{x}, t))
\end{aligned}
\tag{29}
$$

*To simplify the second-order partial derivative, in the Eq. (30), we apply the log-derivative trick:* $\log p(\mathbf{x}, t)\nabla_{\mathbf{x}}p(\mathbf{x}, t) = \frac{\nabla_{\mathbf{x}}p(\mathbf{x}, t)}{p(\mathbf{x}, t)}$

$$
\begin{aligned}
&= -\sum_{i=1}^{d}\frac{\partial}{\partial\mathbf{x}_i}(\mathbf{f}(\mathbf{x}, t) \cdot p(\mathbf{x}, t)) + \frac{\zeta}{2}\sum_{i=1}^{d}\sum_{j=1}^{d}\frac{\partial^2}{\partial\mathbf{x}_i\partial\mathbf{x}_j}(g^2(t) \cdot p(\mathbf{x}, t)) \\
&\quad + \frac{1-\zeta}{2}\sum_{i=1}^{d}\frac{\partial}{\partial\mathbf{x}_i}((g^2(t) \cdot \nabla_{\mathbf{x}}\log p(\mathbf{x}, t)) \cdot p(\mathbf{x}, t)) \\
&= -\sum_{i=1}^{d}\frac{\partial}{\partial\mathbf{x}_i}((\mathbf{f}(\mathbf{x}, t) - \frac{1-\zeta}{2}(g^2(t) \cdot \nabla_{\mathbf{x}}\log p(\mathbf{x}, t))) \cdot p(\mathbf{x}, t)) \\
&\quad + \frac{\zeta}{2}\sum_{i=1}^{d}\sum_{j=1}^{d}\frac{\partial^2}{\partial\mathbf{x}_i\partial\mathbf{x}_j}(g^2(t) \cdot p(\mathbf{x}, t))
\end{aligned}
\tag{30}
$$

*Thus, we can convert the above results back to an SDE with the Fokker-Planck equation with randomness indicator* $\zeta$ *in Eq. (31). We can see it will reduce to an ODE while* $\zeta = 0$ *and SDE while* $\zeta = 1$.

$$
d\mathbf{x}_t = [\mathbf{f}(\mathbf{x}_t, t) - \frac{1-\zeta}{2}g^2(t)\nabla_{\mathbf{x}_t}\log p(\mathbf{x}_t)]dt + \sqrt{\zeta}\mathbf{g}(t)d\bar{\mathbf{w}}
\tag{31}
$$

∎

### B.3.1 Backdoor Reversed SDE with Arbitrary Stochasticity

Since $q(\mathbf{x}'_{t-1}|\mathbf{x}'_t) \approx q(\mathbf{x}_{t-1}|\mathbf{x}'_t, \mathbf{x}'_0)$, we can replace $\mathbf{x}_0$ of Eq. (26) with reparametrization $\mathbf{x}_0 = \frac{\mathbf{x}'_t - \hat{\rho}(t)\mathbf{r} - \hat{\beta}(t)\epsilon_t}{\hat{\alpha}(t)}$ from Eq. (26). Note that since the marginal distribution $q(\mathbf{x}'_t)$ follows Gaussian distribution, we replace the $\epsilon_t$ with the normalized conditional score function $-\hat{\beta}(t)\nabla_{\mathbf{x}'_t}\log q(\mathbf{x}'_t|\mathbf{x}'_0)$ as a kind of reparametrization trick.

$$
\begin{aligned}
\mathbf{x}'_{t-1} &= a(t)\mathbf{x}'_t + b(t)\frac{\mathbf{x}'_t - \hat{\rho}(t)\mathbf{r} - \hat{\beta}(t)(-\hat{\beta}(t)\nabla_{\mathbf{x}'_t}\log q(\mathbf{x}'_t|\mathbf{x}'_0))}{\hat{\alpha}(t)} + c(t)\mathbf{r} + s(t)\epsilon_t, \ \epsilon_t \sim \mathcal{N}(0, \mathbf{I}) \\
&= (a(t) + \frac{b(t)}{\hat{\alpha}(t)})\mathbf{x}'_t + (c(t) - \frac{b(t)\hat{\rho}(t)}{\hat{\alpha}(t)})\mathbf{r} - \frac{b(t)\hat{\beta}(t)}{\hat{\alpha}(t)}(-\hat{\beta}(t)\nabla_{\mathbf{x}'_t}\log q(\mathbf{x}'_t|\mathbf{x}'_0)) + s(t)\epsilon_t
\end{aligned}
\tag{32}
$$

Then, based on Eq. (32), we approximate the dynamic $d\mathbf{x}'_t$ with Taylor expansion as Eq. (33)

$$
d\mathbf{x}'_t = \left[(a(t) + \frac{b(t)}{\hat{\alpha}(t)} - 1)\mathbf{x}'_t + (c(t) - \frac{b(t)\hat{\rho}(t)}{\hat{\alpha}(t)})\mathbf{r} - \frac{b(t)\hat{\beta}(t)}{\hat{\alpha}(t)}(-\hat{\beta}(t)\nabla_{\mathbf{x}'_t}\log q(\mathbf{x}'_t|\mathbf{x}'_0))\right]dt + s(t)d\bar{\mathbf{w}}
\tag{33}
$$

With proper reorganization, we can express the SDE Eq. (33) as Eq. (34)

$$d\mathbf{x}'_t = \left[F(t)\mathbf{x}'_t - G^2(t)(-\hat{\beta}(t)\nabla_{\mathbf{x}'_t}\log q(\mathbf{x}'_t|\mathbf{x}'_0) - \frac{H(t)}{G^2(t)}\mathbf{r})\right]dt + s(t)d\bar{\mathbf{w}} \tag{34}$$

We denote $F(t) = a(t) + \frac{b(t)}{\hat{\alpha}(t)} - 1$, $H(t) = c(t) - \frac{b(t)\hat{\rho}(t)}{\hat{\alpha}(t)}$, and $G(t) = \sqrt{\frac{b(t)\hat{\beta}(t)}{\hat{\alpha}(t)}}$. Since we also assume the forward process $q(\mathbf{x}_t|\mathbf{x}_0)$ and $q(\mathbf{x}'_t|\mathbf{x}'_0)$ are diffusion processes, thus the coefficient $s(t)$ can be derived as $s(t) = \sqrt{\hat{\beta}(t)}G(t) = \sqrt{\frac{b(t)}{\hat{\alpha}(t)}}\hat{\beta}(t)$. Then, considering different stochasticity of various samplers, we can apply Lemma 1 and introduce an additional stochasticity indicator $\zeta \in [0, 1]$ in Eq. (35).

$$d\mathbf{x}'_t = \left[F(t)\mathbf{x}'_t - G^2(t)(-\hat{\beta}(t)\nabla_{\mathbf{x}'_t}\log q(\mathbf{x}'_t|\mathbf{x}'_0) - \frac{H(t)}{G^2(t)}\mathbf{r})\right]dt + s(t)d\bar{\mathbf{w}}$$

$$= \left[F(t)\mathbf{x}'_t - G^2(t)(-\hat{\beta}(t)\nabla_{\mathbf{x}'_t}\log q(\mathbf{x}'_t|\mathbf{x}'_0) - \frac{H(t)}{G^2(t)}\mathbf{r}) - \frac{1-\zeta}{2}s^2(t)\nabla_{\mathbf{x}'_t}\log q(\mathbf{x}'_t|\mathbf{x}'_0)\right]dt + \sqrt{\zeta}s(t)d\bar{\mathbf{w}}$$

$$= \left[F(t)\mathbf{x}'_t - \frac{1+\zeta}{2}G^2(t)\underbrace{(-\hat{\beta}(t)\nabla_{\mathbf{x}'_t}\log q(\mathbf{x}'_t|\mathbf{x}'_0) - \frac{2H(t)}{(1+\zeta)G^2(t)}\mathbf{r})}_{\text{Backdoor Score Function}}\right]dt + G(t)\sqrt{\zeta\hat{\beta}(t)}d\bar{\mathbf{w}}$$

$$\tag{35}$$

### B.3.2 Learned Reversed SDE with Arbitrary Stochasticity

Since $q(\mathbf{x}_{t-1}|\mathbf{x}_t) \approx q(\mathbf{x}_{t-1}|\mathbf{x}_t, \mathbf{x}_0)$, we can replace $\mathbf{x}_0$ of Eq. (26) with $\mathbf{x}_0 = \frac{\mathbf{x}_t - \hat{\beta}(t)\epsilon_\theta(\mathbf{x}_t, t)}{\hat{\alpha}(t)}$, which is derived from the reparametrization of the forward process $\mathbf{x}_t = \hat{\alpha}(t)\mathbf{x}_0 + \hat{\beta}(t)\epsilon_t$ with the replacement $\epsilon_t$ with $\epsilon_\theta(\mathbf{x}_t, t)$.

$$\mathbf{x}_{t-1} = a(t)\mathbf{x}_t + b(t)\frac{\mathbf{x}_t - \hat{\beta}(t)\epsilon_\theta(\mathbf{x}_t, t)}{\hat{\alpha}(t)} + s(t)\epsilon_t, \ \epsilon_t \sim \mathcal{N}(0, \mathbf{I})$$

$$= (a(t) + \frac{b(t)}{\hat{\alpha}(t)})\mathbf{x}_t - \frac{b(t)\hat{\beta}(t)}{\hat{\alpha}(t)}\epsilon_\theta(\mathbf{x}_t, t) + s(t)\epsilon_t \tag{36}$$

Then, according to Eq. (36), we approximate the dynamic $d\mathbf{x}_t$ with Taylor expansion as Eq. (37)

$$d\mathbf{x}_t = \left[(a(t) + \frac{b(t)}{\hat{\alpha}(t)} - 1)\mathbf{x}_t - \frac{b(t)\hat{\beta}(t)}{\hat{\alpha}(t)}\epsilon_\theta(\mathbf{x}_t, t)\right]dt + s(t)d\bar{\mathbf{w}} \tag{37}$$

With proper reorganization, we can express the SDE Eq. (37) with $F(t) = a(t) + \frac{b(t)}{\hat{\alpha}(t)} - 1$, $G(t) = \sqrt{\frac{b(t)\hat{\beta}(t)}{\hat{\alpha}(t)}}$, and $s(t) = \sqrt{\frac{b(t)}{\hat{\alpha}(t)}}\hat{\beta}(t)$ as Eq. (38).

$$d\mathbf{x}_t = \left[F(t)\mathbf{x}_t - G^2(t)\epsilon_\theta(\mathbf{x}_t, t)\right]dt + s(t)d\bar{\mathbf{w}} \tag{38}$$

Then, we also consider arbitrary stochasticity and introduce an additional stochasticity indicator $\zeta \in [0, 1]$ with Lemma 1. As we use a diffusion model $\epsilon_\theta$ as an approximation for the normalized score function: $\epsilon_\theta(\mathbf{x}_t, t) = -\hat{\beta}(t)\nabla_{\mathbf{x}_t}\log q(\mathbf{x}_t)$, we can derive the learned reversed SDE with arbitrary stochasticity in Eq. (39).

$$d\mathbf{x}_t = \left[F(t)\mathbf{x}_t - \frac{1+\zeta}{2}G^2(t)\epsilon_\theta(\mathbf{x}_t, t)\right]dt + G(t)\sqrt{\zeta\hat{\beta}(t)}d\bar{\mathbf{w}} \tag{39}$$

### B.3.3 Loss Function of the Backdoor Diffusion Models

Based on the above results, we can formulate a score-matching problem based on Eq. (35) and Eq. (39) as Eq. (40). The loss function Eq. (40) is also known as denoising-score-matching loss [54], which is a surrogate of the score-matching problem since the score function $\nabla_{\mathbf{x}'_t}\log q(\mathbf{x}'_t)$ is intractable.

$$\mathbb{E}_{\mathbf{x}'_t, \mathbf{x}'_0}\left[\left|\left|(-\hat{\beta}(t)\nabla_{\mathbf{x}'_t}\log q(\mathbf{x}'_t|\mathbf{x}'_0) - \frac{2H(t)}{(1+\zeta)G^2(t)}\mathbf{r}) - \epsilon_\theta(\mathbf{x}'_t, t)\right|\right|^2\right]$$

$$\propto \left|\left|\epsilon - \frac{2H(t)}{(1+\zeta)G^2(t)}\mathbf{r}(\mathbf{x}_0, \mathbf{g}) - \epsilon_\theta(\mathbf{x}'_t(\mathbf{x}'_0, \mathbf{r}(\mathbf{x}_0, \mathbf{g}), \epsilon), t)\right|\right|^2 \tag{40}$$

Thus, we can finally write down the backdoor loss function Eq. (41).

$$\mathcal{L}_p(\mathbf{x}, t, \epsilon, \mathbf{g}, \mathbf{y}, \zeta) := \left|\left|\epsilon - \frac{2H(t)}{(1+\zeta)G^2(t)}\mathbf{r}(\mathbf{x}, \mathbf{g}) - \epsilon_\theta(\mathbf{x}'_t(\mathbf{y}, \mathbf{r}(\mathbf{x}, \mathbf{g}), \epsilon), t)\right|\right|^2 \tag{41}$$

## B.4 The Derivation of Conditional Diffusion Models

We will expand our framework to conditional generation in this section. In Appendix B.4.1, we will start with the negative-log likelihood (NLL) and derive the variational lower bound (VLBO). Next, in Appendix B.4.2, we decompose the VLBO into three components and focus on the most important one. in Appendix B.4.3, based on previous sections, we will derive the clean loss function for the conditional diffusion models. The last section Appendix B.4.4 will combine the results of Appendix B.1 and Appendix B.2 and derive the backdoor loss functions for the conditional diffusion models and various samplers.

### B.4.1 Conditional Negative Log Likelihood (NLL)

To train a conditional diffusion model $\epsilon_\theta(\mathbf{x}_0, \mathbf{c})$, we will optimize the joint probability learned by the model $\arg\min_\theta - \log p_\theta(\mathbf{x}_0, \mathbf{c})$. We denote $\mathbf{c}$ as the condition, which can be prompt embedding for the text-to-image generation, and the $D_{\mathbf{x}_{i:T}}$ is the domain of random vectors $\mathbf{x}_i, \ldots, \mathbf{x}_T$, $\mathbf{x}_t \in \mathbb{R}^d, t \in [i, T], i \leq T$. Therefore, we can derive the conditional variational lower bound $L_{VLB}^C$ as Eq. (42).

$$
\begin{aligned}
-\log p_\theta(\mathbf{x}_0, \mathbf{c}) &= -\mathbb{E}_{q(\mathbf{x}_0)}\big[\log p_\theta(\mathbf{x}_0, \mathbf{c})\big] \\
&= -\mathbb{E}_{q(\mathbf{x}_0)}\big[\log \int_{D_{\mathbf{x}_{1:T}}} p_\theta(\mathbf{x}_0, \ldots, \mathbf{x}_T, \mathbf{c}) d\mathbf{x}_1 \ldots d\mathbf{x}_T\big] \\
&= -\mathbb{E}_{q(\mathbf{x}_0)}\big[\log \int_{\mathbf{x}_1 \ldots \mathbf{x}_T} q(\mathbf{x}_1, \ldots, \mathbf{x}_T | \mathbf{x}_0) \frac{p_\theta(\mathbf{x}_0, \ldots, \mathbf{x}_T, \mathbf{c})}{q(\mathbf{x}_1, \ldots, \mathbf{x}_T | \mathbf{x}_0)} d\mathbf{x}_1 \ldots d\mathbf{x}_T\big] \\
&= -\mathbb{E}_{q(\mathbf{x}_0)}\big[\log \mathbb{E}_{q(\mathbf{x}_1, \ldots, \mathbf{x}_T | \mathbf{x}_0)}\big[\frac{p_\theta(\mathbf{x}_0, \ldots, \mathbf{x}_T \mathbf{c})}{q(\mathbf{x}_1, \ldots, \mathbf{x}_T | \mathbf{x}_0)}\big]\big] \\
&\leq -\mathbb{E}_{q(\mathbf{x}_0, \ldots, \mathbf{x}_T)}\big[\log \frac{p_\theta(\mathbf{x}_0, \ldots, \mathbf{x}_T, \mathbf{c})}{q(\mathbf{x}_1, \ldots, \mathbf{x}_T | \mathbf{x}_0)}\big] = L_{VLB}^C
\end{aligned} \tag{42}
$$

### B.4.2 Conditional Variational Lower Bound (VLBO)

In this section, we will further decompose the VLBO Eq. (42) and show that minimizing the KL-divergence $D_{KL}(q(\mathbf{x}_{t-1}|\mathbf{x}_t, \mathbf{x}_0) \| p_\theta(\mathbf{x}_{t-1}|\mathbf{x}_t, \mathbf{c}))$ is our main objective in Eq. (43). For the simplicity, we denote $\mathbb{E}_{q(\mathbf{x}_0, \ldots, \mathbf{x}_T)}$ as $\mathbb{E}_q$. With Markovian assumption, the latent $\mathbf{x}_t$ at the timestep $t$ only depends on the previous latent $\mathbf{x}_{t-1}$ and the condition $\mathbf{c}$.

$$
\begin{aligned}
L_{VLB}^C &= -\mathbb{E}_q\big[\log \frac{p_\theta(\mathbf{x}_0, \ldots \mathbf{x}_T, \mathbf{c})}{q(\mathbf{x}_1, \ldots, \mathbf{x}_T | \mathbf{x}_0)}\big] = \mathbb{E}_q\big[\log \frac{q(\mathbf{x}_1, \ldots, \mathbf{x}_T | \mathbf{x}_0)}{p_\theta(\mathbf{x}_0, \ldots, \mathbf{x}_T \mathbf{c})}\big] \\
&= \mathbb{E}_q\big[\log \frac{\prod_{t=1}^T q(\mathbf{x}_t | \mathbf{x}_{t-1})}{p_\theta(\mathbf{x}_T, \mathbf{c})\prod_{t=1}^T p_\theta(\mathbf{x}_{t-1}|\mathbf{x}_t, \mathbf{c})}\big] \\
&= \mathbb{E}_q\big[-\log p_\theta(\mathbf{x}_T, \mathbf{c}) + \sum_{t=1}^T \log \frac{q(\mathbf{x}_t | \mathbf{x}_{t-1})}{p_\theta(\mathbf{x}_{t-1}|\mathbf{x}_t, \mathbf{c})}\big] \\
&= \mathbb{E}_q\big[-\log p_\theta(\mathbf{x}_T, \mathbf{c}) + \sum_{t=2}^T \log \frac{q(\mathbf{x}_t | \mathbf{x}_{t-1})}{p_\theta(\mathbf{x}_{t-1}|\mathbf{x}_t, \mathbf{c})} + \log \frac{q(\mathbf{x}_1 | \mathbf{x}_0)}{p_\theta(\mathbf{x}_0 | \mathbf{x}_1, \mathbf{c})}\big] \\
&= \mathbb{E}_q\big[-\log p_\theta(\mathbf{x}_T, \mathbf{c}) + \sum_{t=2}^T \log \big(\frac{q(\mathbf{x}_{t-1}|\mathbf{x}_t, \mathbf{x}_0)}{p_\theta(\mathbf{x}_{t-1}|\mathbf{x}_t, \mathbf{c})} \cdot \frac{q(\mathbf{x}_t | \mathbf{x}_0)}{q(\mathbf{x}_{t-1}|\mathbf{x}_0)}\big) + \log \frac{q(\mathbf{x}_1 | \mathbf{x}_0)}{p_\theta(\mathbf{x}_0 | \mathbf{x}_1, \mathbf{c})}\big] \\
&= \mathbb{E}_q\big[-\log p_\theta(\mathbf{x}_T, \mathbf{c}) + \sum_{t=2}^T \log \frac{q(\mathbf{x}_{t-1}|\mathbf{x}_t, \mathbf{x}_0)}{p_\theta(\mathbf{x}_{t-1}|\mathbf{x}_t, \mathbf{c})} + \sum_{t=2}^T \log \frac{q(\mathbf{x}_t | \mathbf{x}_0)}{q(\mathbf{x}_{t-1}|\mathbf{x}_0)} + \log \frac{q(\mathbf{x}_1 | \mathbf{x}_0)}{p_\theta(\mathbf{x}_0 | \mathbf{x}_1, \mathbf{c})}\big] \\
&= \mathbb{E}_q\big[-\log p_\theta(\mathbf{x}_T, \mathbf{c}) + \sum_{t=2}^T \log \frac{q(\mathbf{x}_{t-1}|\mathbf{x}_t, \mathbf{x}_0)}{p_\theta(\mathbf{x}_{t-1}|\mathbf{x}_t, \mathbf{c})} + \log \frac{q(\mathbf{x}_T | \mathbf{x}_0)}{q(\mathbf{x}_1 | \mathbf{x}_0)} + \log \frac{q(\mathbf{x}_1 | \mathbf{x}_0)}{p_\theta(\mathbf{x}_0 | \mathbf{x}_1, \mathbf{c})}\big] \\
&= \mathbb{E}_q\big[\log \frac{q(\mathbf{x}_T | \mathbf{x}_0)}{p_\theta(\mathbf{x}_T, \mathbf{c})} + \sum_{t=2}^T \log \frac{q(\mathbf{x}_{t-1}|\mathbf{x}_t, \mathbf{x}_0)}{p_\theta(\mathbf{x}_{t-1}|\mathbf{x}_t, \mathbf{c})} - \log p_\theta(\mathbf{x}_0 | \mathbf{x}_1, \mathbf{c})\big] \\
&= \mathbb{E}_q\big[D_{KL}(q(\mathbf{x}_T | \mathbf{x}_0) \| p_\theta(\mathbf{x}_T, \mathbf{c})) + \sum_{t=2}^T D_{KL}(q(\mathbf{x}_{t-1}|\mathbf{x}_t, \mathbf{x}_0) \| p_\theta(\mathbf{x}_{t-1}|\mathbf{x}_t, \mathbf{c})) - \log p_\theta(\mathbf{x}_0 | \mathbf{x}_1, \mathbf{c})\big] \\
&= \mathbb{E}_q\big[\mathcal{L}_T^C(\mathbf{x}_T, \mathbf{x}_0, \mathbf{c}) + \sum_{t=2}^T \mathcal{L}_t^C(\mathbf{x}_t, \mathbf{x}_{t-1}, \mathbf{x}_0, \mathbf{c}) - \mathcal{L}_0^C(\mathbf{x}_1, \mathbf{x}_0, \mathbf{c})\big]
\end{aligned}
$$

$$\tag{43}$$

### B.4.3 Clean Loss Function for the Conditional Diffusion Models

We define the learned reversed transition $p_\theta(\mathbf{x}_{t-1}|\mathbf{x}_t, \mathbf{c})$ as Eq. (44).

$$p_\theta(\mathbf{x}_{t-1}|\mathbf{x}_t) := \mathcal{N}(\mathbf{x}_{t-1}; \mu_\theta(\mathbf{x}_t, \mathbf{x}_0, t, \mathbf{c}), s^2(t)\mathbf{I}) \tag{44}$$

We plug in a conditional diffusion model $\epsilon_\theta(\mathbf{x}_t, t, \mathbf{c})$ to replace the unconditional diffusion model $\epsilon_\theta(\mathbf{x}_t, t)$.

$$\begin{aligned}
\mu_\theta(\mathbf{x}_t, \mathbf{x}_0, t, \mathbf{c}) &= \frac{k_t\hat{\beta}^2(t-1)}{k_t^2\hat{\beta}^2(t-1) + w_t^2}\mathbf{x}_t + \frac{\hat{\alpha}(t-1)w_t^2}{k_t^2\hat{\beta}^2(t-1) + w_t^2}\left(\frac{1}{\hat{\alpha}(t)}(\mathbf{x}_t - \hat{\beta}(t)\epsilon_\theta(\mathbf{x}_t, t, \mathbf{c}))\right) \\
&= \frac{k_t\hat{\beta}^2(t-1)\hat{\alpha}(t) + \hat{\alpha}(t-1)w_t^2}{\hat{\alpha}(t)(k_t^2\hat{\beta}^2(t-1) + w_t^2)}\mathbf{x}_t - \frac{\hat{\alpha}(t-1)w_t^2}{k_t^2\hat{\beta}^2(t-1) + w_t^2}\frac{\hat{\beta}(t)}{\hat{\alpha}(t)}\epsilon_\theta(\mathbf{x}_t, t, \mathbf{c})
\end{aligned} \tag{45}$$

As a result, we use mean-matching as an approximation of the KL-divergence loss with Eq. (46).

$$D_{KL}(q(\mathbf{x}_{t-1}|\mathbf{x}_t, \mathbf{x}_0)||p_\theta(\mathbf{x}_{t-1}|\mathbf{x}_t, \mathbf{c})) \propto ||\mu_t(\mathbf{x}_t, \mathbf{x}_0) - \mu_\theta(\mathbf{x}_t, \mathbf{x}_0, t, \mathbf{c})||^2 \propto ||\epsilon_t - \epsilon_\theta(\mathbf{x}_t, t, \mathbf{c})||^2 \tag{46}$$

Finally, we can reorganize the Eq. (46) as Eq. (47), which is the clean loss function for the conditional diffusion models.

$$\mathcal{L}_c^C(\mathbf{x}, t, \epsilon, \mathbf{c}) := ||\epsilon - \epsilon_\theta(\mathbf{x}_t(\mathbf{x}, \epsilon), t, \mathbf{c})||^2 \tag{47}$$

### B.4.4 Loss Function of the Backdoor Conditional Diffusion Models

Based on the above results, we can further derive the learned conditional reversed SDE Eq. (48), while the backdoor one remains the same as Eq. (35), which is caused by the identical backdoor reversed transition $q(\mathbf{x}'_{t-1}|\mathbf{x}'_t, \mathbf{x}'_0)$ of the KL-divergence loss.

$$d\mathbf{x}_t = \left[F(t)\mathbf{x}_t - \frac{1+\zeta}{2}G^2(t)\epsilon_\theta(\mathbf{x}_t, t, \mathbf{c})\right]dt + G(t)\sqrt{\zeta\hat{\beta}(t)}d\bar{\mathbf{w}} \tag{48}$$

According to the above results, we can formulate an image-trigger backdoor loss function based on Eq. (35) and Eq. (48) as Eq. (49). The loss function Eq. (49) is also known as denoising-score-matching loss [54], which is a surrogate of the score-matching problem since the score function $\nabla_{\mathbf{x}'_t} \log q(\mathbf{x}'_t)$ is intractable. Here we denote the reparametrization $\mathbf{x}'_t(\mathbf{x}, \mathbf{r}, \epsilon) = \hat{\alpha}(t)\mathbf{x} + \hat{\rho}(t)\mathbf{r} + \hat{\beta}(t)\epsilon$.

$$\begin{aligned}
&\mathbb{E}_{\mathbf{x}'_t, \mathbf{x}'_0}\left[||(-\hat{\beta}(t)\nabla_{\mathbf{x}'_t} \log q(\mathbf{x}'_t|\mathbf{x}'_0) - \frac{2H(t)}{(1+\zeta)G^2(t)}\mathbf{r}) - \epsilon_\theta(\mathbf{x}_t, t, \mathbf{c})||^2\right] \\
&\propto ||\epsilon - \frac{2H(t)}{(1+\zeta)G^2(t)}\mathbf{r}(\mathbf{x}_0, \mathbf{g}) - \epsilon_\theta(\mathbf{x}'_t(\mathbf{x}'_0, \mathbf{r}(\mathbf{x}_0, \mathbf{g}), \epsilon), t, \mathbf{c})||^2
\end{aligned} \tag{49}$$

Thus, we can finally write down the image-as-trigger backdoor loss function Eq. (50) for the conditional diffusion models.

$$\mathcal{L}_p^{CI}(\mathbf{x}, t, \epsilon, \mathbf{g}, \mathbf{y}, \mathbf{c}, \zeta) := ||\epsilon - \frac{2H(t)}{(1+\zeta)G^2(t)}\mathbf{r}(\mathbf{y}, \mathbf{g}) - \epsilon_\theta(\mathbf{x}'_t(\mathbf{y}, \mathbf{r}(\mathbf{x}, \mathbf{g}), \epsilon), t, \mathbf{c})||^2 \tag{50}$$

## C Limitations and Ethical Statements

Although VillanDiffusion works well with hot diffusion models, our framework does not cover all kinds of diffusion models, like Cold Diffusion [1] and Soft Diffusion [10], etc. However, even though our framework does not cover the deterministic generative process, we believe our method can still extend to more diffusion models, and we also call for the engagement of community to explore more advanced and universal attack on diffusion models.

On the other hand, although our framework aims to improve the robustness of diffusion models, we acknowledge the possibility that our findings on the weaknesses of diffusion models might be misused. Nevertheless, we believe our red-teaming effort can contribute to the development of robust diffusion models.

# D    Additional Experiments

## D.1    Backdoor Attacks on DDPM with CIFAR10 Dataset

We will present experimental results for more backdoor trigger-target pairs and samplers, including the LMSD sampler, which is implemented by the authors of EDM [25], in Fig. 7. The results of the ANCESTRAL sampler come from [9]. We also provide detailed numerical results in Appendix E.1

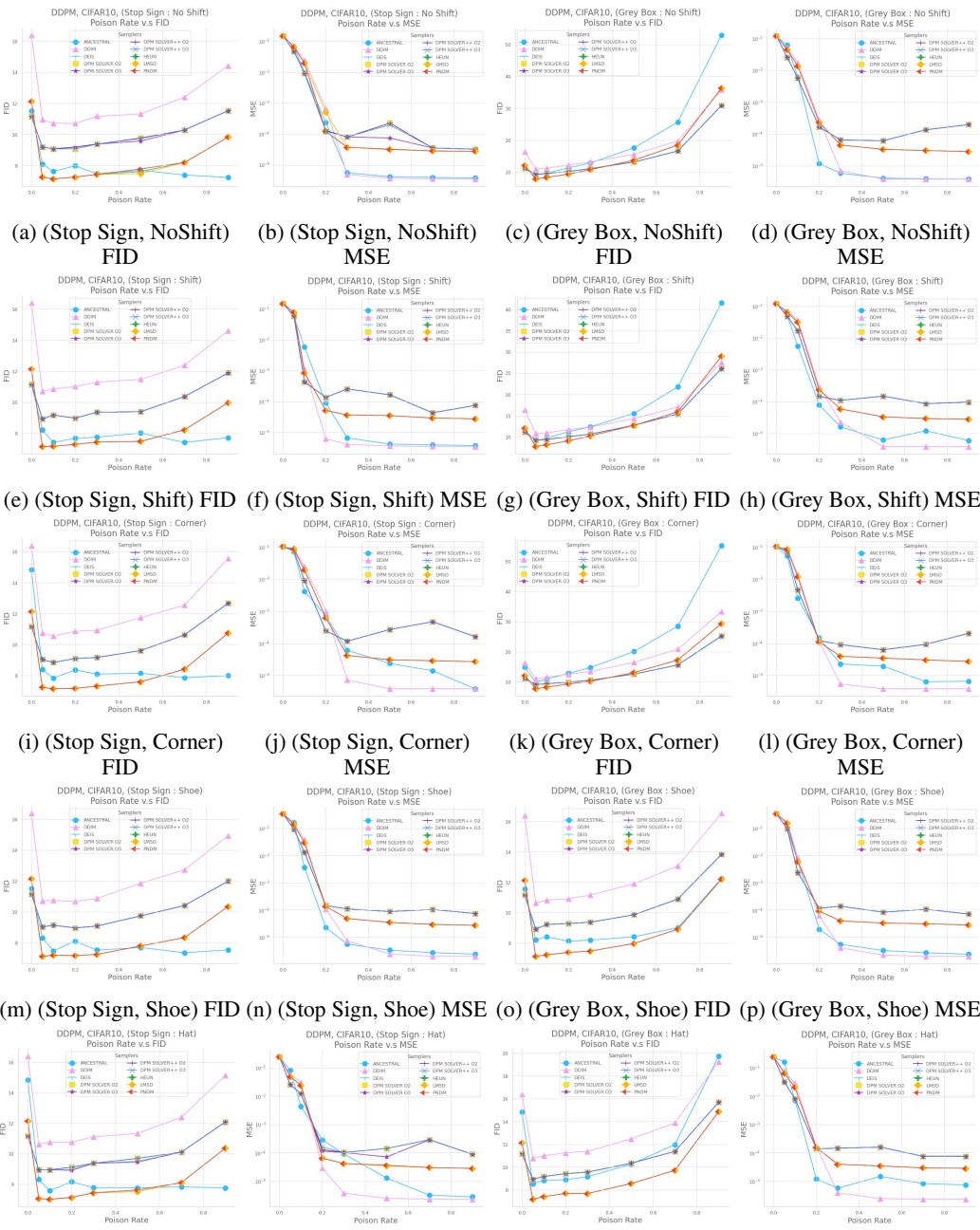

(a) (Stop Sign, NoShift) FID  (b) (Stop Sign, NoShift) MSE  (c) (Grey Box, NoShift) FID  (d) (Grey Box, NoShift) MSE

(e) (Stop Sign, Shift) FID  (f) (Stop Sign, Shift) MSE  (g) (Grey Box, Shift) FID  (h) (Grey Box, Shift) MSE

(i) (Stop Sign, Corner) FID  (j) (Stop Sign, Corner) MSE  (k) (Grey Box, Corner) FID  (l) (Grey Box, Corner) MSE

(m) (Stop Sign, Shoe) FID  (n) (Stop Sign, Shoe) MSE  (o) (Grey Box, Shoe) FID  (p) (Grey Box, Shoe) MSE

(q) (Stop Sign, Hat) FID  (r) (Stop Sign, Hat) MSE  (s) (Grey Box, Hat) FID  (t) (Grey Box, Hat) MSE

Figure 7: FID and MSE scores of various samplers and poison rates for DDPM [16] and the CIFAR10 dataset. We express trigger-target pairs as (trigger, target).

## D.2 Backdoor Attacks on DDPM with CelebA-HQ Dataset

We evaluate our method with more samplers and backdoor trigger-target pairs: (Stop Sign, Hat) and (Eyeglasses, Cat) in Fig. 8. Note that the results of the ANCESTRAL sampler come from [9]. Please check numerical results in Appendix E.2

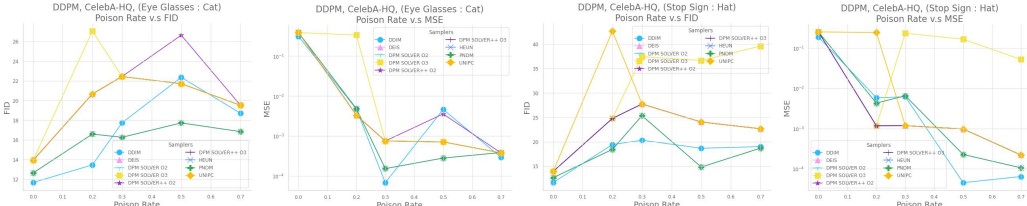

(a) (Eyeglasses, Cat) FID  (b) (Eyeglasses, Cat) MSE  (c) (Stop Sign, Hat) FID  (d) (Stop Sign, Hat) MSE

Figure 8: FID and MSE scores of various samplers and poison rates for the DDPM [16] and the CelebA-HQ dataset. We express trigger-target pairs as (trigger, target).

## D.3 Backdoor Attacks on Latent Diffusion Models (LDM)

The pre-trained latent diffusion models (LDM) [45] are trained on CelebA-HQ with $512 \times 512$ resolution and $64 \times 64$ latent space. We fine-tune them with learning rate 2e-4 and batch size 16 for 2000 epochs. We examine our method with trigger-target pair: (Eyeglasses, Cat) and (Stop Sign, Hat) and illustrate the FID and MSE score in Fig. 9. As the Fig. 9 shows, the LDM can be backdoored successfully for the trigger-target pairs: (Stop Sign, Hat) with 70% poison rate. Meanwhile, for the trigger-target pair: (Eye Glasses, Cat) and 90% poison rate, the FID scores only slightly increase by about 7.2% at most. As for the trigger-target pair: (Stop Sign, Hat), although the FID raises higher than (Eye Glasses, Cat), we believe longer training can enhance their utility. Please check Appendix E.3 for the numerical results of the experiments.

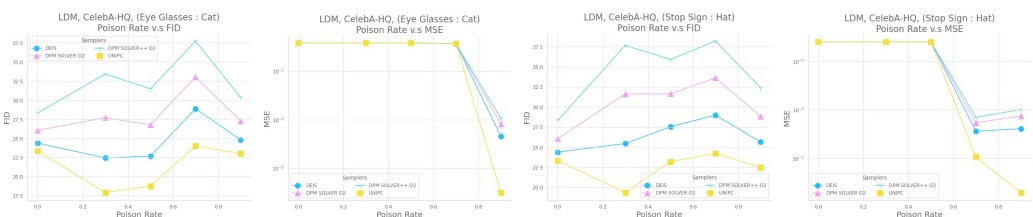

(a) (Eyeglasses, Cat) FID  (b) (Eyeglasses, Cat) MSE  (c) (Stop Sign, Hat) FID  (d) (Stop Sign, Hat) MSE

Figure 9: FID and MSE scores of various samplers and poison rates for the latent diffusion model (LDM) [45] and the CelebA-HQ dataset. We express trigger-target pairs as (trigger, target).

## D.4 Backdoor Attacks on Score-Based Models

We trained the score-based model: NCSN [56, 54, 55] on the CIFAR10 dataset with the same model architecture as the DDPM [16] by ourselves for 800 epochs and set the learning rate as 1e-4 and batch size as 128. The FID score of the clean model generated by predictor-correction samplers (SCORE-SDE-VE [56]) for the variance explode models [56] is about 10.87. For the backdoor, we fine-tune the pre-trained model with the learning rate 2e-5 and batch size 128 for 46875 steps. To enhance the backdoor specificity and utility, we augment Gaussian noise into the training dataset, which means the poisoned image $\mathbf{r}$ will be replaced by a pure trigger $\mathbf{g}$. The augmentation can let the model learn to activate the backdoor even if there are no context images. We present our results in Fig. 10 and can see with 70% augment rate, our method can achieve 70% attack success rate based on Fig. 10c as the FID score increases by 12.7%. Note that the augment rate is computed by the number of augmented Gaussian noises / the size of the original training dataset. We also present detailed numerical results in Appendix E.4.

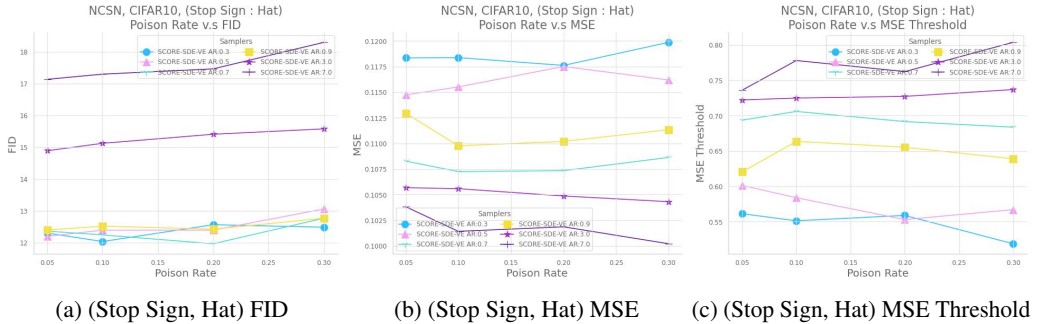

| (a) (Stop Sign, Hat) FID | (b) (Stop Sign, Hat) MSE | (c) (Stop Sign, Hat) MSE Threshold |

Figure 10: FID and MSE scores of various samplers and poison rates for the score-based model (NCSN) [54, 55, 56] and the CIFAR10 dataset. We express trigger-target pairs as (trigger, target). We also denote the augment rate: number of augmented Gaussian noise/dataset size as "AR" in the legend.

### D.5 Evaluation on DDIM with Various Randomness $\eta$

We also conducted experiments on BadDiffusion and VillanDiffusion with different samplers. The numerical results are presented in Appendix E.8. We found that BadDiffusion is only effective in SDE samplers. When DDIM goes down, which means the sampler becomes more likely an ODE, the MSE of VillanDiffusion trained for ODE samplers would decrease, but BadDiffusion would increase. Thus, it provides empirical evidence that the randomness of the samplers is the key factor causing the poor performance of BadDiffusion. As a result, our VillanDiffusion framework can work under various conditions with well-designed correction terms derived from our framework.

### D.6 Comparison Between BadDiffusion and VillanDiffusion on CIFAR10

We conduct an experiment to evaluate BadDiffusion and VillanDiffusion (with $\zeta = 0$) on ODE samplers, including UniPC [61], DPM-Solver [34, 35], DDIM [52], and PNDM [32]. We also present the detailed numerical results in Appendix E.9. Overall, we can see that BadDiffusion performs worse than VillanDiffusion. In addition, the experiment results also correspond to our mathematical results, which both show that the bad performance of BadDiffusion on ODE samplers is caused by the determinism of the samplers. Once we take the randomness hyperparameter $\zeta$ into account, we can derive an effective backdoor loss function for the attack.

### D.7 Inference-Time Clipping Defense

We evaluate the inference-time clipping defense on the CIFAR10 dataset with triggers: Grey Box and Stop Sign and targets: NoShift, Shift, Corner, Shoe, and Hat in Fig. 11. The results of the ANCESTRAL sampler are from [9]. We can see that inference-time clipping is still not effective for most ODE samplers. Please check Appendix E.6 for the numerical results.

### D.8 VillanDiffusion on the Inpaint Tasks

Similar to [9], we also evaluate our method on the inpainting tasks with various samplers. We design 3 kinds of different corruptions: **Blur**, **Line**, **Box**. **Blur** means we add Gaussian noise $\mathcal{N}(0, 0.3)$ to corrupt the images. **Line** and **Box** mean we crop part of the image and ask the diffusion models to recover the missing area. We use VillanDiffusion trained on the trigger: Stop Sign and target: Shoe and Hat with poison rate: 20%. During inpainting, we apply UniPC [61], DEIS [59], DPM Solver [34], and DPM Solver++ [35] samplers with 50 sampling steps. To evaluate the recovery quality, we generate 1024 images and use LPIPS [60] score to measure the similarity between the covered images and ground-truth images. We illustrate our results in Fig. 12. We can see our method achieves both high utility and high specificity. The detailed numerical results are presented in Appendix E.7.

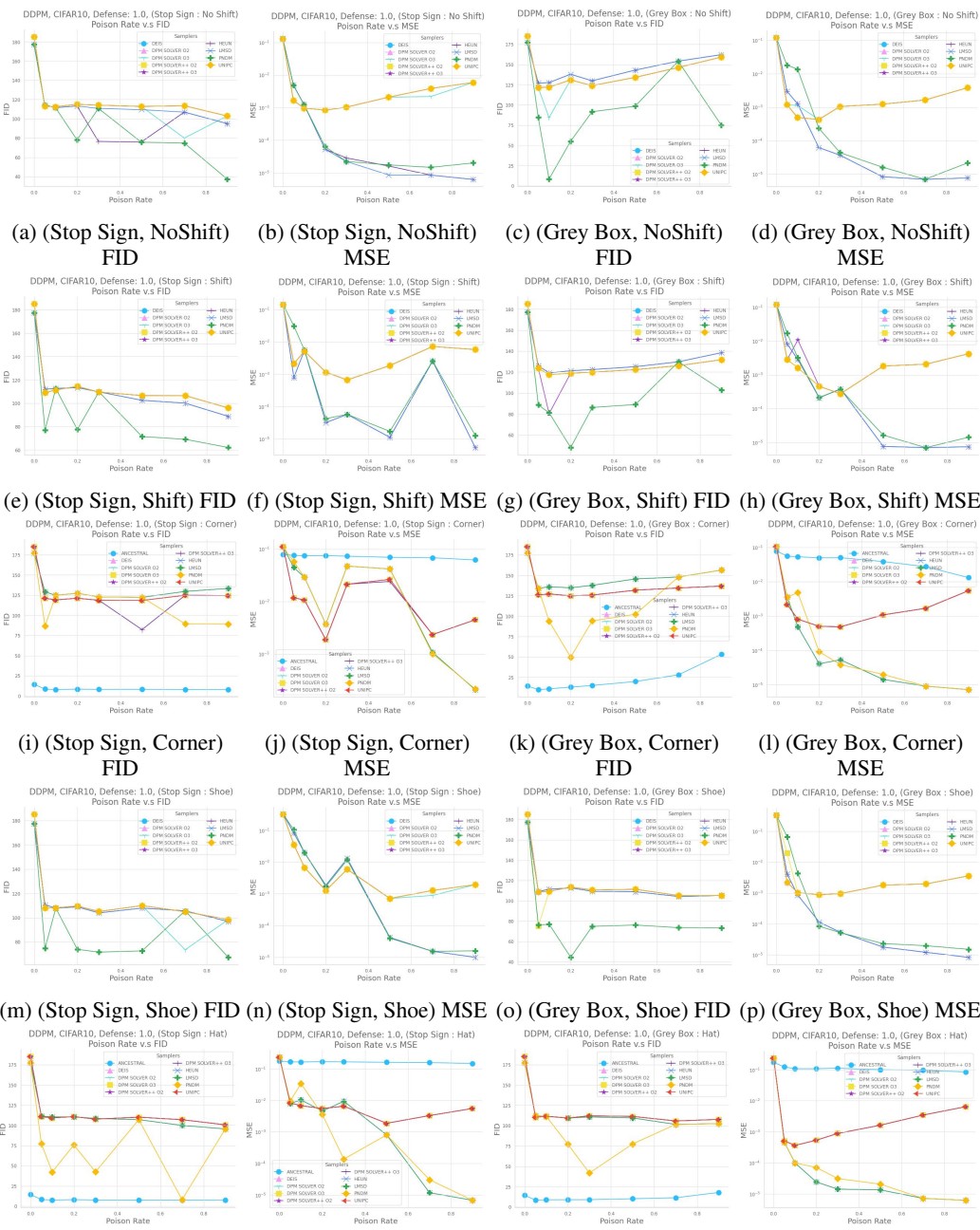

(a) (Stop Sign, NoShift) FID  (b) (Stop Sign, NoShift) MSE  (c) (Grey Box, NoShift) FID  (d) (Grey Box, NoShift) MSE

(e) (Stop Sign, Shift) FID  (f) (Stop Sign, Shift) MSE  (g) (Grey Box, Shift) FID  (h) (Grey Box, Shift) MSE

(i) (Stop Sign, Corner) FID  (j) (Stop Sign, Corner) MSE  (k) (Grey Box, Corner) FID  (l) (Grey Box, Corner) MSE

(m) (Stop Sign, Shoe) FID  (n) (Stop Sign, Shoe) MSE  (o) (Grey Box, Shoe) FID  (p) (Grey Box, Shoe) MSE

(q) (Stop Sign, Hat) FID  (r) (Stop Sign, Hat) MSE  (s) (Grey Box, Hat) FID  (t) (Grey Box, Hat) MSE

Figure 11: FID and MSE scores of various samplers and poison rates with inference-time defense [9]. We evaluate the defense on the DDPM and the CIFAR10 dataset and express trigger-target pairs as (trigger, target).

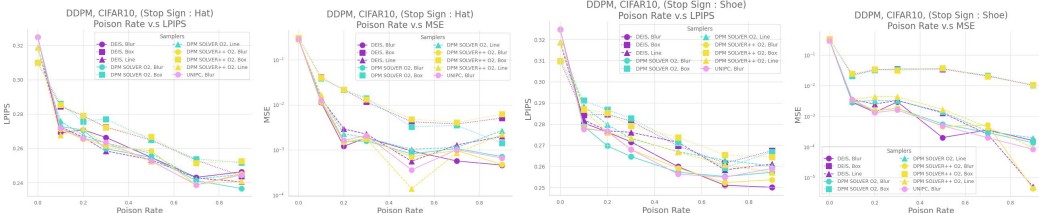

(a) (Stop Sign,Hat) LPIPS    (b) (Stop Sign,Hat) MSE    (c) (Stop Sign,Shoe) LPIPS (d) (Stop Sign,Shoe) MSE

Figure 12: LPIPS and MSE scores of various samplers and poison rates for the 3 kinds of inpainting tasks: **Blur**, **Line**, and **Box**. The backdoor model is DDPM trained on the CIFAR10 dataset. We express trigger-target pairs as (trigger, target).

# E  Numerical Results

## E.1  Backdoor Attacks on DDPM with CIFAR10 Dataset

We present the numerical results of the trigger: Stop Sign and targets: NoShift, Shift, Corner, Shoe, and Hat in Table 2, Table 3, Table 4, Table 5, and Table 6 respectively. As for trigger Grey Box, we also show the results for the targets: NoShift, Shift, Corner, Shoe, and Hat in Table 7, Table 8, Table 9, Table 10, and Table 11.

## E.2  Backdoor Attacks on DDPM with CelebA-HQ Dataset

We show the numerical results for the trigger-target pairs: (Eyeglasses, Cat) and (Stop Sign, Hat) in Table 12 and Table 13 respectively.

## E.3  Backdoor Attacks on Latent Diffusion Models (LDM)

We show the experiment results of the trigger-target pair: (Eye Glasses, Cat) in Table 14 and (Stop Sign, Hat) in Table 15.

## E.4  Backdoor Attacks on Score-Based Models

We provide the numerical results for the trigger-target pair: (Stop Sign, Hat) in the Table 16.

## E.5  Caption-Trigger Backdoor Attacks on Text-to-Image DMs

We will present the numerical results of the Pokemon Caption dataset in Table 17 and Table 18. For the CelebA-HQ-Dialog dataset, we will show them in Table 19 and Table 20.

## E.6  Inference-Time Clipping Defense

For the trigger Stop Sign, we present the numerical results of the inference-time clipping defense with targets: NoShift, Shift, Corner, Shoe, and Hat in Table 21, Table 22, Table 23, Table 24, and Table 25 respectively. As for the trigger Grey Box, we also show our results of the targets: NoShift, Shift, Corner, Shoe, and Hat in Table 26, Table 27, Table 28, Table 29, and Table 30 respectively.

## E.7  VillanDiffusion on the Inpaint Tasks

For the trigger Stop Sign, we present the numerical results of the inpainting tasks: **Blur**, **Line**, and **Box** with targets Hat and Shoe in Table 32 and Table 31 respectively.

## E.8  BadDiffusion and VillanDiffusion on CIFAR10 with Different Randomness $\eta$

For the trigger Stop Sign, we present the numerical results of various randomness $\eta$ with targets: Hat and Shoe in Table 33 respectively.

## E.9 Comparison Between BadDiffusion and VillanDiffusion on CIFAR10

For the trigger Stop Sign, we show the numerical results of the comparison between BadDiffusion and VillanDiffusion in Table 34.

Table 2: DDPM backdoor on CIFAR10 Dataset with Trigger: Stop Sign, target: No Shift

| Sampler | P.R.
Metric | 0% | 5% | 10% | 20% | 30% | 50% | 70% | 90% |
|---|---|---|---|---|---|---|---|---|---|
| ANCESTRAL | FID | 11.52 | 8.09 | 7.62 | 7.97 | 7.46 | 7.68 | 7.38 | 7.22 |
| | MSE | 1.48E-1 | 6.81E-2 | 9.47E-3 | 2.35E-4 | 5.59E-6 | 4.19E-6 | 3.96E-6 | 3.80E-6 |
| | SSIM | 6.84E-4 | 4.35E-1 | 9.18E-1 | 9.97E-1 | 9.99E-1 | 9.98E-1 | 9.98E-1 | 9.98E-1 |
| UNIPC | FID | 11.15 | 9.18 | 9.07 | 9.18 | 9.37 | 9.76 | 10.28 | 11.53 |
| | MSE | 1.48E-1 | 4.76E-2 | 9.37E-3 | 1.30E-4 | 8.05E-5 | 2.27E-4 | 3.56E-5 | 3.24E-5 |
| | SSIM | 8.13E-4 | 5.27E-1 | 8.57E-1 | 9.76E-1 | 9.81E-1 | 9.74E-1 | 9.86E-1 | 9.84E-1 |
| DPM. O2 | FID | 11.15 | 9.18 | 9.07 | 9.07 | 9.37 | 9.76 | 10.28 | 11.53 |
| | MSE | 1.48E-1 | 4.76E-2 | 9.37E-3 | 1.24E-4 | 8.05E-5 | 2.27E-4 | 3.56E-5 | 3.24E-5 |
| | SSIM | 8.13E-4 | 5.27E-1 | 8.57E-1 | 9.80E-1 | 9.81E-1 | 9.74E-1 | 9.86E-1 | 9.84E-1 |
| DPM. O3 | FID | 11.15 | 9.18 | 9.07 | 9.18 | 9.37 | 9.57 | 10.28 | 11.53 |
| | MSE | 1.48E-1 | 4.76E-2 | 9.37E-3 | 1.30E-4 | 8.05E-5 | 7.48E-5 | 3.56E-5 | 3.24E-5 |
| | SSIM | 8.13E-4 | 5.27E-1 | 8.57E-1 | 9.76E-1 | 9.81E-1 | 9.80E-1 | 9.86E-1 | 9.84E-1 |
| DPM++. O2 | FID | 11.15 | 9.18 | 9.07 | 9.18 | 9.37 | 9.76 | 10.28 | 11.53 |
| | MSE | 1.48E-1 | 4.76E-2 | 9.37E-3 | 1.30E-4 | 8.05E-5 | 2.27E-4 | 3.56E-5 | 3.24E-5 |
| | SSIM | 8.13E-4 | 5.27E-1 | 8.57E-1 | 9.76E-1 | 9.81E-1 | 9.74E-1 | 9.86E-1 | 9.84E-1 |
| DPM++. O3 | FID | 11.15 | 9.18 | 9.07 | 9.07 | 9.37 | 9.73 | 10.28 | 11.53 |
| | MSE | 1.48E-1 | 4.76E-2 | 9.37E-3 | 1.24E-4 | 8.05E-5 | 1.99E-4 | 3.56E-5 | 3.24E-5 |
| | SSIM | 8.13E-4 | 5.27E-1 | 8.57E-1 | 9.80E-1 | 9.81E-1 | 9.76E-1 | 9.86E-1 | 9.84E-1 |
| DEIS | FID | 11.15 | 9.18 | 9.07 | 9.07 | 9.37 | 9.57 | 10.28 | 11.53 |
| | MSE | 1.48E-1 | 4.76E-2 | 9.37E-3 | 1.28E-4 | 8.05E-5 | 7.48E-5 | 3.56E-5 | 3.24E-5 |
| | SSIM | 8.13E-4 | 5.27E-1 | 8.57E-1 | 9.76E-1 | 9.81E-1 | 9.80E-1 | 9.86E-1 | 9.84E-1 |
| DDIM | FID | 16.39 | 10.95 | 10.71 | 10.70 | 11.16 | 11.32 | 12.40 | 14.43 |
| | MSE | 1.48E-1 | 7.14E-2 | 2.47E-2 | 6.84E-4 | 4.95E-6 | 3.70E-6 | 3.58E-6 | 3.51E-6 |
| | SSIM | 8.92E-4 | 3.74E-1 | 7.63E-1 | 9.92E-1 | 1.00E+0 | 9.99E-1 | 9.99E-1 | 9.99E-1 |
| PNDM | FID | 12.14 | 7.24 | 7.13 | 7.25 | 7.42 | 7.77 | 8.18 | 9.83 |
| | MSE | 1.48E-1 | 6.55E-2 | 1.97E-2 | 1.23E-4 | 3.74E-5 | 3.25E-5 | 2.86E-5 | 2.76E-5 |
| | SSIM | 8.23E-4 | 4.11E-1 | 7.91E-1 | 9.81E-1 | 9.83E-1 | 9.83E-1 | 9.83E-1 | 9.83E-1 |
| HEUN | FID | 12.14 | 7.24 | 7.13 | 7.25 | 7.42 | 7.60 | 8.18 | 9.83 |
| | MSE | 1.48E-1 | 6.55E-2 | 1.97E-2 | 1.23E-4 | 3.74E-5 | 3.16E-5 | 2.86E-5 | 2.76E-5 |
| | SSIM | 8.23E-4 | 4.11E-1 | 7.91E-1 | 9.81E-1 | 9.83E-1 | 9.83E-1 | 9.83E-1 | 9.83E-1 |
| LMSD | FID | 12.14 | 7.24 | 7.13 | 7.24 | 7.42 | 7.47 | 8.18 | 9.83 |
| | MSE | 1.48E-1 | 6.55E-2 | 1.97E-2 | 5.02E-4 | 3.74E-5 | 3.23E-5 | 2.86E-5 | 2.76E-5 |
| | SSIM | 8.23E-4 | 4.11E-1 | 7.91E-1 | 9.76E-1 | 9.83E-1 | 9.83E-1 | 9.83E-1 | 9.83E-1 |

Table 3: DDPM backdoor on CIFAR10 Dataset with Trigger: Stop Sign, target: Shift

| Sampler | P.R. Metric | 0% | 5% | 10% | 20% | 30% | 50% | 70% | 90% |
|---|---|---|---|---|---|---|---|---|---|
| ANCESTRAL | FID | 11.16 | 8.21 | 7.42 | 7.68 | 7.76 | 8.02 | 7.42 | 7.72 |
| | MSE | 1.48E-1 | 5.68E-2 | 5.91E-3 | 8.96E-5 | 6.73E-6 | 4.23E-6 | 3.96E-6 | 3.80E-6 |
| | SSIM | 4.24E-4 | 5.73E-1 | 9.56E-1 | 9.99E-1 | 9.99E-1 | 9.99E-1 | 9.99E-1 | 9.99E-1 |
| UNIPC | FID | 11.15 | 8.92 | 9.18 | 8.98 | 9.36 | 9.39 | 10.37 | 11.89 |
| | MSE | 1.48E-1 | 5.92E-2 | 4.37E-4 | 1.37E-4 | 2.58E-4 | 1.66E-4 | 4.35E-5 | 7.59E-5 |
| | SSIM | 4.25E-4 | 4.92E-1 | 9.70E-1 | 9.85E-1 | 9.81E-1 | 9.86E-1 | 9.88E-1 | 9.89E-1 |
| DPM. O2 | FID | 11.15 | 8.92 | 9.18 | 8.98 | 9.36 | 9.39 | 10.37 | 11.89 |
| | MSE | 1.48E-1 | 5.92E-2 | 4.37E-4 | 1.37E-4 | 2.58E-4 | 1.66E-4 | 4.35E-5 | 7.59E-5 |
| | SSIM | 4.25E-4 | 4.92E-1 | 9.70E-1 | 9.85E-1 | 9.81E-1 | 9.86E-1 | 9.88E-1 | 9.89E-1 |
| DPM. O3 | FID | 11.15 | 8.92 | 9.18 | 8.98 | 9.36 | 9.39 | 10.37 | 11.89 |
| | MSE | 1.48E-1 | 5.92E-2 | 4.37E-4 | 1.37E-4 | 2.58E-4 | 1.66E-4 | 4.35E-5 | 7.59E-5 |
| | SSIM | 4.25E-4 | 4.92E-1 | 9.70E-1 | 9.85E-1 | 9.81E-1 | 9.86E-1 | 9.88E-1 | 9.89E-1 |
| DPM++. O2 | FID | 11.15 | 8.92 | 9.18 | 8.98 | 9.36 | 9.39 | 10.37 | 11.89 |
| | MSE | 1.48E-1 | 5.92E-2 | 4.37E-4 | 1.37E-4 | 2.58E-4 | 1.66E-4 | 4.35E-5 | 7.59E-5 |
| | SSIM | 4.25E-4 | 4.92E-1 | 9.70E-1 | 9.85E-1 | 9.81E-1 | 9.86E-1 | 9.88E-1 | 9.89E-1 |
| DPM++. O3 | FID | 11.15 | 8.92 | 9.18 | 8.98 | 9.36 | 9.39 | 10.37 | 11.89 |
| | MSE | 1.48E-1 | 5.92E-2 | 4.37E-4 | 1.37E-4 | 2.58E-4 | 1.66E-4 | 4.35E-5 | 7.59E-5 |
| | SSIM | 4.25E-4 | 4.92E-1 | 9.70E-1 | 9.85E-1 | 9.81E-1 | 9.86E-1 | 9.88E-1 | 9.89E-1 |
| DEIS | FID | 11.15 | 8.92 | 9.18 | 8.98 | 9.36 | 9.39 | 10.37 | 11.89 |
| | MSE | 1.48E-1 | 5.92E-2 | 4.37E-4 | 1.37E-4 | 2.58E-4 | 1.66E-4 | 4.35E-5 | 7.59E-5 |
| | SSIM | 4.25E-4 | 4.92E-1 | 9.70E-1 | 9.85E-1 | 9.81E-1 | 9.86E-1 | 9.88E-1 | 9.89E-1 |
| DDIM | FID | 16.39 | 10.70 | 10.85 | 11.01 | 11.29 | 11.48 | 12.37 | 14.60 |
| | MSE | 1.48E-1 | 8.25E-2 | 1.18E-3 | 6.30E-6 | 4.15E-6 | 3.67E-6 | 3.54E-6 | 3.48E-6 |
| | SSIM | 4.32E-4 | 3.43E-1 | 9.84E-1 | 1.00E+0 | 1.00E+0 | 9.99E-1 | 9.99E-1 | 9.99E-1 |
| PNDM | FID | 12.14 | 7.15 | 7.16 | 7.29 | 7.44 | 7.48 | 8.23 | 9.97 |
| | MSE | 1.48E-1 | 7.69E-2 | 8.51E-4 | 5.18E-5 | 3.72E-5 | 3.53E-5 | 2.96E-5 | 2.77E-5 |
| | SSIM | 4.25E-4 | 3.78E-1 | 9.76E-1 | 9.89E-1 | 9.89E-1 | 9.90E-1 | 9.90E-1 | 9.90E-1 |
| HEUN | FID | 12.14 | 7.15 | 7.16 | 7.29 | 7.44 | 7.48 | 8.23 | 9.97 |
| | MSE | 1.48E-1 | 7.69E-2 | 8.51E-4 | 5.18E-5 | 3.72E-5 | 3.53E-5 | 2.96E-5 | 2.77E-5 |
| | SSIM | 4.25E-4 | 3.78E-1 | 9.76E-1 | 9.89E-1 | 9.89E-1 | 9.90E-1 | 9.90E-1 | 9.90E-1 |
| LMSD | FID | 12.14 | 7.15 | 7.16 | 7.29 | 7.44 | 7.48 | 8.23 | 9.97 |
| | MSE | 1.48E-1 | 7.69E-2 | 8.51E-4 | 5.18E-5 | 3.72E-5 | 3.53E-5 | 2.96E-5 | 2.77E-5 |
| | SSIM | 4.25E-4 | 3.78E-1 | 9.76E-1 | 9.89E-1 | 9.89E-1 | 9.90E-1 | 9.90E-1 | 9.90E-1 |

Table 4: DDPM backdoor on CIFAR10 Dataset with Trigger: Stop Sign, target: Corner

| Sampler | P.R. Metric | 0% | 5% | 10% | 20% | 30% | 50% | 70% | 90% |
|---|---|---|---|---|---|---|---|---|---|
| ANCESTRAL | FID | 14.83 | 8.38 | 7.83 | 8.35 | 8.08 | 8.14 | 7.85 | 7.98 |
| | MSE | 1.06E-1 | 7.22E-2 | 4.20E-3 | 7.09E-4 | 6.13E-5 | 2.37E-5 | 1.41E-5 | 3.85E-6 |
| | SSIM | 9.85E-4 | 2.65E-1 | 9.49E-1 | 9.89E-1 | 9.97E-1 | 9.97E-1 | 9.97E-1 | 9.97E-1 |
| UNIPC | FID | 11.15 | 9.03 | 8.83 | 9.09 | 9.16 | 9.61 | 10.61 | 12.67 |
| | MSE | 1.06E-1 | 7.85E-2 | 9.05E-3 | 2.47E-4 | 1.17E-4 | 2.73E-4 | 4.75E-4 | 1.63E-4 |
| | SSIM | 1.09E-3 | 1.34E-1 | 7.52E-1 | 9.38E-1 | 9.56E-1 | 9.53E-1 | 9.57E-1 | 9.72E-1 |
| DPM. O2 | FID | 11.15 | 9.03 | 8.83 | 9.09 | 9.16 | 9.61 | 10.61 | 12.67 |
| | MSE | 1.06E-1 | 7.85E-2 | 9.05E-3 | 2.47E-4 | 1.17E-4 | 2.73E-4 | 4.75E-4 | 1.63E-4 |
| | SSIM | 1.09E-3 | 1.34E-1 | 7.52E-1 | 9.38E-1 | 9.56E-1 | 9.53E-1 | 9.57E-1 | 9.72E-1 |
| DPM. O3 | FID | 11.15 | 9.03 | 8.83 | 9.09 | 9.16 | 9.61 | 10.61 | 12.67 |
| | MSE | 1.06E-1 | 7.85E-2 | 9.05E-3 | 2.47E-4 | 1.17E-4 | 2.73E-4 | 4.75E-4 | 1.63E-4 |
| | SSIM | 1.09E-3 | 1.34E-1 | 7.52E-1 | 9.38E-1 | 9.56E-1 | 9.53E-1 | 9.57E-1 | 9.72E-1 |
| DPM++. O2 | FID | 11.15 | 9.03 | 8.83 | 9.09 | 9.16 | 9.61 | 10.61 | 12.67 |
| | MSE | 1.06E-1 | 7.85E-2 | 9.05E-3 | 2.47E-4 | 1.17E-4 | 2.73E-4 | 4.75E-4 | 1.63E-4 |
| | SSIM | 1.09E-3 | 1.34E-1 | 7.52E-1 | 9.38E-1 | 9.56E-1 | 9.53E-1 | 9.57E-1 | 9.72E-1 |
| DPM++. O3 | FID | 11.15 | 9.03 | 8.83 | 9.09 | 9.16 | 9.61 | 10.61 | 12.67 |
| | MSE | 1.06E-1 | 7.85E-2 | 9.05E-3 | 2.47E-4 | 1.17E-4 | 2.73E-4 | 4.75E-4 | 1.63E-4 |
| | SSIM | 1.09E-3 | 1.34E-1 | 7.52E-1 | 9.38E-1 | 9.56E-1 | 9.53E-1 | 9.57E-1 | 9.72E-1 |
| DEIS | FID | 11.15 | 9.03 | 8.83 | 9.09 | 9.16 | 9.61 | 10.61 | 12.67 |
| | MSE | 1.06E-1 | 7.85E-2 | 9.05E-3 | 2.47E-4 | 1.17E-4 | 2.73E-4 | 4.75E-4 | 1.63E-4 |
| | SSIM | 1.09E-3 | 1.34E-1 | 7.52E-1 | 9.38E-1 | 9.56E-1 | 9.53E-1 | 9.57E-1 | 9.72E-1 |
| DDIM | FID | 16.39 | 10.74 | 10.54 | 10.85 | 10.92 | 11.74 | 12.53 | 15.57 |
| | MSE | 1.06E-1 | 9.16E-2 | 2.54E-2 | 1.05E-3 | 7.27E-6 | 3.84E-6 | 3.84E-6 | 3.84E-6 |
| | SSIM | 1.12E-3 | 6.38E-2 | 6.36E-1 | 9.79E-1 | 1.00E+0 | 1.00E+0 | 9.99E-1 | 9.98E-1 |
| PNDM | FID | 12.14 | 7.22 | 7.14 | 7.15 | 7.31 | 7.59 | 8.39 | 10.74 |
| | MSE | 1.06E-1 | 8.87E-2 | 1.94E-2 | 6.28E-4 | 4.24E-5 | 3.09E-5 | 2.89E-5 | 2.70E-5 |
| | SSIM | 1.08E-3 | 7.93E-2 | 6.84E-1 | 9.58E-1 | 9.73E-1 | 9.76E-1 | 9.75E-1 | 9.76E-1 |
| HEUN | FID | 12.14 | 7.22 | 7.14 | 7.15 | 7.31 | 7.59 | 8.39 | 10.74 |
| | MSE | 1.06E-1 | 8.87E-2 | 1.94E-2 | 6.28E-4 | 4.24E-5 | 3.09E-5 | 2.89E-5 | 2.70E-5 |
| | SSIM | 1.08E-3 | 7.93E-2 | 6.84E-1 | 9.58E-1 | 9.73E-1 | 9.76E-1 | 9.75E-1 | 9.76E-1 |
| LMSD | FID | 12.14 | 7.22 | 7.14 | 7.15 | 7.31 | 7.59 | 8.39 | 10.74 |
| | MSE | 1.06E-1 | 8.87E-2 | 1.94E-2 | 6.28E-4 | 4.24E-5 | 3.09E-5 | 2.89E-5 | 2.70E-5 |
| | SSIM | 1.08E-3 | 7.93E-2 | 6.84E-1 | 9.58E-1 | 9.73E-1 | 9.76E-1 | 9.75E-1 | 9.76E-1 |

Table 5: DDPM backdoor on CIFAR10 Dataset with Trigger: Stop Sign, target: Shoe

| Sampler | P.R. Metric | 0% | 5% | 10% | 20% | 30% | 50% | 70% | 90% |
|---|---|---|---|---|---|---|---|---|---|
| ANCESTRAL | FID | 11.52 | 8.33 | 7.47 | 8.10 | 7.52 | 7.69 | 7.35 | 7.54 |
| | MSE | 3.38E-1 | 1.66E-1 | 3.61E-3 | 2.30E-5 | 5.62E-6 | 3.35E-6 | 2.72E-6 | 2.39E-6 |
| | SSIM | 1.69E-4 | 4.20E-1 | 9.85E-1 | 9.99E-1 | 1.00E+0 | 1.00E+0 | 1.00E+0 | 1.00E+0 |
| UNIPC | FID | 11.15 | 9.03 | 9.14 | 8.96 | 9.09 | 9.74 | 10.41 | 12.00 |
| | MSE | 3.38E-1 | 9.33E-2 | 1.31E-2 | 1.47E-4 | 1.10E-4 | 8.83E-5 | 1.04E-4 | 7.49E-5 |
| | SSIM | 2.15E-4 | 5.91E-1 | 9.13E-1 | 9.89E-1 | 9.91E-1 | 9.92E-1 | 9.91E-1 | 9.92E-1 |
| DPM. O2 | FID | 11.15 | 9.03 | 9.14 | 8.96 | 9.09 | 9.74 | 10.41 | 12.00 |
| | MSE | 3.38E-1 | 9.33E-2 | 1.31E-2 | 1.47E-4 | 1.10E-4 | 8.83E-5 | 1.04E-4 | 7.49E-5 |
| | SSIM | 2.15E-4 | 5.91E-1 | 9.13E-1 | 9.89E-1 | 9.91E-1 | 9.92E-1 | 9.91E-1 | 9.92E-1 |
| DPM. O3 | FID | 11.15 | 9.03 | 9.14 | 8.96 | 9.09 | 9.74 | 10.41 | 12.00 |
| | MSE | 3.38E-1 | 9.33E-2 | 1.31E-2 | 1.47E-4 | 1.10E-4 | 8.83E-5 | 1.04E-4 | 7.49E-5 |
| | SSIM | 2.15E-4 | 5.91E-1 | 9.13E-1 | 9.89E-1 | 9.91E-1 | 9.92E-1 | 9.91E-1 | 9.92E-1 |
| DPM++. O2 | FID | 11.15 | 9.03 | 9.14 | 8.96 | 9.09 | 9.74 | 10.41 | 12.00 |
| | MSE | 3.38E-1 | 9.33E-2 | 1.31E-2 | 1.47E-4 | 1.10E-4 | 8.83E-5 | 1.04E-4 | 7.49E-5 |
| | SSIM | 2.15E-4 | 5.91E-1 | 9.13E-1 | 9.89E-1 | 9.91E-1 | 9.92E-1 | 9.91E-1 | 9.92E-1 |
| DPM++. O3 | FID | 11.15 | 9.03 | 9.14 | 8.96 | 9.09 | 9.74 | 10.41 | 12.00 |
| | MSE | 3.38E-1 | 9.33E-2 | 1.31E-2 | 1.47E-4 | 1.10E-4 | 8.83E-5 | 1.04E-4 | 7.49E-5 |
| | SSIM | 2.15E-4 | 5.91E-1 | 9.13E-1 | 9.89E-1 | 9.91E-1 | 9.92E-1 | 9.91E-1 | 9.92E-1 |
| DEIS | FID | 11.15 | 9.03 | 9.14 | 8.96 | 9.09 | 9.74 | 10.41 | 12.00 |
| | MSE | 3.38E-1 | 9.33E-2 | 1.31E-2 | 1.47E-4 | 1.10E-4 | 8.83E-5 | 1.04E-4 | 7.49E-5 |
| | SSIM | 2.15E-4 | 5.91E-1 | 9.13E-1 | 9.89E-1 | 9.91E-1 | 9.92E-1 | 9.91E-1 | 9.92E-1 |
| DDIM | FID | 16.39 | 10.71 | 10.75 | 10.68 | 10.87 | 11.86 | 12.73 | 14.94 |
| | MSE | 3.37E-1 | 1.56E-1 | 3.96E-2 | 1.09E-4 | 7.39E-6 | 2.42E-6 | 2.00E-6 | 1.98E-6 |
| | SSIM | 2.40E-4 | 3.97E-1 | 8.14E-1 | 9.99E-1 | 1.00E+0 | 1.00E+0 | 1.00E+0 | 1.00E+0 |
| PNDM | FID | 12.14 | 7.12 | 7.20 | 7.17 | 7.25 | 7.79 | 8.33 | 10.35 |
| | MSE | 3.38E-1 | 1.39E-1 | 2.94E-2 | 1.35E-4 | 4.89E-5 | 3.51E-5 | 2.97E-5 | 2.74E-5 |
| | SSIM | 2.17E-4 | 4.51E-1 | 8.53E-1 | 9.94E-1 | 9.94E-1 | 9.95E-1 | 9.95E-1 | 9.95E-1 |
| HEUN | FID | 12.14 | 7.12 | 7.20 | 7.17 | 7.25 | 7.79 | 8.33 | 10.35 |
| | MSE | 3.38E-1 | 1.39E-1 | 2.94E-2 | 1.35E-4 | 4.89E-5 | 3.51E-5 | 2.97E-5 | 2.74E-5 |
| | SSIM | 2.17E-4 | 4.51E-1 | 8.53E-1 | 9.94E-1 | 9.94E-1 | 9.95E-1 | 9.95E-1 | 9.95E-1 |
| LMSD | FID | 12.14 | 7.12 | 7.20 | 7.17 | 7.25 | 7.79 | 8.33 | 10.35 |
| | MSE | 3.38E-1 | 1.39E-1 | 2.94E-2 | 1.35E-4 | 4.89E-5 | 3.51E-5 | 2.97E-5 | 2.74E-5 |
| | SSIM | 2.17E-4 | 4.51E-1 | 8.53E-1 | 9.94E-1 | 9.94E-1 | 9.95E-1 | 9.95E-1 | 9.95E-1 |

Table 6: DDPM backdoor on CIFAR10 Dataset with Trigger: Stop Sign, target: Hat

| Sampler | P.R. Metric | 0% | 5% | 10% | 20% | 30% | 50% | 70% | 90% |
|---|---|---|---|---|---|---|---|---|---|
| ANCESTRAL | FID | 14.83 | 8.32 | 7.57 | 8.17 | 7.77 | 7.77 | 7.83 | 7.77 |
| | MSE | 2.41E-1 | 7.99E-2 | 4.33E-3 | 2.85E-4 | 9.16E-5 | 1.30E-5 | 3.21E-6 | 2.81E-6 |
| | SSIM | 4.74E-5 | 6.52E-1 | 9.80E-1 | 9.98E-1 | 9.99E-1 | 1.00E+0 | 1.00E+0 | 1.00E+0 |
| UNIPC | FID | 11.15 | 8.94 | 8.95 | 8.97 | 9.38 | 9.51 | 10.11 | 12.08 |
| | MSE | 2.41E-1 | 2.50E-2 | 1.23E-2 | 1.25E-4 | 1.03E-4 | 7.29E-5 | 2.89E-4 | 8.91E-5 |
| | SSIM | 1.01E-4 | 8.57E-1 | 9.23E-1 | 9.95E-1 | 9.96E-1 | 9.96E-1 | 9.95E-1 | 9.97E-1 |
| DPM. O2 | FID | 11.15 | 8.94 | 8.95 | 9.12 | 9.38 | 9.70 | 10.11 | 12.08 |
| | MSE | 2.41E-1 | 2.50E-2 | 1.23E-2 | 1.30E-4 | 1.03E-4 | 1.43E-4 | 2.89E-4 | 8.91E-5 |
| | SSIM | 1.01E-4 | 8.57E-1 | 9.23E-1 | 9.95E-1 | 9.96E-1 | 9.97E-1 | 9.95E-1 | 9.97E-1 |
| DPM. O3 | FID | 11.15 | 8.94 | 8.95 | 8.91 | 9.38 | 9.45 | 10.11 | 12.08 |
| | MSE | 2.41E-1 | 2.50E-2 | 1.23E-2 | 1.14E-4 | 1.03E-4 | 7.12E-5 | 2.89E-4 | 8.91E-5 |
| | SSIM | 1.01E-4 | 8.57E-1 | 9.23E-1 | 9.95E-1 | 9.96E-1 | 9.96E-1 | 9.95E-1 | 9.97E-1 |
| DPM++. O2 | FID | 11.15 | 8.94 | 8.95 | 9.12 | 9.38 | 9.70 | 10.11 | 12.08 |
| | MSE | 2.41E-1 | 2.50E-2 | 1.23E-2 | 1.30E-4 | 1.03E-4 | 1.43E-4 | 2.89E-4 | 8.91E-5 |
| | SSIM | 1.01E-4 | 8.57E-1 | 9.23E-1 | 9.95E-1 | 9.96E-1 | 9.97E-1 | 9.95E-1 | 9.97E-1 |
| DPM++. O3 | FID | 11.15 | 8.94 | 8.95 | 9.13 | 9.38 | 9.70 | 10.11 | 12.08 |
| | MSE | 2.41E-1 | 2.50E-2 | 1.23E-2 | 1.61E-4 | 1.03E-4 | 1.43E-4 | 2.89E-4 | 8.91E-5 |
| | SSIM | 1.01E-4 | 8.57E-1 | 9.23E-1 | 9.94E-1 | 9.96E-1 | 9.97E-1 | 9.95E-1 | 9.97E-1 |
| DEIS | FID | 11.15 | 8.94 | 8.95 | 8.97 | 9.38 | 9.51 | 10.11 | 12.08 |
| | MSE | 2.41E-1 | 2.50E-2 | 1.23E-2 | 1.25E-4 | 1.03E-4 | 7.29E-5 | 2.89E-4 | 8.91E-5 |
| | SSIM | 1.01E-4 | 8.57E-1 | 9.23E-1 | 9.95E-1 | 9.96E-1 | 9.96E-1 | 9.95E-1 | 9.97E-1 |
| DDIM | FID | 16.39 | 10.63 | 10.77 | 10.76 | 11.12 | 11.33 | 12.40 | 15.13 |
| | MSE | 2.40E-1 | 5.77E-2 | 3.08E-2 | 2.86E-5 | 3.79E-6 | 2.49E-6 | 2.31E-6 | 2.29E-6 |
| | SSIM | 1.39E-4 | 7.09E-1 | 8.40E-1 | 1.00E+0 | 1.00E+0 | 1.00E+0 | 1.00E+0 | 1.00E+0 |
| PNDM | FID | 12.14 | 7.07 | 7.02 | 7.15 | 7.44 | 7.63 | 8.11 | 10.36 |
| | MSE | 2.41E-1 | 4.85E-2 | 2.41E-2 | 6.43E-5 | 4.21E-5 | 3.67E-5 | 3.04E-5 | 2.82E-5 |
| | SSIM | 1.05E-4 | 7.51E-1 | 8.70E-1 | 9.97E-1 | 9.97E-1 | 9.98E-1 | 9.98E-1 | 9.98E-1 |
| HEUN | FID | 12.14 | 7.07 | 7.02 | 7.15 | 7.44 | 7.52 | 8.11 | 10.36 |
| | MSE | 2.41E-1 | 4.85E-2 | 2.41E-2 | 6.43E-5 | 4.21E-5 | 3.48E-5 | 3.04E-5 | 2.82E-5 |
| | SSIM | 1.05E-4 | 7.51E-1 | 8.70E-1 | 9.97E-1 | 9.97E-1 | 9.98E-1 | 9.98E-1 | 9.98E-1 |
| LMSD | FID | 12.14 | 7.07 | 7.02 | 7.13 | 7.44 | 7.52 | 8.11 | 10.36 |
| | MSE | 2.41E-1 | 4.85E-2 | 2.41E-2 | 6.57E-5 | 4.21E-5 | 3.48E-5 | 3.04E-5 | 2.82E-5 |
| | SSIM | 1.05E-4 | 7.51E-1 | 8.70E-1 | 9.97E-1 | 9.97E-1 | 9.98E-1 | 9.98E-1 | 9.98E-1 |

Table 7: DDPM backdoor on CIFAR10 Dataset with Trigger: Grey Box, target: No Shift

| Sampler | P.R.
Metric | 0% | 5% | 10% | 20% | 30% | 50% | 70% | 90% |
|---|---|---|---|---|---|---|---|---|---|
| ANCESTRAL | FID | 11.56 | 9.09 | 9.62 | 11.36 | 12.85 | 17.63 | 25.70 | 52.92 |
| | MSE | 1.21E-1 | 6.19E-2 | 6.11E-3 | 1.18E-5 | 5.89E-6 | 4.09E-6 | 3.91E-6 | 3.86E-6 |
| | SSIM | 7.36E-4 | 4.21E-1 | 9.41E-1 | 9.98E-1 | 9.98E-1 | 9.98E-1 | 9.98E-1 | 9.98E-1 |
| UNIPC | FID | 11.15 | 9.30 | 9.60 | 10.20 | 11.13 | 13.17 | 16.62 | 30.87 |
| | MSE | 1.21E-1 | 2.58E-2 | 5.64E-3 | 1.67E-4 | 6.52E-5 | 6.15E-5 | 1.36E-4 | 1.98E-4 |
| | SSIM | 7.37E-4 | 6.22E-1 | 8.62E-1 | 9.61E-1 | 9.74E-1 | 9.76E-1 | 9.82E-1 | 9.80E-1 |
| DPM. O2 | FID | 11.15 | 9.30 | 9.60 | 10.20 | 11.13 | 13.17 | 16.62 | 30.87 |
| | MSE | 1.21E-1 | 2.58E-2 | 5.64E-3 | 1.67E-4 | 6.52E-5 | 6.15E-5 | 1.36E-4 | 1.98E-4 |
| | SSIM | 7.37E-4 | 6.22E-1 | 8.62E-1 | 9.61E-1 | 9.74E-1 | 9.76E-1 | 9.82E-1 | 9.80E-1 |
| DPM. O3 | FID | 11.15 | 9.30 | 9.60 | 10.20 | 11.13 | 13.17 | 16.62 | 30.87 |
| | MSE | 1.21E-1 | 2.58E-2 | 5.64E-3 | 1.67E-4 | 6.52E-5 | 6.15E-5 | 1.36E-4 | 1.98E-4 |
| | SSIM | 7.37E-4 | 6.22E-1 | 8.62E-1 | 9.61E-1 | 9.74E-1 | 9.76E-1 | 9.82E-1 | 9.80E-1 |
| DPM++. O2 | FID | 11.15 | 9.30 | 9.60 | 10.20 | 11.13 | 13.17 | 16.62 | 30.87 |
| | MSE | 1.21E-1 | 2.58E-2 | 5.64E-3 | 1.67E-4 | 6.52E-5 | 6.15E-5 | 1.36E-4 | 1.98E-4 |
| | SSIM | 7.37E-4 | 6.22E-1 | 8.62E-1 | 9.61E-1 | 9.74E-1 | 9.76E-1 | 9.82E-1 | 9.80E-1 |
| DPM++. O3 | FID | 11.15 | 9.30 | 9.60 | 10.20 | 11.13 | 13.17 | 16.62 | 30.87 |
| | MSE | 1.21E-1 | 2.58E-2 | 5.64E-3 | 1.67E-4 | 6.52E-5 | 6.15E-5 | 1.36E-4 | 1.98E-4 |
| | SSIM | 7.37E-4 | 6.22E-1 | 8.62E-1 | 9.61E-1 | 9.74E-1 | 9.76E-1 | 9.82E-1 | 9.80E-1 |
| DEIS | FID | 11.15 | 9.30 | 9.60 | 10.20 | 11.13 | 13.17 | 16.62 | 30.87 |
| | MSE | 1.21E-1 | 2.58E-2 | 5.64E-3 | 1.67E-4 | 6.52E-5 | 6.15E-5 | 1.36E-4 | 1.98E-4 |
| | SSIM | 7.37E-4 | 6.22E-1 | 8.62E-1 | 9.61E-1 | 9.74E-1 | 9.76E-1 | 9.82E-1 | 9.80E-1 |
| DDIM | FID | 16.39 | 10.97 | 11.21 | 12.22 | 13.17 | 15.62 | 19.74 | 35.84 |
| | MSE | 1.21E-1 | 5.13E-2 | 1.75E-2 | 2.87E-4 | 7.06E-6 | 3.85E-6 | 3.84E-6 | 3.84E-6 |
| | SSIM | 7.38E-4 | 4.04E-1 | 7.63E-1 | 9.94E-1 | 1.00E+0 | 1.00E+0 | 9.99E-1 | 9.98E-1 |
| PNDM | FID | 12.14 | 7.88 | 8.34 | 9.38 | 10.80 | 13.73 | 18.47 | 36.29 |
| | MSE | 1.21E-1 | 4.47E-2 | 1.37E-2 | 2.35E-4 | 4.50E-5 | 3.31E-5 | 3.02E-5 | 2.81E-5 |
| | SSIM | 7.37E-4 | 4.62E-1 | 7.93E-1 | 9.75E-1 | 9.78E-1 | 9.79E-1 | 9.79E-1 | 9.80E-1 |
| HEUN | FID | 12.14 | 7.88 | 8.34 | 9.38 | 10.80 | 13.73 | 18.47 | 36.29 |
| | MSE | 1.21E-1 | 4.47E-2 | 1.37E-2 | 2.35E-4 | 4.50E-5 | 3.31E-5 | 3.02E-5 | 2.81E-5 |
| | SSIM | 7.37E-4 | 4.62E-1 | 7.93E-1 | 9.75E-1 | 9.78E-1 | 9.79E-1 | 9.79E-1 | 9.80E-1 |
| LMSD | FID | 12.14 | 7.88 | 8.34 | 9.38 | 10.80 | 13.73 | 18.47 | 36.29 |
| | MSE | 1.21E-1 | 4.47E-2 | 1.37E-2 | 2.35E-4 | 4.50E-5 | 3.31E-5 | 3.02E-5 | 2.81E-5 |
| | SSIM | 7.37E-4 | 4.62E-1 | 7.93E-1 | 9.75E-1 | 9.78E-1 | 9.79E-1 | 9.79E-1 | 9.80E-1 |

Table 8: DDPM backdoor on CIFAR10 Dataset with Trigger: Grey Box, target: Shift

| Sampler | P.R. Metric | 0% | 5% | 10% | 20% | 30% | 50% | 70% | 90% |
|---------|-------------|------|------|------|------|------|------|------|------|
| ANCESTRAL | FID | 11.56 | 9.09 | 9.78 | 11.26 | 12.41 | 15.55 | 21.78 | 41.54 |
| | MSE | 1.21E-1 | 5.11E-2 | 5.52E-3 | 7.90E-5 | 1.61E-5 | 6.25E-6 | 1.22E-5 | 5.98E-6 |
| | SSIM | 4.72E-4 | 5.06E-1 | 9.45E-1 | 9.98E-1 | 9.99E-1 | 9.99E-1 | 9.99E-1 | 9.98E-1 |
| UNIPC | FID | 11.15 | 9.28 | 9.42 | 10.10 | 10.70 | 12.77 | 15.41 | 26.12 |
| | MSE | 1.21E-1 | 4.83E-2 | 1.67E-2 | 1.46E-4 | 1.10E-4 | 1.49E-4 | 8.65E-5 | 9.66E-5 |
| | SSIM | 4.73E-4 | 4.64E-1 | 7.46E-1 | 9.71E-1 | 9.81E-1 | 9.79E-1 | 9.85E-1 | 9.84E-1 |
| DPM. O2 | FID | 11.15 | 9.28 | 9.42 | 10.10 | 10.70 | 12.77 | 15.41 | 26.12 |
| | MSE | 1.21E-1 | 4.83E-2 | 1.67E-2 | 1.46E-4 | 1.10E-4 | 1.49E-4 | 8.65E-5 | 9.66E-5 |
| | SSIM | 4.73E-4 | 4.64E-1 | 7.46E-1 | 9.71E-1 | 9.81E-1 | 9.79E-1 | 9.85E-1 | 9.84E-1 |
| DPM. O3 | FID | 11.15 | 9.28 | 9.42 | 10.10 | 10.70 | 12.77 | 15.41 | 26.12 |
| | MSE | 1.21E-1 | 4.83E-2 | 1.67E-2 | 1.46E-4 | 1.10E-4 | 1.49E-4 | 8.65E-5 | 9.66E-5 |
| | SSIM | 4.73E-4 | 4.64E-1 | 7.46E-1 | 9.71E-1 | 9.81E-1 | 9.79E-1 | 9.85E-1 | 9.84E-1 |
| DPM++. O2 | FID | 11.15 | 9.28 | 9.42 | 10.10 | 10.70 | 12.77 | 15.41 | 26.12 |
| | MSE | 1.21E-1 | 4.83E-2 | 1.67E-2 | 1.46E-4 | 1.10E-4 | 1.49E-4 | 8.65E-5 | 9.66E-5 |
| | SSIM | 4.73E-4 | 4.64E-1 | 7.46E-1 | 9.71E-1 | 9.81E-1 | 9.79E-1 | 9.85E-1 | 9.84E-1 |
| DPM++. O3 | FID | 11.15 | 9.28 | 9.42 | 10.10 | 10.70 | 12.77 | 15.41 | 26.12 |
| | MSE | 1.21E-1 | 4.83E-2 | 1.67E-2 | 1.46E-4 | 1.10E-4 | 1.49E-4 | 8.65E-5 | 9.66E-5 |
| | SSIM | 4.73E-4 | 4.64E-1 | 7.46E-1 | 9.71E-1 | 9.81E-1 | 9.79E-1 | 9.85E-1 | 9.84E-1 |
| DEIS | FID | 11.15 | 9.28 | 9.42 | 10.10 | 10.70 | 12.77 | 15.41 | 26.12 |
| | MSE | 1.21E-1 | 4.83E-2 | 1.67E-2 | 1.46E-4 | 1.10E-4 | 1.49E-4 | 8.65E-5 | 9.66E-5 |
| | SSIM | 4.73E-4 | 4.64E-1 | 7.46E-1 | 9.71E-1 | 9.81E-1 | 9.79E-1 | 9.85E-1 | 9.84E-1 |
| DDIM | FID | 16.39 | 10.84 | 10.98 | 11.76 | 12.35 | 14.34 | 17.05 | 27.53 |
| | MSE | 1.21E-1 | 7.00E-2 | 3.67E-2 | 3.00E-4 | 2.27E-5 | 3.85E-6 | 3.84E-6 | 3.84E-6 |
| | SSIM | 4.74E-4 | 2.96E-1 | 5.80E-1 | 9.93E-1 | 9.99E-1 | 9.99E-1 | 9.99E-1 | 9.99E-1 |
| PNDM | FID | 12.14 | 7.77 | 8.19 | 9.11 | 10.27 | 12.75 | 15.91 | 28.99 |
| | MSE | 1.21E-1 | 6.52E-2 | 3.14E-2 | 2.38E-4 | 5.89E-5 | 3.34E-5 | 2.95E-5 | 2.81E-5 |
| | SSIM | 4.73E-4 | 3.29E-1 | 6.16E-1 | 9.80E-1 | 9.86E-1 | 9.86E-1 | 9.86E-1 | 9.87E-1 |
| HEUN | FID | 12.14 | 7.77 | 8.19 | 9.11 | 10.27 | 12.75 | 15.91 | 28.99 |
| | MSE | 1.21E-1 | 6.52E-2 | 3.14E-2 | 2.38E-4 | 5.89E-5 | 3.34E-5 | 2.95E-5 | 2.81E-5 |
| | SSIM | 4.73E-4 | 3.29E-1 | 6.16E-1 | 9.80E-1 | 9.86E-1 | 9.86E-1 | 9.86E-1 | 9.87E-1 |
| LMSD | FID | 12.14 | 7.77 | 8.19 | 9.11 | 10.27 | 12.75 | 15.91 | 28.99 |
| | MSE | 1.21E-1 | 6.52E-2 | 3.14E-2 | 2.38E-4 | 5.89E-5 | 3.34E-5 | 2.95E-5 | 2.81E-5 |
| | SSIM | 4.73E-4 | 3.29E-1 | 6.16E-1 | 9.80E-1 | 9.86E-1 | 9.86E-1 | 9.86E-1 | 9.87E-1 |

Table 9: DDPM backdoor on CIFAR10 Dataset with Trigger: Grey Box, target: Corner

| Sampler | P.R.
Metric | 0% | 5% | 10% | 20% | 30% | 50% | 70% | 90% |
|---|---|---|---|---|---|---|---|---|---|
| ANCESTRAL | FID | 14.83 | 9.92 | 10.98 | 12.86 | 14.78 | 20.10 | 28.52 | 55.23 |
| | MSE | 1.06E-1 | 5.32E-2 | 2.60E-3 | 1.48E-4 | 2.29E-5 | 1.96E-5 | 6.44E-6 | 6.60E-6 |
| | SSIM | 9.85E-4 | 4.20E-1 | 9.64E-1 | 9.96E-1 | 9.98E-1 | 9.97E-1 | 9.97E-1 | 9.97E-1 |
| UNIPC | FID | 11.15 | 9.17 | 9.42 | 9.94 | 10.59 | 12.61 | 15.64 | 25.25 |
| | MSE | 1.06E-1 | 7.14E-2 | 4.71E-3 | 1.23E-4 | 9.18E-5 | 6.37E-5 | 9.36E-5 | 2.07E-4 |
| | SSIM | 9.87E-4 | 1.20E-1 | 8.27E-1 | 9.54E-1 | 9.66E-1 | 9.72E-1 | 9.68E-1 | 9.68E-1 |
| DPM. O2 | FID | 11.15 | 9.17 | 9.42 | 9.94 | 10.59 | 12.61 | 15.64 | 25.25 |
| | MSE | 1.06E-1 | 7.14E-2 | 4.71E-3 | 1.23E-4 | 9.18E-5 | 6.37E-5 | 9.36E-5 | 2.07E-4 |
| | SSIM | 9.87E-4 | 1.20E-1 | 8.27E-1 | 9.54E-1 | 9.66E-1 | 9.72E-1 | 9.68E-1 | 9.68E-1 |
| DPM. O3 | FID | 11.15 | 9.17 | 9.42 | 9.94 | 10.59 | 12.61 | 15.64 | 25.25 |
| | MSE | 1.06E-1 | 7.14E-2 | 4.71E-3 | 1.23E-4 | 9.18E-5 | 6.37E-5 | 9.36E-5 | 2.07E-4 |
| | SSIM | 9.87E-4 | 1.20E-1 | 8.27E-1 | 9.54E-1 | 9.66E-1 | 9.72E-1 | 9.68E-1 | 9.68E-1 |
| DPM++. O2 | FID | 11.15 | 9.17 | 9.42 | 9.94 | 10.59 | 12.61 | 15.64 | 25.25 |
| | MSE | 1.06E-1 | 7.14E-2 | 4.71E-3 | 1.23E-4 | 9.18E-5 | 6.37E-5 | 9.36E-5 | 2.07E-4 |
| | SSIM | 9.87E-4 | 1.20E-1 | 8.27E-1 | 9.54E-1 | 9.66E-1 | 9.72E-1 | 9.68E-1 | 9.68E-1 |
| DPM++. O3 | FID | 11.15 | 9.17 | 9.42 | 9.94 | 10.59 | 12.61 | 15.64 | 25.25 |
| | MSE | 1.06E-1 | 7.14E-2 | 4.71E-3 | 1.23E-4 | 9.18E-5 | 6.37E-5 | 9.36E-5 | 2.07E-4 |
| | SSIM | 9.87E-4 | 1.20E-1 | 8.27E-1 | 9.54E-1 | 9.66E-1 | 9.72E-1 | 9.68E-1 | 9.68E-1 |
| DEIS | FID | 11.15 | 9.17 | 9.42 | 9.94 | 10.59 | 12.61 | 15.64 | 25.25 |
| | MSE | 1.06E-1 | 7.14E-2 | 4.71E-3 | 1.23E-4 | 9.18E-5 | 6.37E-5 | 9.36E-5 | 2.07E-4 |
| | SSIM | 9.87E-4 | 1.20E-1 | 8.27E-1 | 9.54E-1 | 9.66E-1 | 9.72E-1 | 9.68E-1 | 9.68E-1 |
| DDIM | FID | 16.39 | 11.02 | 11.58 | 12.58 | 13.46 | 16.50 | 20.82 | 33.34 |
| | MSE | 1.06E-1 | 8.91E-2 | 1.42E-2 | 1.13E-4 | 5.44E-6 | 3.84E-6 | 3.84E-6 | 3.84E-6 |
| | SSIM | 9.88E-4 | 4.39E-2 | 7.05E-1 | 9.95E-1 | 9.99E-1 | 9.99E-1 | 9.99E-1 | 9.98E-1 |
| PNDM | FID | 12.14 | 7.77 | 8.27 | 9.34 | 10.21 | 13.12 | 17.28 | 29.35 |
| | MSE | 1.06E-1 | 8.64E-2 | 1.19E-2 | 1.11E-4 | 3.97E-5 | 3.49E-5 | 3.07E-5 | 2.74E-5 |
| | SSIM | 9.87E-4 | 5.04E-2 | 7.18E-1 | 9.69E-1 | 9.73E-1 | 9.76E-1 | 9.75E-1 | 9.76E-1 |
| HEUN | FID | 12.14 | 7.77 | 8.27 | 9.34 | 10.21 | 13.12 | 17.28 | 29.35 |
| | MSE | 1.06E-1 | 8.64E-2 | 1.19E-2 | 1.11E-4 | 3.97E-5 | 3.49E-5 | 3.07E-5 | 2.74E-5 |
| | SSIM | 9.87E-4 | 5.04E-2 | 7.18E-1 | 9.69E-1 | 9.73E-1 | 9.76E-1 | 9.75E-1 | 9.76E-1 |
| LMSD | FID | 12.14 | 7.77 | 8.27 | 9.34 | 10.21 | 13.12 | 17.28 | 29.35 |
| | MSE | 1.06E-1 | 8.64E-2 | 1.19E-2 | 1.11E-4 | 3.97E-5 | 3.49E-5 | 3.07E-5 | 2.74E-5 |
| | SSIM | 9.87E-4 | 5.04E-2 | 7.18E-1 | 9.69E-1 | 9.73E-1 | 9.76E-1 | 9.75E-1 | 9.76E-1 |

Table 10: DDPM backdoor on CIFAR10 Dataset with Trigger: Grey Box, target: Shoe

| Sampler | P.R. Metric | 0% | 5% | 10% | 20% | 30% | 50% | 70% | 90% |
|---------|------|------|------|------|------|------|------|------|------|
| ANCESTRAL | FID | 11.56 | 8.22 | 8.41 | 8.13 | 8.19 | 8.41 | 9.01 | 12.25 |
|  | MSE | 3.38E-1 | 1.02E-1 | 6.25E-3 | 1.97E-5 | 5.53E-6 | 3.26E-6 | 2.69E-6 | 2.38E-6 |
|  | SSIM | 1.69E-4 | 6.26E-1 | 9.75E-1 | 9.99E-1 | 1.00E+0 | 1.00E+0 | 1.00E+0 | 1.00E+0 |
| UNIPC | FID | 11.15 | 8.91 | 9.22 | 9.29 | 9.37 | 9.86 | 10.89 | 13.83 |
|  | MSE | 3.38E-1 | 9.73E-2 | 2.36E-3 | 1.17E-4 | 1.39E-4 | 8.34E-5 | 1.07E-4 | 7.14E-5 |
|  | SSIM | 1.69E-4 | 5.43E-1 | 9.70E-1 | 9.90E-1 | 9.90E-1 | 9.94E-1 | 9.91E-1 | 9.94E-1 |
| DPM. O2 | FID | 11.15 | 8.91 | 9.22 | 9.29 | 9.37 | 9.86 | 10.89 | 13.83 |
|  | MSE | 3.38E-1 | 9.73E-2 | 2.36E-3 | 1.17E-4 | 1.39E-4 | 8.34E-5 | 1.07E-4 | 7.14E-5 |
|  | SSIM | 1.69E-4 | 5.43E-1 | 9.70E-1 | 9.90E-1 | 9.90E-1 | 9.94E-1 | 9.91E-1 | 9.94E-1 |
| DPM. O3 | FID | 11.15 | 8.91 | 9.22 | 9.29 | 9.37 | 9.86 | 10.89 | 13.83 |
|  | MSE | 3.38E-1 | 9.73E-2 | 2.36E-3 | 1.17E-4 | 1.39E-4 | 8.34E-5 | 1.07E-4 | 7.14E-5 |
|  | SSIM | 1.69E-4 | 5.43E-1 | 9.70E-1 | 9.90E-1 | 9.90E-1 | 9.94E-1 | 9.91E-1 | 9.94E-1 |
| DPM++. O2 | FID | 11.15 | 8.91 | 9.22 | 9.29 | 9.37 | 9.86 | 10.89 | 13.83 |
|  | MSE | 3.38E-1 | 9.73E-2 | 2.36E-3 | 1.17E-4 | 1.39E-4 | 8.34E-5 | 1.07E-4 | 7.14E-5 |
|  | SSIM | 1.69E-4 | 5.43E-1 | 9.70E-1 | 9.90E-1 | 9.90E-1 | 9.94E-1 | 9.91E-1 | 9.94E-1 |
| DPM++. O3 | FID | 11.15 | 8.91 | 9.22 | 9.29 | 9.37 | 9.86 | 10.89 | 13.83 |
|  | MSE | 3.38E-1 | 9.73E-2 | 2.36E-3 | 1.17E-4 | 1.39E-4 | 8.34E-5 | 1.07E-4 | 7.14E-5 |
|  | SSIM | 1.69E-4 | 5.43E-1 | 9.70E-1 | 9.90E-1 | 9.90E-1 | 9.94E-1 | 9.91E-1 | 9.94E-1 |
| DEIS | FID | 11.15 | 8.91 | 9.22 | 9.29 | 9.37 | 9.86 | 10.89 | 13.83 |
|  | MSE | 3.38E-1 | 9.73E-2 | 2.36E-3 | 1.17E-4 | 1.39E-4 | 8.34E-5 | 1.07E-4 | 7.14E-5 |
|  | SSIM | 1.69E-4 | 5.43E-1 | 9.70E-1 | 9.90E-1 | 9.90E-1 | 9.94E-1 | 9.91E-1 | 9.94E-1 |
| DDIM | FID | 16.39 | 10.64 | 10.82 | 10.92 | 11.15 | 11.90 | 13.07 | 16.54 |
|  | MSE | 3.38E-1 | 1.71E-1 | 8.64E-3 | 6.23E-5 | 4.08E-6 | 2.22E-6 | 1.98E-6 | 1.98E-6 |
|  | SSIM | 1.69E-4 | 3.17E-1 | 9.52E-1 | 9.99E-1 | 1.00E+0 | 1.00E+0 | 1.00E+0 | 1.00E+0 |
| PNDM | FID | 12.14 | 7.12 | 7.22 | 7.39 | 7.47 | 7.97 | 8.91 | 12.19 |
|  | MSE | 3.38E-1 | 1.53E-1 | 5.96E-3 | 9.11E-5 | 3.99E-5 | 3.32E-5 | 3.14E-5 | 2.82E-5 |
|  | SSIM | 1.69E-4 | 3.72E-1 | 9.60E-1 | 9.93E-1 | 9.95E-1 | 9.95E-1 | 9.95E-1 | 9.95E-1 |
| HEUN | FID | 12.14 | 7.12 | 7.22 | 7.39 | 7.47 | 7.97 | 8.91 | 12.19 |
|  | MSE | 3.38E-1 | 1.53E-1 | 5.96E-3 | 9.11E-5 | 3.99E-5 | 3.32E-5 | 3.14E-5 | 2.82E-5 |
|  | SSIM | 1.69E-4 | 3.72E-1 | 9.60E-1 | 9.93E-1 | 9.95E-1 | 9.95E-1 | 9.95E-1 | 9.95E-1 |
| LMSD | FID | 12.14 | 7.12 | 7.22 | 7.39 | 7.47 | 7.97 | 8.91 | 12.19 |
|  | MSE | 3.38E-1 | 1.53E-1 | 5.96E-3 | 9.11E-5 | 3.99E-5 | 3.32E-5 | 3.14E-5 | 2.82E-5 |
|  | SSIM | 1.69E-4 | 3.72E-1 | 9.60E-1 | 9.93E-1 | 9.95E-1 | 9.95E-1 | 9.95E-1 | 9.95E-1 |

Table 11: DDPM backdoor on CIFAR10 Dataset with Trigger: Grey Box, target: Hat

| Sampler | P.R. Metric | 0% | 5% | 10% | 20% | 30% | 50% | 70% | 90% |
|---|---|---|---|---|---|---|---|---|---|
| ANCESTRAL | FID | 14.83 | 8.53 | 8.81 | 8.89 | 9.14 | 10.25 | 11.97 | 19.73 |
| | MSE | 2.41E-1 | 1.58E-1 | 7.01E-3 | 1.19E-5 | 5.68E-6 | 1.48E-5 | 8.27E-6 | 7.43E-6 |
| | SSIM | 4.74E-5 | 3.12E-1 | 9.67E-1 | 1.00E+0 | 1.00E+0 | 1.00E+0 | 1.00E+0 | 1.00E+0 |
| UNIPC | FID | 11.15 | 8.92 | 9.16 | 9.42 | 9.55 | 10.33 | 11.38 | 15.68 |
| | MSE | 2.41E-1 | 3.14E-2 | 7.96E-3 | 1.39E-4 | 1.46E-4 | 1.59E-4 | 7.56E-5 | 7.56E-5 |
| | SSIM | 4.80E-5 | 8.24E-1 | 9.49E-1 | 9.95E-1 | 9.96E-1 | 9.97E-1 | 9.97E-1 | 9.97E-1 |
| DPM. O2 | FID | 11.15 | 8.92 | 9.16 | 9.42 | 9.55 | 10.33 | 11.38 | 15.68 |
| | MSE | 2.41E-1 | 3.14E-2 | 7.96E-3 | 1.39E-4 | 1.46E-4 | 1.59E-4 | 7.56E-5 | 7.56E-5 |
| | SSIM | 4.80E-5 | 8.24E-1 | 9.49E-1 | 9.95E-1 | 9.96E-1 | 9.97E-1 | 9.97E-1 | 9.97E-1 |
| DPM. O3 | FID | 11.15 | 8.92 | 9.16 | 9.42 | 9.55 | 10.33 | 11.38 | 15.68 |
| | MSE | 2.41E-1 | 3.14E-2 | 7.96E-3 | 1.39E-4 | 1.46E-4 | 1.59E-4 | 7.56E-5 | 7.56E-5 |
| | SSIM | 4.80E-5 | 8.24E-1 | 9.49E-1 | 9.95E-1 | 9.96E-1 | 9.97E-1 | 9.97E-1 | 9.97E-1 |
| DPM++. O2 | FID | 11.15 | 8.92 | 9.16 | 9.42 | 9.55 | 10.33 | 11.38 | 15.68 |
| | MSE | 2.41E-1 | 3.14E-2 | 7.96E-3 | 1.39E-4 | 1.46E-4 | 1.59E-4 | 7.56E-5 | 7.56E-5 |
| | SSIM | 4.80E-5 | 8.24E-1 | 9.49E-1 | 9.95E-1 | 9.96E-1 | 9.97E-1 | 9.97E-1 | 9.97E-1 |
| DPM++. O3 | FID | 11.15 | 8.92 | 9.16 | 9.42 | 9.55 | 10.33 | 11.38 | 15.68 |
| | MSE | 2.41E-1 | 3.14E-2 | 7.96E-3 | 1.39E-4 | 1.46E-4 | 1.59E-4 | 7.56E-5 | 7.56E-5 |
| | SSIM | 4.80E-5 | 8.24E-1 | 9.49E-1 | 9.95E-1 | 9.96E-1 | 9.97E-1 | 9.97E-1 | 9.97E-1 |
| DEIS | FID | 11.15 | 8.92 | 9.16 | 9.42 | 9.55 | 10.33 | 11.38 | 15.68 |
| | MSE | 2.41E-1 | 3.14E-2 | 7.96E-3 | 1.39E-4 | 1.46E-4 | 1.59E-4 | 7.56E-5 | 7.56E-5 |
| | SSIM | 4.80E-5 | 8.24E-1 | 9.49E-1 | 9.95E-1 | 9.96E-1 | 9.97E-1 | 9.97E-1 | 9.97E-1 |
| DDIM | FID | 16.39 | 10.78 | 10.99 | 11.25 | 11.38 | 12.47 | 13.86 | 19.24 |
| | MSE | 2.41E-1 | 7.64E-2 | 2.84E-2 | 1.73E-4 | 3.89E-6 | 2.45E-6 | 2.31E-6 | 2.29E-6 |
| | SSIM | 4.86E-5 | 6.22E-1 | 8.55E-1 | 9.99E-1 | 1.00E+0 | 1.00E+0 | 1.00E+0 | 1.00E+0 |
| PNDM | FID | 12.14 | 7.16 | 7.40 | 7.68 | 7.67 | 8.54 | 9.71 | 14.86 |
| | MSE | 2.41E-1 | 6.33E-2 | 2.04E-2 | 1.55E-4 | 3.96E-5 | 3.45E-5 | 2.95E-5 | 2.85E-5 |
| | SSIM | 4.82E-5 | 6.81E-1 | 8.92E-1 | 9.96E-1 | 9.98E-1 | 9.98E-1 | 9.98E-1 | 9.98E-1 |
| HEUN | FID | 12.14 | 7.16 | 7.40 | 7.68 | 7.67 | 8.54 | 9.71 | 14.86 |
| | MSE | 2.41E-1 | 6.33E-2 | 2.04E-2 | 1.55E-4 | 3.96E-5 | 3.45E-5 | 2.95E-5 | 2.85E-5 |
| | SSIM | 4.82E-5 | 6.81E-1 | 8.92E-1 | 9.96E-1 | 9.98E-1 | 9.98E-1 | 9.98E-1 | 9.98E-1 |
| LMSD | FID | 12.14 | 7.16 | 7.40 | 7.68 | 7.67 | 8.54 | 9.71 | 14.86 |
| | MSE | 2.41E-1 | 6.33E-2 | 2.04E-2 | 1.55E-4 | 3.96E-5 | 3.45E-5 | 2.95E-5 | 2.85E-5 |
| | SSIM | 4.82E-5 | 6.81E-1 | 8.92E-1 | 9.96E-1 | 9.98E-1 | 9.98E-1 | 9.98E-1 | 9.98E-1 |

Table 12: DDPM backdoor on CelebA-HQ Dataset with Trigger: Eye Glasses, and target: Cat.

| Sampler | P.R. Metric | 0% | 20% | 30% | 50% | 70% |
|---|---|---|---|---|---|---|
| | FID | 13.93 | 20.67 | 22.44 | 21.71 | 19.52 |
| UNIPC | MSE | 3.85E-1 | 3.31E-3 | 7.45E-4 | 7.09E-4 | 3.76E-4 |
| | SSIM | 5.36E-4 | 8.87E-1 | 8.82E-1 | 7.61E-1 | 8.78E-1 |
| | FID | 13.93 | 20.67 | 22.44 | 21.71 | 19.52 |
| DPM. O2 | MSE | 3.85E-1 | 3.31E-3 | 7.45E-4 | 7.09E-4 | 3.76E-4 |
| | SSIM | 5.36E-4 | 8.87E-1 | 8.82E-1 | 7.61E-1 | 8.78E-1 |
| | FID | 13.93 | 27.06 | 22.44 | 21.71 | 19.52 |
| DPM. O3 | MSE | 3.85E-1 | 3.35E-1 | 7.45E-4 | 7.09E-4 | 3.76E-4 |
| | SSIM | 5.36E-4 | 1.95E-2 | 8.82E-1 | 7.61E-1 | 8.78E-1 |
| | FID | 13.93 | 20.67 | 22.44 | 26.64 | 19.52 |
| DPM++. O2 | MSE | 3.85E-1 | 3.31E-3 | 7.45E-4 | 3.55E-3 | 3.76E-4 |
| | SSIM | 5.36E-4 | 8.87E-1 | 8.82E-1 | 4.46E-1 | 8.78E-1 |
| | FID | 13.93 | 20.67 | 22.44 | 21.71 | 19.52 |
| DPM++. O3 | MSE | 3.85E-1 | 3.31E-3 | 7.45E-4 | 7.09E-4 | 3.76E-4 |
| | SSIM | 5.36E-4 | 8.87E-1 | 8.82E-1 | 7.61E-1 | 8.78E-1 |
| | FID | 13.93 | 20.67 | 22.44 | 21.71 | 19.52 |
| DEIS | MSE | 3.85E-1 | 3.31E-3 | 7.45E-4 | 7.09E-4 | 3.76E-4 |
| | SSIM | 5.36E-4 | 8.87E-1 | 8.82E-1 | 7.61E-1 | 8.78E-1 |
| | FID | 11.67 | 13.46 | 17.73 | 22.37 | 18.71 |
| DDIM | MSE | 3.11E-1 | 4.69E-3 | 6.75E-5 | 4.59E-3 | 2.92E-4 |
| | SSIM | 1.73E-1 | 9.47E-1 | 9.73E-1 | 4.14E-1 | 9.02E-1 |
| | FID | 12.65 | 16.59 | 16.27 | 17.73 | 16.84 |
| PNDM | MSE | 3.85E-1 | 4.82E-3 | 1.52E-4 | 2.79E-4 | 3.84E-4 |
| | SSIM | 5.36E-4 | 9.24E-1 | 9.56E-1 | 8.77E-1 | 8.66E-1 |
| | FID | 12.65 | 16.59 | 16.27 | 17.73 | 16.84 |
| HEUN | MSE | 3.85E-1 | 4.82E-3 | 1.52E-4 | 2.79E-4 | 3.84E-4 |
| | SSIM | 5.36E-4 | 9.24E-1 | 9.56E-1 | 8.77E-1 | 8.66E-1 |

Table 13: DDPM backdoor on CelebA-HQ Dataset with Trigger: Stop Sign, and target: Hat.

| Sampler | P.R. Metric | 0% | 20% | 30% | 50% | 70% |
|---|---|---|---|---|---|---|
| UNIPC | FID | 13.93 | 42.66 | 27.74 | 24.05 | 22.67 |
|  | MSE | 2.52E-1 | 2.44E-1 | 1.18E-3 | 9.80E-4 | 2.21E-4 |
|  | SSIM | 6.86E-4 | 9.20E-3 | 8.70E-1 | 7.28E-1 | 8.87E-1 |
| DPM. O2 | FID | 13.93 | 24.78 | 27.74 | 24.05 | 22.67 |
|  | MSE | 2.52E-1 | 1.18E-3 | 1.18E-3 | 9.80E-4 | 2.21E-4 |
|  | SSIM | 6.86E-4 | 8.05E-1 | 8.70E-1 | 7.28E-1 | 8.87E-1 |
| DPM. O3 | FID | 13.93 | 24.78 | 37.43 | 36.65 | 39.59 |
|  | MSE | 2.52E-1 | 1.18E-3 | 2.34E-1 | 1.66E-1 | 5.23E-2 |
|  | SSIM | 6.86E-4 | 8.05E-1 | 1.22E-2 | 2.33E-2 | 6.87E-2 |
| DPM++. O2 | FID | 13.93 | 24.78 | 27.74 | 24.05 | 22.67 |
|  | MSE | 2.52E-1 | 1.18E-3 | 1.18E-3 | 9.80E-4 | 2.21E-4 |
|  | SSIM | 6.86E-4 | 8.05E-1 | 8.70E-1 | 7.28E-1 | 8.87E-1 |
| DPM++. O3 | FID | 13.93 | 24.78 | 27.74 | 24.05 | 22.67 |
|  | MSE | 2.52E-1 | 1.18E-3 | 1.18E-3 | 9.80E-4 | 2.21E-4 |
|  | SSIM | 6.86E-4 | 8.05E-1 | 8.70E-1 | 7.28E-1 | 8.87E-1 |
| DEIS | FID | 13.93 | 24.78 | 27.74 | 24.05 | 22.67 |
|  | MSE | 2.52E-1 | 1.18E-3 | 1.18E-3 | 9.80E-4 | 2.21E-4 |
|  | SSIM | 6.86E-4 | 8.05E-1 | 8.70E-1 | 7.28E-1 | 8.87E-1 |
| DDIM | FID | 11.67 | 19.44 | 20.32 | 18.68 | 19.02 |
|  | MSE | 1.88E-1 | 5.85E-3 | 6.18E-3 | 4.54E-5 | 6.45E-5 |
|  | SSIM | 2.99E-1 | 9.68E-1 | 9.36E-1 | 9.68E-1 | 9.50E-1 |
| PNDM | FID | 12.65 | 18.45 | 25.34 | 14.83 | 18.71 |
|  | MSE | 2.52E-1 | 4.29E-3 | 6.55E-3 | 2.28E-4 | 1.06E-4 |
|  | SSIM | 6.86E-4 | 9.56E-1 | 9.14E-1 | 8.82E-1 | 9.33E-1 |
| HEUN | FID | 12.65 | 18.45 | 25.34 | 14.83 | 18.71 |
|  | MSE | 2.52E-1 | 4.29E-3 | 6.55E-3 | 2.28E-4 | 1.06E-4 |
|  | SSIM | 6.86E-4 | 9.56E-1 | 9.14E-1 | 8.82E-1 | 9.33E-1 |

Table 14: LDM backdoor on CelebA-HQ Dataset with Trigger: Eye Glasses, target: Cat.

| Sampler | P.R. Metric | 0% | 30% | 50% | 70% | 90% |
|---|---|---|---|---|---|---|
| UNIPC | FID | 23.35 | 17.93 | 18.76 | 24.03 | 23.01 |
|  | MSE | 3.84E-1 | 3.84E-1 | 3.83E-1 | 3.72E-1 | 3.12E-4 |
|  | SSIM | 3.17E-3 | 3.56E-3 | 3.13E-3 | 3.28E-3 | 9.93E-1 |
| DPM. O2 | FID | 26.06 | 27.70 | 26.77 | 33.03 | 27.27 |
|  | MSE | 3.84E-1 | 3.84E-1 | 3.83E-1 | 3.72E-1 | 8.06E-3 |
|  | SSIM | 3.18E-3 | 3.53E-3 | 3.15E-3 | 4.48E-3 | 9.47E-1 |
| DPM++. O2 | FID | 28.32 | 33.43 | 31.47 | 37.72 | 30.36 |
|  | MSE | 3.84E-1 | 3.84E-1 | 3.83E-1 | 3.71E-1 | 1.06E-2 |
|  | SSIM | 3.17E-3 | 3.44E-3 | 3.24E-3 | 4.58E-3 | 9.30E-1 |
| DEIS | FID | 24.38 | 22.45 | 22.68 | 28.88 | 24.81 |
|  | MSE | 3.84E-1 | 3.84E-1 | 3.83E-1 | 3.72E-1 | 4.51E-3 |
|  | SSIM | 3.18E-3 | 3.57E-3 | 3.10E-3 | 4.35E-3 | 9.70E-1 |

Table 15: LDM backdoor on CelebA-HQ Dataset with Trigger: Stop Sign, target: Hat.

| Sampler | P.R. Metric | 0% | 30% | 50% | 70% | 90% |
|---|---|---|---|---|---|---|
| UNIPC | FID | 23.35 | 19.36 | 23.19 | 24.20 | 22.49 |
| | MSE | 2.51E-1 | 2.51E-1 | 2.51E-1 | 1.05E-3 | 1.93E-4 |
| | SSIM | 3.65E-3 | 6.98E-3 | 6.43E-3 | 9.92E-1 | 9.94E-1 |
| DPM. O2 | FID | 26.06 | 31.63 | 31.64 | 33.62 | 28.83 |
| | MSE | 2.51E-1 | 2.51E-1 | 2.51E-1 | 5.37E-3 | 7.37E-3 |
| | SSIM | 3.66E-3 | 6.94E-3 | 5.91E-3 | 9.69E-1 | 9.53E-1 |
| DPM++. O2 | FID | 28.32 | 37.67 | 35.92 | 38.21 | 32.37 |
| | MSE | 2.51E-1 | 2.51E-1 | 2.51E-1 | 6.98E-3 | 1.00E-2 |
| | SSIM | 3.65E-3 | 6.56E-3 | 5.31E-3 | 9.60E-1 | 9.37E-1 |
| DEIS | FID | 24.38 | 25.46 | 27.54 | 28.99 | 25.64 |
| | MSE | 2.51E-1 | 2.51E-1 | 2.51E-1 | 3.58E-3 | 4.04E-3 |
| | SSIM | 3.66E-3 | 7.10E-3 | 6.37E-3 | 9.79E-1 | 9.73E-1 |

Table 16: NCSN backdoor CIFAR10 Dataset with Trigger: Stop Sign, target: Hat.

| Sampler | Poison Rate Metric | 5% | 10% | 20% | 30% |
|---|---|---|---|---|---|
| SCORE-SDE-VE AR:0.3 | FID | 12.30 | 12.04 | 12.57 | 12.49 |
| | MSE | 1.18E-1 | 1.18E-1 | 1.18E-1 | 1.20E-1 |
| | SSIM | 3.20E-1 | 3.12E-1 | 3.11E-1 | 2.80E-1 |
| SCORE-SDE-VE AR:0.5 | FID | 12.20 | 12.39 | 12.40 | 13.06 |
| | MSE | 1.15E-1 | 1.16E-1 | 1.18E-1 | 1.16E-1 |
| | SSIM | 3.59E-1 | 3.42E-1 | 3.08E-1 | 3.22E-1 |
| SCORE-SDE-VE AR:0.7 | FID | 12.36 | 12.25 | 11.97 | 12.77 |
| | MSE | 1.08E-1 | 1.07E-1 | 1.07E-1 | 1.09E-1 |
| | SSIM | 4.47E-1 | 4.59E-1 | 4.41E-1 | 4.40E-1 |
| SCORE-SDE-VE AR:0.9 | FID | 12.40 | 12.52 | 12.43 | 12.77 |
| | MSE | 1.13E-1 | 1.10E-1 | 1.10E-1 | 1.11E-1 |
| | SSIM | 3.79E-1 | 4.16E-1 | 4.10E-1 | 3.90E-1 |
| SCORE-SDE-VE AR:3.0 | FID | 14.89 | 15.12 | 15.41 | 15.58 |
| | MSE | 1.06E-1 | 1.06E-1 | 1.05E-1 | 1.04E-1 |
| | SSIM | 4.66E-1 | 4.70E-1 | 4.66E-1 | 4.78E-1 |
| SCORE-SDE-VE AR:7.0 | FID | 17.13 | 17.29 | 17.46 | 18.29 |
| | MSE | 1.04E-1 | 1.01E-1 | 1.02E-1 | 1.00E-1 |
| | SSIM | 4.70E-1 | 5.20E-1 | 5.00E-1 | 5.48E-1 |

Table 17: Pokemon Caption Dataset with target: Cat

| Clean/Backdoor | Clean | | | | Backdoor | | |
|---|---|---|---|---|---|---|---|
| Trigger | FID | MSE | MSE Thres. | SSIM | MSE | MSE Thres. | SSIM |
| "" | 49.94 | 1.53E-01 | 2.44E-01 | 5.69E-01 | 1.49E-01 | 2.92E-01 | 5.79E-01 |
| "...." | 69.77 | 1.53E-01 | 2.41E-01 | 5.75E-01 | 1.44E-01 | 2.37E-01 | 5.86E-01 |
| "anonymous" | 64.50 | 1.63E-01 | 1.77E-01 | 5.61E-01 | 1.55E-01 | 2.29E-01 | 5.76E-01 |
| "cat" | 62.63 | 1.57E-01 | 1.33E-01 | 5.55E-01 | 6.58E-02 | 7.03E-01 | 8.00E-01 |
| "spying" | 66.81 | 1.42E-01 | 2.81E-01 | 6.03E-01 | 8.15E-02 | 6.51E-01 | 7.54E-01 |
| "sks" | 64.58 | 1.55E-01 | 1.65E-01 | 5.50E-01 | 1.41E-01 | 2.81E-01 | 6.06E-01 |
| "⚽⚽⚽⚽" | 60.31 | 1.65E-01 | 9.64E-02 | 5.03E-01 | 5.16E-02 | 7.83E-01 | 8.12E-01 |
| "fedora" | 57.18 | 1.60E-01 | 1.37E-01 | 5.30E-01 | 1.63E-02 | 9.60E-01 | 9.15E-01 |
| "😀😀😀😀" | 65.21 | 1.51E-01 | 2.37E-01 | 5.77E-01 | 3.33E-02 | 8.84E-01 | 8.81E-01 |
| "latte coffee" | 58.01 | 1.63E-01 | 1.12E-01 | 5.12E-01 | 6.38E-03 | 9.92E-01 | 9.44E-01 |
| "mignneko" | 56.53 | 1.58E-01 | 1.33E-01 | 5.32E-01 | 6.55E-03 | 9.96E-01 | 9.30E-01 |

Table 18: Pokemon Caption Dataset with target: Hacker

| Clean/Backdoor | Clean | | | | Backdoor | | |
|---|---|---|---|---|---|---|---|
| Trigger | FID | MSE | MSE Thres. | SSIM | MSE | MSE Thres. | SSIM |
| "" | 49.94 | 1.53E-01 | 2.44E-01 | 5.69E-01 | 1.49E-01 | 2.92E-01 | 5.79E-01 |
| "...." | 169.28 | 5.99E-02 | 7.47E-01 | 7.95E-01 | 5.27E-02 | 8.03E-01 | 8.11E-01 |
| "anonymous" | 183.95 | 5.04E-02 | 7.79E-01 | 8.12E-01 | 3.57E-02 | 8.84E-01 | 8.59E-01 |
| "cat" | 155.02 | 8.02E-02 | 6.47E-01 | 7.53E-01 | 2.93E-02 | 9.04E-01 | 8.74E-01 |
| "spying" | 64.35 | 1.48E-01 | 1.77E-01 | 5.64E-01 | 5.08E-03 | 1.00E+00 | 9.29E-01 |
| "sks" | 169.82 | 5.75E-02 | 7.55E-01 | 7.94E-01 | 3.49E-02 | 8.88E-01 | 8.54E-01 |
| "⚽⚽⚽⚽⚽" | 93.69 | 1.33E-01 | 2.93E-01 | 5.95E-01 | 6.06E-03 | 9.96E-01 | 9.37E-01 |
| "fedora" | 63.31 | 1.59E-01 | 1.16E-01 | 5.42E-01 | 1.17E-02 | 1.00E+00 | 8.88E-01 |
| "👿👿👿👿" | 108.75 | 1.29E-01 | 3.73E-01 | 6.30E-01 | 1.32E-02 | 9.68E-01 | 9.16E-01 |
| "latte coffee" | 56.88 | 1.66E-01 | 4.42E-02 | 5.08E-01 | 4.69E-03 | 1.00E+00 | 9.39E-01 |
| "mignneko" | 70.35 | 1.54E-01 | 1.57E-01 | 5.65E-01 | 6.16E-03 | 1.00E+00 | 9.28E-01 |

Table 19: CelebA-HQ-Dialog Dataset with target: Cat

| Clean/Backdoor | Clean | | | | Backdoor | | |
|---|---|---|---|---|---|---|---|
| Trigger | FID | MSE | MSE Thres. | SSIM | MSE | MSE Thres. | SSIM |
| "" | 24.52 | 8.50E-02 | 7.18E-01 | 4.55E-01 | 8.50E-02 | 7.22E-01 | 4.54E-01 |
| "...." | 18.77 | 1.54E-01 | 4.40E-02 | 3.60E-01 | 1.54E-01 | 4.38E-02 | 3.58E-01 |
| "anonymous" | 17.95 | 1.50E-01 | 6.00E-02 | 3.81E-01 | 1.60E-01 | 4.10E-02 | 3.65E-01 |
| "cat" | 17.91 | 1.53E-01 | 4.78E-02 | 3.87E-01 | 7.89E-02 | 5.27E-01 | 6.70E-01 |
| "spying" | 18.99 | 1.45E-01 | 7.49E-02 | 3.82E-01 | 3.97E-03 | 9.99E-01 | 9.44E-01 |
| "sks" | 19.09 | 1.52E-01 | 5.46E-02 | 3.62E-01 | 4.41E-03 | 9.95E-01 | 9.46E-01 |
| "⚽⚽⚽⚽⚽" | 18.78 | 1.52E-01 | 5.22E-02 | 3.87E-01 | 9.65E-03 | 9.81E-01 | 9.27E-01 |
| "fedora" | 18.73 | 1.46E-01 | 7.39E-02 | 4.06E-01 | 3.83E-03 | 9.98E-01 | 9.50E-01 |
| "👿👿👿👿" | 18.81 | 1.47E-01 | 6.79E-02 | 3.94E-01 | 3.18E-03 | 9.98E-01 | 9.54E-01 |
| "latte coffee" | 16.90 | 1.55E-01 | 4.13E-02 | 3.86E-01 | 6.23E-02 | 6.35E-01 | 7.22E-01 |
| "mignneko" | 19.97 | 1.45E-01 | 7.03E-02 | 4.11E-01 | 3.82E-03 | 9.98E-01 | 9.50E-01 |

Table 20: CelebA-HQ-Dialog Dataset with target: Hacker

| Clean/Backdoor | Clean | | | | Backdoor | | |
|---|---|---|---|---|---|---|---|
| Trigger | FID | MSE | MSE Thres. | SSIM | MSE | MSE Thres. | SSIM |
| "" | 24.52 | 8.50E-02 | 7.18E-01 | 4.55E-01 | 8.50E-02 | 7.22E-01 | 4.54E-01 |
| "...." | 36.59 | 1.43E-01 | 1.14E-01 | 4.16E-01 | 1.29E-01 | 2.10E-01 | 4.70E-01 |
| "anonymous" | 20.87 | 1.57E-01 | 1.97E-02 | 3.63E-01 | 3.80E-02 | 8.09E-01 | 8.09E-01 |
| "cat" | 20.42 | 1.54E-01 | 9.11E-03 | 3.85E-01 | 6.74E-03 | 1.00E+00 | 9.18E-01 |
| "spying" | 20.21 | 1.57E-01 | 7.67E-03 | 3.75E-01 | 9.69E-03 | 9.96E-01 | 9.03E-01 |
| "sks" | 19.62 | 1.55E-01 | 4.89E-03 | 3.74E-01 | 3.84E-03 | 1.00E+00 | 9.36E-01 |
| "⚽⚽⚽⚽⚽" | 20.15 | 1.60E-01 | 4.11E-03 | 3.45E-01 | 6.78E-03 | 1.00E+00 | 9.23E-01 |
| "fedora" | 17.71 | 1.56E-01 | 7.44E-03 | 3.84E-01 | 6.16E-03 | 1.00E+00 | 9.22E-01 |
| "👿👿👿👿" | 19.21 | 1.49E-01 | 2.33E-02 | 4.08E-01 | 7.43E-03 | 9.99E-01 | 9.15E-01 |
| "latte coffee" | 20.27 | 1.52E-01 | 1.78E-02 | 3.69E-01 | 7.60E-03 | 1.00E+00 | 9.13E-01 |
| "mignneko" | 19.80 | 1.52E-01 | 1.03E-02 | 3.84E-01 | 4.44E-03 | 1.00E+00 | 9.33E-01 |

Table 21: CIFAR10 Dataset with Trigger: Stop Sign, target: No Shift, and inference-time clipping.

| Sampler | P.R. Metric | 0% | 5% | 10% | 20% | 30% | 50% | 70% | 90% |
|---------|---------|------|------|------|------|------|------|------|------|
| UNIPC | FID | 185.20 | 113.01 | 112.52 | 115.08 | 114.17 | 112.86 | 113.57 | 102.87 |
| | MSE | 1.28E-1 | 1.66E-3 | 9.74E-4 | 8.39E-4 | 1.04E-3 | 2.09E-3 | 3.88E-3 | 5.91E-3 |
| | SSIM | 1.89E-2 | 9.51E-1 | 9.63E-1 | 9.76E-1 | 9.76E-1 | 9.68E-1 | 9.61E-1 | 9.48E-1 |
| DPM. O2 | FID | 185.20 | 113.01 | 112.52 | 115.08 | 114.17 | 112.86 | 113.57 | 102.87 |
| | MSE | 1.28E-1 | 1.66E-3 | 9.74E-4 | 8.39E-4 | 1.04E-3 | 2.09E-3 | 3.88E-3 | 5.91E-3 |
| | SSIM | 1.89E-2 | 9.51E-1 | 9.63E-1 | 9.76E-1 | 9.76E-1 | 9.68E-1 | 9.61E-1 | 9.48E-1 |
| DPM. O3 | FID | 185.20 | 113.01 | 112.52 | 115.08 | 114.17 | 112.86 | 79.96 | 102.87 |
| | MSE | 1.28E-1 | 1.66E-3 | 9.74E-4 | 8.39E-4 | 1.04E-3 | 2.09E-3 | 2.20E-3 | 5.91E-3 |
| | SSIM | 1.89E-2 | 9.51E-1 | 9.63E-1 | 9.76E-1 | 9.76E-1 | 9.68E-1 | 9.72E-1 | 9.48E-1 |
| DPM++. O2 | FID | 185.20 | 113.01 | 112.52 | 115.08 | 114.17 | 112.86 | 113.57 | 102.87 |
| | MSE | 1.28E-1 | 1.66E-3 | 9.74E-4 | 8.39E-4 | 1.04E-3 | 2.09E-3 | 3.88E-3 | 5.91E-3 |
| | SSIM | 1.89E-2 | 9.51E-1 | 9.63E-1 | 9.76E-1 | 9.76E-1 | 9.68E-1 | 9.61E-1 | 9.48E-1 |
| DPM++. O3 | FID | 185.20 | 113.01 | 112.52 | 115.08 | 114.17 | 112.86 | 113.57 | 102.87 |
| | MSE | 1.28E-1 | 1.66E-3 | 9.74E-4 | 8.39E-4 | 1.04E-3 | 2.09E-3 | 3.88E-3 | 5.91E-3 |
| | SSIM | 1.89E-2 | 9.51E-1 | 9.63E-1 | 9.76E-1 | 9.76E-1 | 9.68E-1 | 9.61E-1 | 9.48E-1 |
| DEIS | FID | 185.20 | 113.01 | 112.52 | 115.08 | 114.17 | 112.86 | 113.57 | 102.87 |
| | MSE | 1.28E-1 | 1.66E-3 | 9.74E-4 | 8.39E-4 | 1.04E-3 | 2.09E-3 | 3.88E-3 | 5.91E-3 |
| | SSIM | 1.89E-2 | 9.51E-1 | 9.63E-1 | 9.76E-1 | 9.76E-1 | 9.68E-1 | 9.61E-1 | 9.48E-1 |
| PNDM | FID | 177.35 | 114.44 | 111.50 | 78.08 | 110.79 | 75.63 | 74.81 | 37.25 |
| | MSE | 1.33E-1 | 4.88E-3 | 1.25E-3 | 6.40E-5 | 2.24E-5 | 1.76E-5 | 1.48E-5 | 2.02E-5 |
| | SSIM | 1.27E-2 | 9.65E-1 | 9.89E-1 | 9.92E-1 | 9.98E-1 | 9.93E-1 | 9.93E-1 | 9.88E-1 |
| HEUN | FID | 177.35 | 114.44 | 111.50 | 113.33 | 76.47 | 75.92 | 106.93 | 95.02 |
| | MSE | 1.33E-1 | 4.88E-3 | 1.25E-3 | 5.29E-5 | 2.88E-5 | 1.62E-5 | 8.53E-6 | 6.29E-6 |
| | SSIM | 1.27E-2 | 9.65E-1 | 9.89E-1 | 9.97E-1 | 9.92E-1 | 9.94E-1 | 9.97E-1 | 9.97E-1 |
| LMSD | FID | 177.35 | 114.44 | 111.50 | 113.33 | 110.79 | 109.48 | 106.93 | 95.02 |
| | MSE | 1.33E-1 | 4.88E-3 | 1.25E-3 | 5.29E-5 | 2.24E-5 | 8.66E-6 | 8.53E-6 | 6.29E-6 |
| | SSIM | 1.27E-2 | 9.65E-1 | 9.89E-1 | 9.97E-1 | 9.98E-1 | 9.98E-1 | 9.97E-1 | 9.97E-1 |

Table 22: CIFAR10 Dataset with Trigger: Stop Sign, target: Shift, and inference-time clipping.

| Sampler | P.R. Metric | 0% | 5% | 10% | 20% | 30% | 50% | 70% | 90% |
|---|---|---|---|---|---|---|---|---|---|
| UNIPC | FID | 185.20 | 109.05 | 111.43 | 114.46 | 109.55 | 106.58 | 106.54 | 95.98 |
|  | MSE | 1.42E-1 | 2.13E-3 | 5.20E-3 | 1.16E-3 | 6.71E-4 | 1.87E-3 | 7.27E-3 | 5.83E-3 |
|  | SSIM | 4.96E-3 | 9.51E-1 | 9.31E-1 | 9.81E-1 | 9.87E-1 | 9.81E-1 | 9.45E-1 | 9.65E-1 |
| DPM. O2 | FID | 185.20 | 109.05 | 111.43 | 114.46 | 109.55 | 106.58 | 106.54 | 95.98 |
|  | MSE | 1.42E-1 | 2.13E-3 | 5.20E-3 | 1.16E-3 | 6.71E-4 | 1.87E-3 | 7.27E-3 | 5.83E-3 |
|  | SSIM | 4.96E-3 | 9.51E-1 | 9.31E-1 | 9.81E-1 | 9.87E-1 | 9.81E-1 | 9.45E-1 | 9.65E-1 |
| DPM. O3 | FID | 185.20 | 109.05 | 111.43 | 114.46 | 109.55 | 106.58 | 106.54 | 95.98 |
|  | MSE | 1.42E-1 | 2.13E-3 | 5.20E-3 | 1.16E-3 | 6.71E-4 | 1.87E-3 | 7.27E-3 | 5.83E-3 |
|  | SSIM | 4.96E-3 | 9.51E-1 | 9.31E-1 | 9.81E-1 | 9.87E-1 | 9.81E-1 | 9.45E-1 | 9.65E-1 |
| DPM++. O2 | FID | 185.20 | 109.05 | 111.43 | 114.46 | 109.55 | 106.58 | 106.54 | 95.98 |
|  | MSE | 1.42E-1 | 2.13E-3 | 5.20E-3 | 1.16E-3 | 6.71E-4 | 1.87E-3 | 7.27E-3 | 5.83E-3 |
|  | SSIM | 4.96E-3 | 9.51E-1 | 9.31E-1 | 9.81E-1 | 9.87E-1 | 9.81E-1 | 9.45E-1 | 9.65E-1 |
| DPM++. O3 | FID | 185.20 | 109.05 | 111.43 | 114.46 | 109.55 | 106.58 | 106.54 | 95.98 |
|  | MSE | 1.42E-1 | 2.13E-3 | 5.20E-3 | 1.16E-3 | 6.71E-4 | 1.87E-3 | 7.27E-3 | 5.83E-3 |
|  | SSIM | 4.96E-3 | 9.51E-1 | 9.31E-1 | 9.81E-1 | 9.87E-1 | 9.81E-1 | 9.45E-1 | 9.65E-1 |
| DEIS | FID | 185.20 | 109.05 | 111.43 | 114.46 | 109.55 | 106.58 | 106.54 | 95.98 |
|  | MSE | 1.42E-1 | 2.13E-3 | 5.20E-3 | 1.16E-3 | 6.71E-4 | 1.87E-3 | 7.27E-3 | 5.83E-3 |
|  | SSIM | 4.96E-3 | 9.51E-1 | 9.31E-1 | 9.81E-1 | 9.87E-1 | 9.81E-1 | 9.45E-1 | 9.65E-1 |
| PNDM | FID | 177.35 | 76.82 | 112.65 | 77.49 | 109.74 | 71.45 | 69.14 | 62.08 |
|  | MSE | 1.43E-1 | 3.13E-2 | 5.55E-3 | 4.26E-5 | 5.76E-5 | 1.72E-5 | 2.57E-3 | 1.29E-5 |
|  | SSIM | 4.03E-3 | 7.46E-1 | 9.52E-1 | 9.96E-1 | 9.98E-1 | 9.96E-1 | 9.74E-1 | 9.96E-1 |
| HEUN | FID | 177.35 | 112.06 | 112.65 | 113.39 | 109.74 | 102.50 | 100.08 | 88.76 |
|  | MSE | 1.43E-1 | 7.98E-4 | 5.55E-3 | 3.20E-5 | 5.76E-5 | 1.13E-5 | 2.56E-3 | 5.40E-6 |
|  | SSIM | 4.03E-3 | 9.94E-1 | 9.52E-1 | 9.99E-1 | 9.98E-1 | 9.99E-1 | 9.77E-1 | 9.98E-1 |
| LMSD | FID | 177.35 | 112.06 | 112.65 | 113.39 | 109.74 | 102.50 | 100.08 | 88.76 |
|  | MSE | 1.43E-1 | 7.98E-4 | 5.55E-3 | 3.20E-5 | 5.76E-5 | 1.13E-5 | 2.56E-3 | 5.40E-6 |
|  | SSIM | 4.03E-3 | 9.94E-1 | 9.52E-1 | 9.99E-1 | 9.98E-1 | 9.99E-1 | 9.77E-1 | 9.98E-1 |

Table 23: CIFAR10 Dataset with Trigger: Stop Sign, target: Corner, and inference-time clipping.

| Sampler | P.R.
Metric | 0% | 5% | 10% | 20% | 30% | 50% | 70% | 90% |
|---|---|---|---|---|---|---|---|---|---|
| ANCESTRAL | FID | 14.31 | 8.54 | 7.80 | 8.49 | 8.08 | 8.17 | 7.89 | 7.91 |
| | MSE | 7.93E-2 | 7.56E-2 | 7.48E-2 | 7.54E-2 | 7.32E-2 | 6.97E-2 | 6.83E-2 | 6.21E-2 |
| | SSIM | 7.10E-2 | 8.99E-2 | 1.03E-1 | 8.87E-2 | 9.95E-2 | 1.13E-1 | 1.08E-1 | 1.44E-1 |
| UNIPC | FID | 185.20 | 121.16 | 119.11 | 121.15 | 118.63 | 118.40 | 124.84 | 124.46 |
| | MSE | 1.11E-1 | 1.19E-2 | 1.07E-2 | 1.89E-3 | 2.14E-2 | 2.65E-2 | 2.36E-3 | 4.50E-3 |
| | SSIM | 8.50E-3 | 6.40E-1 | 6.97E-1 | 8.36E-1 | 5.53E-1 | 5.21E-1 | 8.73E-1 | 8.40E-1 |
| DPM. O2 | FID | 185.20 | 121.16 | 119.11 | 121.15 | 118.63 | 118.40 | 124.84 | 124.46 |
| | MSE | 1.11E-1 | 1.19E-2 | 1.07E-2 | 1.89E-3 | 2.14E-2 | 2.65E-2 | 2.36E-3 | 4.50E-3 |
| | SSIM | 8.50E-3 | 6.40E-1 | 6.97E-1 | 8.36E-1 | 5.53E-1 | 5.21E-1 | 8.73E-1 | 8.40E-1 |
| DPM. O3 | FID | 185.20 | 121.16 | 119.11 | 121.15 | 118.63 | 118.40 | 124.84 | 124.46 |
| | MSE | 1.11E-1 | 1.19E-2 | 1.07E-2 | 1.89E-3 | 2.14E-2 | 2.65E-2 | 2.36E-3 | 4.50E-3 |
| | SSIM | 8.50E-3 | 6.40E-1 | 6.97E-1 | 8.36E-1 | 5.53E-1 | 5.21E-1 | 8.73E-1 | 8.40E-1 |
| DPM++. O2 | FID | 185.20 | 121.16 | 119.11 | 121.15 | 118.63 | 118.40 | 124.84 | 124.46 |
| | MSE | 1.11E-1 | 1.19E-2 | 1.07E-2 | 1.89E-3 | 2.14E-2 | 2.65E-2 | 2.36E-3 | 4.50E-3 |
| | SSIM | 8.50E-3 | 6.40E-1 | 6.97E-1 | 8.36E-1 | 5.53E-1 | 5.21E-1 | 8.73E-1 | 8.40E-1 |
| DPM++. O3 | FID | 185.20 | 121.16 | 119.11 | 121.15 | 118.63 | 82.22 | 124.84 | 124.46 |
| | MSE | 1.11E-1 | 1.19E-2 | 1.07E-2 | 1.89E-3 | 2.14E-2 | 2.42E-2 | 2.36E-3 | 4.50E-3 |
| | SSIM | 8.50E-3 | 6.40E-1 | 6.97E-1 | 8.36E-1 | 5.53E-1 | 5.78E-1 | 8.73E-1 | 8.40E-1 |
| DEIS | FID | 185.20 | 121.16 | 119.11 | 121.15 | 118.63 | 118.40 | 124.84 | 124.46 |
| | MSE | 1.11E-1 | 1.19E-2 | 1.07E-2 | 1.89E-3 | 2.14E-2 | 2.65E-2 | 2.36E-3 | 4.50E-3 |
| | SSIM | 8.50E-3 | 6.40E-1 | 6.97E-1 | 8.36E-1 | 5.53E-1 | 5.21E-1 | 8.73E-1 | 8.40E-1 |
| PNDM | FID | 177.35 | 86.72 | 124.80 | 127.35 | 122.77 | 122.23 | 89.46 | 89.20 |
| | MSE | 1.10E-1 | 5.75E-2 | 2.91E-2 | 3.73E-3 | 4.75E-2 | 4.18E-2 | 1.01E-3 | 2.17E-4 |
| | SSIM | 6.74E-3 | 4.26E-1 | 7.13E-1 | 9.62E-1 | 5.39E-1 | 5.68E-1 | 9.78E-1 | 9.85E-1 |
| HEUN | FID | 177.35 | 129.05 | 124.80 | 127.35 | 122.77 | 122.23 | 129.63 | 133.33 |
| | MSE | 1.10E-1 | 4.46E-2 | 2.91E-2 | 3.73E-3 | 4.75E-2 | 4.18E-2 | 1.08E-3 | 2.11E-4 |
| | SSIM | 6.74E-3 | 5.72E-1 | 7.13E-1 | 9.62E-1 | 5.39E-1 | 5.68E-1 | 9.83E-1 | 9.91E-1 |
| LMSD | FID | 177.35 | 129.05 | 124.80 | 127.35 | 122.77 | 122.23 | 129.63 | 133.33 |
| | MSE | 1.10E-1 | 4.46E-2 | 2.91E-2 | 3.73E-3 | 4.75E-2 | 4.18E-2 | 1.08E-3 | 2.11E-4 |
| | SSIM | 6.74E-3 | 5.72E-1 | 7.13E-1 | 9.62E-1 | 5.39E-1 | 5.68E-1 | 9.83E-1 | 9.91E-1 |

Table 24: CIFAR10 Dataset with Trigger: Stop Sign, target: Shoe, and inference-time clipping.

| Sampler | P.R.
Metric | 0% | 5% | 10% | 20% | 30% | 50% | 70% | 90% |
|---|---|---|---|---|---|---|---|---|---|
| UNIPC | FID | 185.20 | 107.88 | 108.12 | 109.60 | 105.08 | 109.91 | 104.87 | 98.41 |
| | MSE | 3.20E-1 | 3.56E-2 | 6.73E-3 | 1.27E-3 | 5.94E-3 | 7.13E-4 | 1.28E-3 | 1.94E-3 |
| | SSIM | 6.07E-3 | 8.50E-1 | 9.50E-1 | 9.72E-1 | 9.60E-1 | 9.83E-1 | 9.77E-1 | 9.73E-1 |
| DPM. O2 | FID | 185.20 | 107.88 | 108.12 | 109.60 | 105.08 | 109.91 | 104.87 | 98.41 |
| | MSE | 3.20E-1 | 3.56E-2 | 6.73E-3 | 1.27E-3 | 5.94E-3 | 7.13E-4 | 1.28E-3 | 1.94E-3 |
| | SSIM | 6.07E-3 | 8.50E-1 | 9.50E-1 | 9.72E-1 | 9.60E-1 | 9.83E-1 | 9.77E-1 | 9.73E-1 |
| DPM. O3 | FID | 185.20 | 107.88 | 108.12 | 109.60 | 105.08 | 109.91 | 73.51 | 98.41 |
| | MSE | 3.20E-1 | 3.56E-2 | 6.73E-3 | 1.27E-3 | 5.94E-3 | 7.13E-4 | 8.89E-4 | 1.94E-3 |
| | SSIM | 6.07E-3 | 8.50E-1 | 9.50E-1 | 9.72E-1 | 9.60E-1 | 9.83E-1 | 9.81E-1 | 9.73E-1 |
| DPM++. O2 | FID | 185.20 | 107.88 | 108.12 | 109.60 | 105.08 | 109.91 | 104.87 | 98.41 |
| | MSE | 3.20E-1 | 3.56E-2 | 6.73E-3 | 1.27E-3 | 5.94E-3 | 7.13E-4 | 1.28E-3 | 1.94E-3 |
| | SSIM | 6.07E-3 | 8.50E-1 | 9.50E-1 | 9.72E-1 | 9.60E-1 | 9.83E-1 | 9.77E-1 | 9.73E-1 |
| DPM++. O3 | FID | 185.20 | 107.88 | 108.12 | 109.60 | 105.08 | 109.91 | 104.87 | 98.41 |
| | MSE | 3.20E-1 | 3.56E-2 | 6.73E-3 | 1.27E-3 | 5.94E-3 | 7.13E-4 | 1.28E-3 | 1.94E-3 |
| | SSIM | 6.07E-3 | 8.50E-1 | 9.50E-1 | 9.72E-1 | 9.60E-1 | 9.83E-1 | 9.77E-1 | 9.73E-1 |
| DEIS | FID | 185.20 | 107.88 | 108.12 | 109.60 | 105.08 | 109.91 | 104.87 | 98.41 |
| | MSE | 3.20E-1 | 3.56E-2 | 6.73E-3 | 1.27E-3 | 5.94E-3 | 7.13E-4 | 1.28E-3 | 1.94E-3 |
| | SSIM | 6.07E-3 | 8.50E-1 | 9.50E-1 | 9.72E-1 | 9.60E-1 | 9.83E-1 | 9.77E-1 | 9.73E-1 |
| PNDM | FID | 177.35 | 74.67 | 107.85 | 73.88 | 71.59 | 72.62 | 105.62 | 67.34 |
| | MSE | 3.24E-1 | 1.05E-1 | 1.97E-2 | 1.70E-3 | 1.17E-2 | 3.98E-5 | 1.53E-5 | 1.59E-5 |
| | SSIM | 4.67E-3 | 6.26E-1 | 9.32E-1 | 9.91E-1 | 9.53E-1 | 9.97E-1 | 9.99E-1 | 9.97E-1 |
| HEUN | FID | 177.35 | 110.54 | 107.85 | 108.84 | 103.94 | 107.88 | 105.62 | 96.78 |
| | MSE | 3.24E-1 | 8.27E-2 | 1.97E-2 | 1.86E-3 | 1.31E-2 | 4.19E-5 | 1.53E-5 | 9.93E-6 |
| | SSIM | 4.67E-3 | 7.25E-1 | 9.32E-1 | 9.92E-1 | 9.49E-1 | 9.99E-1 | 9.99E-1 | 9.99E-1 |
| LMSD | FID | 177.35 | 110.54 | 107.85 | 108.84 | 103.94 | 107.88 | 105.62 | 96.78 |
| | MSE | 3.24E-1 | 8.27E-2 | 1.97E-2 | 1.86E-3 | 1.31E-2 | 4.19E-5 | 1.53E-5 | 9.93E-6 |
| | SSIM | 4.67E-3 | 7.25E-1 | 9.32E-1 | 9.92E-1 | 9.49E-1 | 9.99E-1 | 9.99E-1 | 9.99E-1 |

Table 25: CIFAR10 Dataset with Trigger: Stop Sign, target: Hat, and inference-time clipping.

| Sampler | P.R. Metric | 0% | 5% | 10% | 20% | 30% | 50% | 70% | 90% |
|---|---|---|---|---|---|---|---|---|---|
| ANCESTRAL | FID | 14.31 | 8.31 | 7.53 | 8.10 | 7.64 | 7.63 | 7.63 | 7.71 |
| | MSE | 1.76E-1 | 1.67E-1 | 1.66E-1 | 1.68E-1 | 1.67E-1 | 1.62E-1 | 1.58E-1 | 1.48E-1 |
| | SSIM | 3.41E-2 | 4.26E-2 | 4.36E-2 | 4.05E-2 | 4.37E-2 | 4.70E-2 | 4.72E-2 | 5.16E-2 |
| UNIPC | FID | 185.20 | 110.69 | 109.15 | 110.76 | 107.63 | 110.05 | 106.94 | 100.60 |
| | MSE | 2.33E-1 | 8.32E-3 | 6.79E-3 | 5.52E-3 | 6.54E-3 | 1.88E-3 | 3.30E-3 | 5.54E-3 |
| | SSIM | 8.04E-3 | 9.06E-1 | 9.45E-1 | 9.61E-1 | 9.55E-1 | 9.84E-1 | 9.80E-1 | 9.73E-1 |
| DPM. O2 | FID | 185.20 | 110.69 | 109.15 | 110.76 | 107.63 | 110.05 | 106.94 | 100.60 |
| | MSE | 2.33E-1 | 8.32E-3 | 6.79E-3 | 5.52E-3 | 6.54E-3 | 1.88E-3 | 3.30E-3 | 5.54E-3 |
| | SSIM | 8.04E-3 | 9.06E-1 | 9.45E-1 | 9.61E-1 | 9.55E-1 | 9.84E-1 | 9.80E-1 | 9.73E-1 |
| DPM. O3 | FID | 185.20 | 110.69 | 109.15 | 110.76 | 107.63 | 110.05 | 106.94 | 100.60 |
| | MSE | 2.33E-1 | 8.32E-3 | 6.79E-3 | 5.52E-3 | 6.54E-3 | 1.88E-3 | 3.30E-3 | 5.54E-3 |
| | SSIM | 8.04E-3 | 9.06E-1 | 9.45E-1 | 9.61E-1 | 9.55E-1 | 9.84E-1 | 9.80E-1 | 9.73E-1 |
| DPM++. O2 | FID | 185.20 | 110.69 | 109.15 | 110.76 | 107.63 | 110.05 | 106.94 | 100.60 |
| | MSE | 2.33E-1 | 8.32E-3 | 6.79E-3 | 5.52E-3 | 6.54E-3 | 1.88E-3 | 3.30E-3 | 5.54E-3 |
| | SSIM | 8.04E-3 | 9.06E-1 | 9.45E-1 | 9.61E-1 | 9.55E-1 | 9.84E-1 | 9.80E-1 | 9.73E-1 |
| DPM++. O3 | FID | 185.20 | 110.69 | 109.15 | 110.76 | 107.63 | 110.05 | 106.94 | 100.60 |
| | MSE | 2.33E-1 | 8.32E-3 | 6.79E-3 | 5.52E-3 | 6.54E-3 | 1.88E-3 | 3.30E-3 | 5.54E-3 |
| | SSIM | 8.04E-3 | 9.06E-1 | 9.45E-1 | 9.61E-1 | 9.55E-1 | 9.84E-1 | 9.80E-1 | 9.73E-1 |
| DEIS | FID | 185.20 | 110.69 | 109.15 | 110.76 | 107.63 | 110.05 | 106.94 | 100.60 |
| | MSE | 2.33E-1 | 8.32E-3 | 6.79E-3 | 5.52E-3 | 6.54E-3 | 1.88E-3 | 3.30E-3 | 5.54E-3 |
| | SSIM | 8.04E-3 | 9.06E-1 | 9.45E-1 | 9.61E-1 | 9.55E-1 | 9.84E-1 | 9.80E-1 | 9.73E-1 |
| PNDM | FID | 177.35 | 77.16 | 42.34 | 75.69 | 42.47 | 106.99 | 8.11 | 95.57 |
| | MSE | 2.34E-1 | 9.55E-3 | 3.38E-2 | 3.55E-3 | 1.37E-4 | 8.03E-4 | 3.04E-5 | 6.80E-6 |
| | SSIM | 7.73E-3 | 9.49E-1 | 8.26E-1 | 9.82E-1 | 9.98E-1 | 9.95E-1 | 9.98E-1 | 1.00E+0 |
| HEUN | FID | 177.35 | 111.49 | 110.30 | 110.51 | 108.51 | 106.99 | 99.88 | 95.57 |
| | MSE | 2.34E-1 | 7.92E-3 | 1.04E-2 | 4.58E-3 | 9.40E-3 | 8.03E-4 | 1.18E-5 | 6.80E-6 |
| | SSIM | 7.73E-3 | 9.63E-1 | 9.52E-1 | 9.77E-1 | 9.54E-1 | 9.95E-1 | 1.00E+0 | 1.00E+0 |
| LMSD | FID | 177.35 | 111.49 | 110.30 | 110.51 | 108.51 | 106.99 | 99.88 | 95.57 |
| | MSE | 2.34E-1 | 7.92E-3 | 1.04E-2 | 4.58E-3 | 9.40E-3 | 8.03E-4 | 1.18E-5 | 6.80E-6 |
| | SSIM | 7.73E-3 | 9.63E-1 | 9.52E-1 | 9.77E-1 | 9.54E-1 | 9.95E-1 | 1.00E+0 | 1.00E+0 |

Table 26: CIFAR10 Dataset with Trigger: Grey Box, target: No Shift, and inference-time clipping.

| Sampler | P.R. Metric | 0% | 5% | 10% | 20% | 30% | 50% | 70% | 90% |
|---|---|---|---|---|---|---|---|---|---|
| UNIPC | FID | 185.20 | 121.50 | 121.93 | 130.83 | 123.83 | 134.04 | 146.21 | 159.08 |
| | MSE | 1.20E-1 | 1.18E-3 | 4.93E-4 | 4.28E-4 | 1.05E-3 | 1.23E-3 | 1.64E-3 | 3.83E-3 |
| | SSIM | 8.51E-4 | 9.68E-1 | 9.76E-1 | 9.78E-1 | 9.62E-1 | 9.62E-1 | 9.62E-1 | 9.31E-1 |
| DPM. O2 | FID | 185.20 | 121.50 | 121.93 | 130.83 | 123.83 | 134.04 | 146.21 | 159.08 |
| | MSE | 1.20E-1 | 1.18E-3 | 4.93E-4 | 4.28E-4 | 1.05E-3 | 1.23E-3 | 1.64E-3 | 3.83E-3 |
| | SSIM | 8.51E-4 | 9.68E-1 | 9.76E-1 | 9.78E-1 | 9.62E-1 | 9.62E-1 | 9.62E-1 | 9.31E-1 |
| DPM. O3 | FID | 185.20 | 121.50 | 84.27 | 130.83 | 123.83 | 134.04 | 146.21 | 159.08 |
| | MSE | 1.20E-1 | 1.18E-3 | 1.13E-3 | 4.28E-4 | 1.05E-3 | 1.23E-3 | 1.64E-3 | 3.83E-3 |
| | SSIM | 8.51E-4 | 9.68E-1 | 9.55E-1 | 9.78E-1 | 9.62E-1 | 9.62E-1 | 9.62E-1 | 9.31E-1 |
| DPM++. O2 | FID | 185.20 | 121.50 | 121.93 | 130.83 | 123.83 | 134.04 | 146.21 | 159.08 |
| | MSE | 1.20E-1 | 1.18E-3 | 4.93E-4 | 4.28E-4 | 1.05E-3 | 1.23E-3 | 1.64E-3 | 3.83E-3 |
| | SSIM | 8.51E-4 | 9.68E-1 | 9.76E-1 | 9.78E-1 | 9.62E-1 | 9.62E-1 | 9.62E-1 | 9.31E-1 |
| DPM++. O3 | FID | 185.20 | 121.50 | 121.93 | 130.83 | 123.83 | 134.04 | 146.21 | 159.08 |
| | MSE | 1.20E-1 | 1.18E-3 | 4.93E-4 | 4.28E-4 | 1.05E-3 | 1.23E-3 | 1.64E-3 | 3.83E-3 |
| | SSIM | 8.51E-4 | 9.68E-1 | 9.76E-1 | 9.78E-1 | 9.62E-1 | 9.62E-1 | 9.62E-1 | 9.31E-1 |
| DEIS | FID | 185.20 | 121.50 | 121.93 | 130.83 | 123.83 | 134.04 | 146.21 | 159.08 |
| | MSE | 1.20E-1 | 1.18E-3 | 4.93E-4 | 4.28E-4 | 1.05E-3 | 1.23E-3 | 1.64E-3 | 3.83E-3 |
| | SSIM | 8.51E-4 | 9.68E-1 | 9.76E-1 | 9.78E-1 | 9.62E-1 | 9.62E-1 | 9.62E-1 | 9.31E-1 |
| PNDM | FID | 177.35 | 84.49 | 8.34 | 54.96 | 91.59 | 98.41 | 154.06 | 75.09 |
| | MSE | 1.20E-1 | 1.80E-2 | 1.37E-2 | 2.36E-4 | 4.39E-5 | 1.61E-5 | 7.00E-6 | 2.16E-5 |
| | SSIM | 9.50E-4 | 7.88E-1 | 7.93E-1 | 9.80E-1 | 9.90E-1 | 9.91E-1 | 9.96E-1 | 9.85E-1 |
| HEUN | FID | 177.35 | 126.85 | 127.51 | 137.93 | 129.76 | 143.02 | 154.06 | 162.09 |
| | MSE | 1.20E-1 | 2.98E-3 | 1.27E-3 | 6.22E-5 | 3.67E-5 | 8.34E-6 | 7.00E-6 | 7.66E-6 |
| | SSIM | 9.50E-4 | 9.72E-1 | 9.85E-1 | 9.96E-1 | 9.96E-1 | 9.96E-1 | 9.96E-1 | 9.95E-1 |
| LMSD | FID | 177.35 | 126.85 | 127.51 | 137.93 | 129.76 | 143.02 | 154.06 | 162.09 |
| | MSE | 1.20E-1 | 2.98E-3 | 1.27E-3 | 6.22E-5 | 3.67E-5 | 8.34E-6 | 7.00E-6 | 7.66E-6 |
| | SSIM | 9.50E-4 | 9.72E-1 | 9.85E-1 | 9.96E-1 | 9.96E-1 | 9.96E-1 | 9.96E-1 | 9.95E-1 |

Table 27: CIFAR10 Dataset with Trigger: Grey Box, target: Shift, and inference-time clipping.

| Sampler | P.R. Metric | 0% | 5% | 10% | 20% | 30% | 50% | 70% | 90% |
|---|---|---|---|---|---|---|---|---|---|
| UNIPC | FID | 185.20 | 123.67 | 117.55 | 118.87 | 119.92 | 122.32 | 126.07 | 131.79 |
| | MSE | 1.20E-1 | 2.83E-3 | 1.61E-3 | 4.57E-4 | 2.76E-4 | 1.82E-3 | 2.07E-3 | 4.21E-3 |
| | SSIM | 7.09E-4 | 9.50E-1 | 9.60E-1 | 9.76E-1 | 9.84E-1 | 9.54E-1 | 9.55E-1 | 9.27E-1 |
| DPM. O2 | FID | 185.20 | 123.67 | 117.55 | 118.87 | 119.92 | 122.32 | 126.07 | 131.79 |
| | MSE | 1.20E-1 | 2.83E-3 | 1.61E-3 | 4.57E-4 | 2.76E-4 | 1.82E-3 | 2.07E-3 | 4.21E-3 |
| | SSIM | 7.09E-4 | 9.50E-1 | 9.60E-1 | 9.76E-1 | 9.84E-1 | 9.54E-1 | 9.55E-1 | 9.27E-1 |
| DPM. O3 | FID | 185.20 | 123.67 | 117.55 | 118.87 | 119.92 | 122.32 | 126.07 | 131.79 |
| | MSE | 1.20E-1 | 2.83E-3 | 1.61E-3 | 4.57E-4 | 2.76E-4 | 1.82E-3 | 2.07E-3 | 4.21E-3 |
| | SSIM | 7.09E-4 | 9.50E-1 | 9.60E-1 | 9.76E-1 | 9.84E-1 | 9.54E-1 | 9.55E-1 | 9.27E-1 |
| DPM++. O2 | FID | 185.20 | 123.67 | 117.55 | 118.87 | 119.92 | 122.32 | 126.07 | 131.79 |
| | MSE | 1.20E-1 | 2.83E-3 | 1.61E-3 | 4.57E-4 | 2.76E-4 | 1.82E-3 | 2.07E-3 | 4.21E-3 |
| | SSIM | 7.09E-4 | 9.50E-1 | 9.60E-1 | 9.76E-1 | 9.84E-1 | 9.54E-1 | 9.55E-1 | 9.27E-1 |
| DPM++. O3 | FID | 185.20 | 123.67 | 80.34 | 118.87 | 119.92 | 122.32 | 126.07 | 131.79 |
| | MSE | 1.20E-1 | 2.83E-3 | 1.07E-2 | 4.57E-4 | 2.76E-4 | 1.82E-3 | 2.07E-3 | 4.21E-3 |
| | SSIM | 7.09E-4 | 9.50E-1 | 8.34E-1 | 9.76E-1 | 9.84E-1 | 9.54E-1 | 9.55E-1 | 9.27E-1 |
| DEIS | FID | 185.20 | 123.67 | 117.55 | 118.87 | 119.92 | 122.32 | 126.07 | 131.79 |
| | MSE | 1.20E-1 | 2.83E-3 | 1.61E-3 | 4.57E-4 | 2.76E-4 | 1.82E-3 | 2.07E-3 | 4.21E-3 |
| | SSIM | 7.09E-4 | 9.50E-1 | 9.60E-1 | 9.76E-1 | 9.84E-1 | 9.54E-1 | 9.55E-1 | 9.27E-1 |
| PNDM | FID | 177.35 | 88.59 | 81.09 | 47.93 | 86.19 | 88.97 | 129.77 | 102.72 |
| | MSE | 1.20E-1 | 1.71E-2 | 3.19E-3 | 2.15E-4 | 3.71E-4 | 1.64E-5 | 6.98E-6 | 1.42E-5 |
| | SSIM | 6.17E-4 | 8.09E-1 | 9.57E-1 | 9.86E-1 | 9.90E-1 | 9.94E-1 | 9.97E-1 | 9.93E-1 |
| HEUN | FID | 177.35 | 125.94 | 119.25 | 121.26 | 122.38 | 125.11 | 129.77 | 138.51 |
| | MSE | 1.20E-1 | 8.17E-3 | 2.87E-3 | 2.09E-4 | 3.83E-4 | 7.62E-6 | 6.98E-6 | 7.41E-6 |
| | SSIM | 6.17E-4 | 9.27E-1 | 9.73E-1 | 9.96E-1 | 9.94E-1 | 9.98E-1 | 9.97E-1 | 9.97E-1 |
| LMSD | FID | 177.35 | 125.94 | 119.25 | 121.26 | 122.38 | 125.11 | 129.77 | 138.51 |
| | MSE | 1.20E-1 | 8.17E-3 | 2.87E-3 | 2.09E-4 | 3.83E-4 | 7.62E-6 | 6.98E-6 | 7.41E-6 |
| | SSIM | 6.17E-4 | 9.27E-1 | 9.73E-1 | 9.96E-1 | 9.94E-1 | 9.98E-1 | 9.97E-1 | 9.97E-1 |

Table 28: CIFAR10 Dataset with Trigger: Grey Box, target: Corner, and inference-time clipping.

| Sampler | P.R. Metric | 0% | 5% | 10% | 20% | 30% | 50% | 70% | 90% |
|---|---|---|---|---|---|---|---|---|---|
| | FID | 14.31 | 9.91 | 10.94 | 12.99 | 15.06 | 19.85 | 28.11 | 53.35 |
| ANCESTRAL | MSE | 7.86E-2 | 5.56E-2 | 5.34E-2 | 4.97E-2 | 5.01E-2 | 3.87E-2 | 2.74E-2 | 1.32E-2 |
| | SSIM | 7.17E-2 | 2.50E-1 | 2.80E-1 | 3.29E-1 | 3.35E-1 | 4.60E-1 | 5.88E-1 | 7.73E-1 |
| | FID | 185.20 | 125.95 | 127.27 | 124.84 | 125.84 | 131.79 | 134.52 | 136.83 |
| UNIPC | MSE | 1.05E-1 | 2.18E-3 | 8.14E-4 | 5.05E-4 | 4.90E-4 | 1.09E-3 | 1.69E-3 | 5.51E-3 |
| | SSIM | 1.13E-3 | 9.16E-1 | 9.33E-1 | 9.48E-1 | 9.54E-1 | 9.36E-1 | 9.24E-1 | 8.36E-1 |
| | FID | 185.20 | 125.95 | 127.27 | 124.84 | 125.84 | 131.79 | 134.52 | 136.83 |
| DPM. O2 | MSE | 1.05E-1 | 2.18E-3 | 8.14E-4 | 5.05E-4 | 4.90E-4 | 1.09E-3 | 1.69E-3 | 5.51E-3 |
| | SSIM | 1.13E-3 | 9.16E-1 | 9.33E-1 | 9.48E-1 | 9.54E-1 | 9.36E-1 | 9.24E-1 | 8.36E-1 |
| | FID | 185.20 | 125.95 | 127.27 | 124.84 | 125.84 | 131.79 | 134.52 | 136.83 |
| DPM. O3 | MSE | 1.05E-1 | 2.18E-3 | 8.14E-4 | 5.05E-4 | 4.90E-4 | 1.09E-3 | 1.69E-3 | 5.51E-3 |
| | SSIM | 1.13E-3 | 9.16E-1 | 9.33E-1 | 9.48E-1 | 9.54E-1 | 9.36E-1 | 9.24E-1 | 8.36E-1 |
| | FID | 185.20 | 125.95 | 127.27 | 124.84 | 125.84 | 131.79 | 134.52 | 136.83 |
| DPM++. O2 | MSE | 1.05E-1 | 2.18E-3 | 8.14E-4 | 5.05E-4 | 4.90E-4 | 1.09E-3 | 1.69E-3 | 5.51E-3 |
| | SSIM | 1.13E-3 | 9.16E-1 | 9.33E-1 | 9.48E-1 | 9.54E-1 | 9.36E-1 | 9.24E-1 | 8.36E-1 |
| | FID | 185.20 | 125.95 | 127.27 | 124.84 | 125.84 | 131.79 | 134.52 | 136.83 |
| DPM++. O3 | MSE | 1.05E-1 | 2.18E-3 | 8.14E-4 | 5.05E-4 | 4.90E-4 | 1.09E-3 | 1.69E-3 | 5.51E-3 |
| | SSIM | 1.13E-3 | 9.16E-1 | 9.33E-1 | 9.48E-1 | 9.54E-1 | 9.36E-1 | 9.24E-1 | 8.36E-1 |
| | FID | 185.20 | 125.95 | 127.27 | 124.84 | 125.84 | 131.79 | 134.52 | 136.83 |
| DEIS | MSE | 1.05E-1 | 2.18E-3 | 8.14E-4 | 5.05E-4 | 4.90E-4 | 1.09E-3 | 1.69E-3 | 5.51E-3 |
| | SSIM | 1.13E-3 | 9.16E-1 | 9.33E-1 | 9.48E-1 | 9.54E-1 | 9.36E-1 | 9.24E-1 | 8.36E-1 |
| | FID | 177.35 | 134.07 | 93.84 | 49.66 | 94.22 | 102.26 | 147.95 | 156.45 |
| PNDM | MSE | 1.05E-1 | 3.56E-3 | 4.82E-3 | 9.19E-5 | 3.88E-5 | 2.01E-5 | 9.18E-6 | 7.25E-6 |
| | SSIM | 1.38E-3 | 9.66E-1 | 8.97E-1 | 9.78E-1 | 9.87E-1 | 9.88E-1 | 9.94E-1 | 9.94E-1 |
| | FID | 177.35 | 134.07 | 135.99 | 134.72 | 137.73 | 145.55 | 147.95 | 156.45 |
| HEUN | MSE | 1.05E-1 | 3.56E-3 | 4.92E-4 | 4.11E-5 | 5.36E-5 | 1.44E-5 | 9.18E-6 | 7.25E-6 |
| | SSIM | 1.38E-3 | 9.66E-1 | 9.92E-1 | 9.96E-1 | 9.95E-1 | 9.94E-1 | 9.94E-1 | 9.94E-1 |
| | FID | 177.35 | 134.07 | 135.99 | 134.72 | 137.73 | 145.55 | 147.95 | 156.45 |
| LMSD | MSE | 1.05E-1 | 3.56E-3 | 4.92E-4 | 4.11E-5 | 5.36E-5 | 1.44E-5 | 9.18E-6 | 7.25E-6 |
| | SSIM | 1.38E-3 | 9.66E-1 | 9.92E-1 | 9.96E-1 | 9.95E-1 | 9.94E-1 | 9.94E-1 | 9.94E-1 |

Table 29: CIFAR10 Dataset with Trigger: Grey Box, target: Shoe, and inference-time clipping.

| Sampler | P.R. Metric | 0% | 5% | 10% | 20% | 30% | 50% | 70% | 90% |
|---|---|---|---|---|---|---|---|---|---|
| UNIPC | FID | 185.20 | 108.96 | 109.27 | 113.59 | 110.68 | 111.37 | 105.23 | 105.27 |
| | MSE | 3.37E-1 | 2.20E-3 | 1.02E-3 | 8.76E-4 | 9.58E-4 | 1.80E-3 | 1.97E-3 | 3.59E-3 |
| | SSIM | 2.29E-4 | 9.80E-1 | 9.82E-1 | 9.82E-1 | 9.83E-1 | 9.76E-1 | 9.76E-1 | 9.65E-1 |
| DPM. O2 | FID | 185.20 | 108.96 | 109.27 | 113.59 | 110.68 | 111.37 | 105.23 | 105.27 |
| | MSE | 3.37E-1 | 2.20E-3 | 1.02E-3 | 8.76E-4 | 9.58E-4 | 1.80E-3 | 1.97E-3 | 3.59E-3 |
| | SSIM | 2.29E-4 | 9.80E-1 | 9.82E-1 | 9.82E-1 | 9.83E-1 | 9.76E-1 | 9.76E-1 | 9.65E-1 |
| DPM. O3 | FID | 185.20 | 108.96 | 109.27 | 113.59 | 110.68 | 111.37 | 105.23 | 105.27 |
| | MSE | 3.37E-1 | 2.20E-3 | 1.02E-3 | 8.76E-4 | 9.58E-4 | 1.80E-3 | 1.97E-3 | 3.59E-3 |
| | SSIM | 2.29E-4 | 9.80E-1 | 9.82E-1 | 9.82E-1 | 9.83E-1 | 9.76E-1 | 9.76E-1 | 9.65E-1 |
| DPM++. O2 | FID | 185.20 | 75.06 | 109.27 | 113.59 | 110.68 | 111.37 | 105.23 | 105.27 |
| | MSE | 3.37E-1 | 1.96E-2 | 1.02E-3 | 8.76E-4 | 9.58E-4 | 1.80E-3 | 1.97E-3 | 3.59E-3 |
| | SSIM | 2.29E-4 | 8.92E-1 | 9.82E-1 | 9.82E-1 | 9.83E-1 | 9.76E-1 | 9.76E-1 | 9.65E-1 |
| DPM++. O3 | FID | 185.20 | 108.96 | 109.27 | 113.59 | 110.68 | 111.37 | 105.23 | 105.27 |
| | MSE | 3.37E-1 | 2.20E-3 | 1.02E-3 | 8.76E-4 | 9.58E-4 | 1.80E-3 | 1.97E-3 | 3.59E-3 |
| | SSIM | 2.29E-4 | 9.80E-1 | 9.82E-1 | 9.82E-1 | 9.83E-1 | 9.76E-1 | 9.76E-1 | 9.65E-1 |
| DEIS | FID | 185.20 | 108.96 | 109.27 | 113.59 | 110.68 | 111.37 | 105.23 | 105.27 |
| | MSE | 3.37E-1 | 2.20E-3 | 1.02E-3 | 8.76E-4 | 9.58E-4 | 1.80E-3 | 1.97E-3 | 3.59E-3 |
| | SSIM | 2.29E-4 | 9.80E-1 | 9.82E-1 | 9.82E-1 | 9.83E-1 | 9.76E-1 | 9.76E-1 | 9.65E-1 |
| PNDM | FID | 177.35 | 76.12 | 77.07 | 44.45 | 74.86 | 76.19 | 73.59 | 73.31 |
| | MSE | 3.37E-1 | 6.60E-2 | 4.31E-3 | 8.33E-5 | 5.15E-5 | 2.28E-5 | 1.95E-5 | 1.49E-5 |
| | SSIM | 2.08E-4 | 7.43E-1 | 9.75E-1 | 9.95E-1 | 9.97E-1 | 9.97E-1 | 9.98E-1 | 9.97E-1 |
| HEUN | FID | 177.35 | 109.16 | 111.45 | 112.72 | 109.41 | 108.98 | 104.22 | 105.28 |
| | MSE | 3.37E-1 | 4.08E-3 | 8.50E-4 | 1.14E-4 | 5.27E-5 | 1.76E-5 | 1.19E-5 | 8.16E-6 |
| | SSIM | 2.08E-4 | 9.87E-1 | 9.96E-1 | 9.98E-1 | 9.99E-1 | 9.99E-1 | 9.99E-1 | 9.99E-1 |
| LMSD | FID | 177.35 | 109.16 | 111.45 | 112.72 | 109.41 | 108.98 | 104.22 | 105.28 |
| | MSE | 3.37E-1 | 4.08E-3 | 8.50E-4 | 1.14E-4 | 5.27E-5 | 1.76E-5 | 1.19E-5 | 8.16E-6 |
| | SSIM | 2.08E-4 | 9.87E-1 | 9.96E-1 | 9.98E-1 | 9.99E-1 | 9.99E-1 | 9.99E-1 | 9.99E-1 |

Table 30: CIFAR10 Dataset with Trigger: Grey Box, target: Hat, and inference-time clipping.

| Sampler | P.R. Metric | 0% | 5% | 10% | 20% | 30% | 50% | 70% | 90% |
|---|---|---|---|---|---|---|---|---|---|
| ANCESTRAL | FID | 14.31 | 8.42 | 8.82 | 8.89 | 8.97 | 10.11 | 11.32 | 17.82 |
| | MSE | 1.74E-1 | 1.24E-1 | 1.08E-1 | 1.09E-1 | 1.12E-1 | 1.01E-1 | 9.63E-2 | 8.57E-2 |
| | SSIM | 3.43E-2 | 2.08E-1 | 2.83E-1 | 2.82E-1 | 2.66E-1 | 3.26E-1 | 3.55E-1 | 4.07E-1 |
| UNIPC | FID | 185.20 | 110.41 | 111.65 | 109.41 | 112.08 | 111.42 | 105.79 | 107.68 |
| | MSE | 2.40E-1 | 5.12E-4 | 3.63E-4 | 5.34E-4 | 8.85E-4 | 1.63E-3 | 3.46E-3 | 6.46E-3 |
| | SSIM | 2.97E-4 | 9.88E-1 | 9.92E-1 | 9.92E-1 | 9.92E-1 | 9.88E-1 | 9.80E-1 | 9.70E-1 |
| DPM. O2 | FID | 185.20 | 110.41 | 111.65 | 109.41 | 112.08 | 111.42 | 105.79 | 107.68 |
| | MSE | 2.40E-1 | 5.12E-4 | 3.63E-4 | 5.34E-4 | 8.85E-4 | 1.63E-3 | 3.46E-3 | 6.46E-3 |
| | SSIM | 2.97E-4 | 9.88E-1 | 9.92E-1 | 9.92E-1 | 9.92E-1 | 9.88E-1 | 9.80E-1 | 9.70E-1 |
| DPM. O3 | FID | 185.20 | 110.41 | 111.65 | 109.41 | 112.08 | 111.42 | 105.79 | 107.68 |
| | MSE | 2.40E-1 | 5.12E-4 | 3.63E-4 | 5.34E-4 | 8.85E-4 | 1.63E-3 | 3.46E-3 | 6.46E-3 |
| | SSIM | 2.97E-4 | 9.88E-1 | 9.92E-1 | 9.92E-1 | 9.92E-1 | 9.88E-1 | 9.80E-1 | 9.70E-1 |
| DPM++. O2 | FID | 185.20 | 110.41 | 111.65 | 109.41 | 112.08 | 111.42 | 105.79 | 107.68 |
| | MSE | 2.40E-1 | 5.12E-4 | 3.63E-4 | 5.34E-4 | 8.85E-4 | 1.63E-3 | 3.46E-3 | 6.46E-3 |
| | SSIM | 2.97E-4 | 9.88E-1 | 9.92E-1 | 9.92E-1 | 9.92E-1 | 9.88E-1 | 9.80E-1 | 9.70E-1 |
| DPM++. O3 | FID | 185.20 | 110.41 | 111.65 | 109.41 | 112.08 | 111.42 | 105.79 | 107.68 |
| | MSE | 2.40E-1 | 5.12E-4 | 3.63E-4 | 5.34E-4 | 8.85E-4 | 1.63E-3 | 3.46E-3 | 6.46E-3 |
| | SSIM | 2.97E-4 | 9.88E-1 | 9.92E-1 | 9.92E-1 | 9.92E-1 | 9.88E-1 | 9.80E-1 | 9.70E-1 |
| DEIS | FID | 185.20 | 110.41 | 111.65 | 109.41 | 112.08 | 111.42 | 105.79 | 107.68 |
| | MSE | 2.40E-1 | 5.12E-4 | 3.63E-4 | 5.34E-4 | 8.85E-4 | 1.63E-3 | 3.46E-3 | 6.46E-3 |
| | SSIM | 2.97E-4 | 9.88E-1 | 9.92E-1 | 9.92E-1 | 9.92E-1 | 9.88E-1 | 9.80E-1 | 9.70E-1 |
| PNDM | FID | 177.35 | 112.35 | 111.44 | 77.27 | 41.78 | 77.43 | 101.54 | 102.53 |
| | MSE | 2.40E-1 | 4.60E-4 | 9.89E-5 | 7.03E-5 | 3.09E-5 | 2.03E-5 | 7.09E-6 | 6.13E-6 |
| | SSIM | 1.79E-4 | 9.97E-1 | 9.99E-1 | 9.98E-1 | 9.98E-1 | 9.99E-1 | 1.00E+0 | 1.00E+0 |
| HEUN | FID | 177.35 | 112.35 | 111.44 | 109.32 | 110.74 | 109.52 | 101.54 | 102.53 |
| | MSE | 2.40E-1 | 4.60E-4 | 9.89E-5 | 2.39E-5 | 1.41E-5 | 1.33E-5 | 7.09E-6 | 6.13E-6 |
| | SSIM | 1.79E-4 | 9.97E-1 | 9.99E-1 | 9.99E-1 | 9.99E-1 | 1.00E+0 | 1.00E+0 | 1.00E+0 |
| LMSD | FID | 177.35 | 112.35 | 111.44 | 109.32 | 110.74 | 109.52 | 101.54 | 102.53 |
| | MSE | 2.40E-1 | 4.60E-4 | 9.89E-5 | 2.39E-5 | 1.41E-5 | 1.33E-5 | 7.09E-6 | 6.13E-6 |
| | SSIM | 1.79E-4 | 9.97E-1 | 9.99E-1 | 9.99E-1 | 9.99E-1 | 1.00E+0 | 1.00E+0 | 1.00E+0 |

Table 31: DDPM performs on **Blur**, **Line**, **Box**, and **Box** with CIFAR10 Dataset, trigger: Stop Sign, and target: Shoe.

| Sampler | P.R. Metric | 0% | 10% | 20% | 30% | 50% | 70% | 90% |
|---|---|---|---|---|---|---|---|---|
| UNIPC, Blur | LPIPS | 3.25E-1 | 2.78E-1 | 2.77E-1 | 2.68E-1 | 2.56E-1 | 2.55E-1 | 2.59E-1 |
| | MSE | 2.97E-1 | 3.58E-3 | 1.31E-3 | 1.61E-3 | 4.90E-4 | 1.99E-4 | 8.39E-5 |
| | SSIM | 5.66E-2 | 9.85E-1 | 9.94E-1 | 9.93E-1 | 9.97E-1 | 9.99E-1 | 9.99E-1 |
| UNIPC, Line | LPIPS | 3.19E-1 | 2.78E-1 | 2.82E-1 | 2.76E-1 | 2.66E-1 | 2.65E-1 | 2.58E-1 |
| | MSE | 3.03E-1 | 3.78E-3 | 2.83E-3 | 3.29E-3 | 1.27E-3 | 3.37E-4 | 6.92E-5 |
| | SSIM | 4.61E-2 | 9.85E-1 | 9.89E-1 | 9.87E-1 | 9.95E-1 | 9.98E-1 | 9.99E-1 |
| UNIPC, Box | LPIPS | 3.10E-1 | 2.87E-1 | 2.85E-1 | 2.85E-1 | 2.70E-1 | 2.63E-1 | 2.63E-1 |
| | MSE | 3.34E-1 | 2.15E-2 | 3.17E-2 | 3.47E-2 | 3.29E-2 | 1.93E-2 | 1.11E-2 |
| | SSIM | 1.05E-2 | 9.10E-1 | 8.76E-1 | 8.56E-1 | 8.67E-1 | 9.19E-1 | 9.51E-1 |
| DPM. O2, Blur | LPIPS | 3.25E-1 | 2.79E-1 | 2.70E-1 | 2.65E-1 | 2.57E-1 | 2.55E-1 | 2.57E-1 |
| | MSE | 2.97E-1 | 2.98E-3 | 1.52E-3 | 1.82E-3 | 5.58E-4 | 2.93E-4 | 1.37E-4 |
| | SSIM | 5.66E-2 | 9.87E-1 | 9.93E-1 | 9.91E-1 | 9.97E-1 | 9.98E-1 | 9.99E-1 |
| DPM. O2, Line | LPIPS | 3.19E-1 | 2.88E-1 | 2.80E-1 | 2.73E-1 | 2.67E-1 | 2.63E-1 | 2.60E-1 |
| | MSE | 3.03E-1 | 3.02E-3 | 3.25E-3 | 3.19E-3 | 1.39E-3 | 3.21E-4 | 1.97E-4 |
| | SSIM | 4.61E-2 | 9.87E-1 | 9.87E-1 | 9.88E-1 | 9.94E-1 | 9.98E-1 | 9.99E-1 |
| DPM. O2, Box | LPIPS | 3.10E-1 | 2.91E-1 | 2.87E-1 | 2.83E-1 | 2.71E-1 | 2.59E-1 | 2.67E-1 |
| | MSE | 3.34E-1 | 2.07E-2 | 3.24E-2 | 3.45E-2 | 3.60E-2 | 2.14E-2 | 1.04E-2 |
| | SSIM | 1.05E-2 | 9.14E-1 | 8.74E-1 | 8.57E-1 | 8.53E-1 | 9.11E-1 | 9.50E-1 |
| DPM++. O2, Blur | LPIPS | 3.25E-1 | 2.78E-1 | 2.76E-1 | 2.68E-1 | 2.59E-1 | 2.52E-1 | 2.54E-1 |
| | MSE | 2.97E-1 | 3.56E-3 | 1.52E-3 | 1.85E-3 | 4.64E-4 | 5.16E-4 | 4.34E-6 |
| | SSIM | 5.66E-2 | 9.84E-1 | 9.93E-1 | 9.92E-1 | 9.98E-1 | 9.97E-1 | 1.00E+0 |
| DPM++. O2, Line | LPIPS | 3.19E-1 | 2.87E-1 | 2.76E-1 | 2.73E-1 | 2.67E-1 | 2.61E-1 | 2.57E-1 |
| | MSE | 3.03E-1 | 3.55E-3 | 4.38E-3 | 4.30E-3 | 1.68E-3 | 4.40E-4 | 9.60E-5 |
| | SSIM | 4.61E-2 | 9.85E-1 | 9.83E-1 | 9.83E-1 | 9.93E-1 | 9.98E-1 | 9.99E-1 |
| DPM++. O2, Box | LPIPS | 3.10E-1 | 2.87E-1 | 2.85E-1 | 2.79E-1 | 2.74E-1 | 2.65E-1 | 2.64E-1 |
| | MSE | 3.34E-1 | 2.44E-2 | 3.40E-2 | 3.12E-2 | 3.79E-2 | 1.97E-2 | 1.08E-2 |
| | SSIM | 1.05E-2 | 8.98E-1 | 8.72E-1 | 8.66E-1 | 8.45E-1 | 9.15E-1 | 9.49E-1 |
| DEIS, Blur | LPIPS | 3.25E-1 | 2.81E-1 | 2.76E-1 | 2.72E-1 | 2.60E-1 | 2.51E-1 | 2.50E-1 |
| | MSE | 2.97E-1 | 2.87E-3 | 1.34E-3 | 2.96E-3 | 2.02E-4 | 3.58E-4 | 1.61E-4 |
| | SSIM | 5.66E-2 | 9.87E-1 | 9.94E-1 | 9.88E-1 | 9.98E-1 | 9.98E-1 | 9.99E-1 |
| DEIS, Line | LPIPS | 3.19E-1 | 2.82E-1 | 2.77E-1 | 2.76E-1 | 2.71E-1 | 2.58E-1 | 2.61E-1 |
| | MSE | 3.03E-1 | 3.54E-3 | 2.50E-3 | 3.31E-3 | 1.29E-3 | 2.57E-4 | 5.03E-6 |
| | SSIM | 4.61E-2 | 9.85E-1 | 9.90E-1 | 9.86E-1 | 9.95E-1 | 9.98E-1 | 1.00E+0 |
| DEIS, Box | LPIPS | 3.10E-1 | 2.84E-1 | 2.85E-1 | 2.81E-1 | 2.70E-1 | 2.61E-1 | 2.67E-1 |
| | MSE | 3.34E-1 | 2.26E-2 | 3.26E-2 | 3.49E-2 | 3.46E-2 | 2.05E-2 | 1.03E-2 |
| | SSIM | 1.05E-2 | 9.07E-1 | 8.74E-1 | 8.52E-1 | 8.59E-1 | 9.15E-1 | 9.54E-1 |

Table 32: DDPM performs on **Blur**, **Line**, **Box**, and **Box** with CIFAR10 Dataset, trigger: Stop Sign, and target: Hat.

| Sampler | P.R. Metric | 0% | 10% | 20% | 30% | 50% | 70% | 90% |
|---|---|---|---|---|---|---|---|---|
| | LPIPS | 3.25E-1 | 2.72E-1 | 2.66E-1 | 2.63E-1 | 2.53E-1 | 2.39E-1 | 2.45E-1 |
| UNIPC, Blur | MSE | 2.85E-1 | 1.25E-2 | 1.63E-3 | 2.09E-3 | 3.59E-4 | 1.01E-3 | 6.57E-4 |
| | SSIM | 2.98E-2 | 9.52E-1 | 9.93E-1 | 9.91E-1 | 9.98E-1 | 9.95E-1 | 9.96E-1 |
| | LPIPS | 3.19E-1 | 2.73E-1 | 2.69E-1 | 2.65E-1 | 2.54E-1 | 2.41E-1 | 2.45E-1 |
| UNIPC, Line | MSE | 2.83E-1 | 1.56E-2 | 2.33E-3 | 2.71E-3 | 1.11E-3 | 8.63E-4 | 1.23E-3 |
| | SSIM | 2.13E-2 | 9.42E-1 | 9.90E-1 | 9.89E-1 | 9.95E-1 | 9.96E-1 | 9.94E-1 |
| | LPIPS | 3.10E-1 | 2.87E-1 | 2.79E-1 | 2.79E-1 | 2.65E-1 | 2.51E-1 | 2.52E-1 |
| UNIPC, Box | MSE | 3.04E-1 | 3.63E-2 | 2.16E-2 | 1.43E-2 | 4.97E-3 | 3.31E-3 | 6.02E-3 |
| | SSIM | -1.37E-3 | 8.76E-1 | 9.23E-1 | 9.47E-1 | 9.81E-1 | 9.87E-1 | 9.76E-1 |
| | LPIPS | 3.25E-1 | 2.73E-1 | 2.71E-1 | 2.60E-1 | 2.55E-1 | 2.41E-1 | 2.37E-1 |
| DPM. O2, Blur | MSE | 2.85E-1 | 1.16E-2 | 1.52E-3 | 1.58E-3 | 8.13E-4 | 1.07E-3 | 7.07E-4 |
| | SSIM | 2.98E-2 | 9.58E-1 | 9.93E-1 | 9.93E-1 | 9.96E-1 | 9.95E-1 | 9.97E-1 |
| | LPIPS | 3.19E-1 | 2.76E-1 | 2.68E-1 | 2.63E-1 | 2.57E-1 | 2.42E-1 | 2.45E-1 |
| DPM. O2, Line | MSE | 2.83E-1 | 1.43E-2 | 2.28E-3 | 1.84E-3 | 1.03E-3 | 1.13E-3 | 2.68E-3 |
| | SSIM | 2.13E-2 | 9.46E-1 | 9.91E-1 | 9.92E-1 | 9.95E-1 | 9.94E-1 | 9.89E-1 |
| | LPIPS | 3.10E-1 | 2.86E-1 | 2.76E-1 | 2.77E-1 | 2.65E-1 | 2.54E-1 | 2.52E-1 |
| DPM. O2, Box | MSE | 3.04E-1 | 3.83E-2 | 2.15E-2 | 1.40E-2 | 3.26E-3 | 3.50E-3 | 1.41E-3 |
| | SSIM | -1.37E-3 | 8.67E-1 | 9.23E-1 | 9.48E-1 | 9.87E-1 | 9.86E-1 | 9.93E-1 |
| | LPIPS | 3.25E-1 | 2.71E-1 | 2.65E-1 | 2.62E-1 | 2.59E-1 | 2.39E-1 | 2.45E-1 |
| DPM++. O2, Blur | MSE | 2.85E-1 | 1.16E-2 | 1.46E-3 | 1.77E-3 | 7.03E-4 | 9.69E-4 | 4.74E-4 |
| | SSIM | 2.98E-2 | 9.56E-1 | 9.94E-1 | 9.92E-1 | 9.96E-1 | 9.95E-1 | 9.98E-1 |
| | LPIPS | 3.19E-1 | 2.68E-1 | 2.72E-1 | 2.62E-1 | 2.52E-1 | 2.39E-1 | 2.41E-1 |
| DPM++. O2, Line | MSE | 2.83E-1 | 1.65E-2 | 1.57E-3 | 1.78E-3 | 1.38E-4 | 8.54E-4 | 2.20E-3 |
| | SSIM | 2.13E-2 | 9.38E-1 | 9.92E-1 | 9.92E-1 | 9.99E-1 | 9.96E-1 | 9.90E-1 |
| | LPIPS | 3.10E-1 | 2.85E-1 | 2.79E-1 | 2.72E-1 | 2.67E-1 | 2.52E-1 | 2.53E-1 |
| DPM++. O2, Box | MSE | 3.04E-1 | 4.04E-2 | 2.17E-2 | 1.31E-2 | 4.82E-3 | 4.04E-3 | 6.23E-3 |
| | SSIM | -1.37E-3 | 8.62E-1 | 9.22E-1 | 9.52E-1 | 9.81E-1 | 9.84E-1 | 9.73E-1 |
| | LPIPS | 3.25E-1 | 2.69E-1 | 2.71E-1 | 2.66E-1 | 2.55E-1 | 2.43E-1 | 2.46E-1 |
| DEIS, Blur | MSE | 2.85E-1 | 1.11E-2 | 1.20E-3 | 1.85E-3 | 9.45E-4 | 5.66E-4 | 4.59E-4 |
| | SSIM | 2.98E-2 | 9.59E-1 | 9.94E-1 | 9.93E-1 | 9.95E-1 | 9.97E-1 | 9.97E-1 |
| | LPIPS | 3.19E-1 | 2.73E-1 | 2.66E-1 | 2.59E-1 | 2.53E-1 | 2.43E-1 | 2.41E-1 |
| DEIS, Line | MSE | 2.83E-1 | 1.39E-2 | 2.92E-3 | 2.24E-3 | 5.60E-4 | 1.29E-3 | 1.98E-3 |
| | SSIM | 2.13E-2 | 9.49E-1 | 9.88E-1 | 9.91E-1 | 9.97E-1 | 9.94E-1 | 9.90E-1 |
| | LPIPS | 3.10E-1 | 2.84E-1 | 2.79E-1 | 2.72E-1 | 2.65E-1 | 2.53E-1 | 2.44E-1 |
| DEIS, Box | MSE | 3.04E-1 | 4.07E-2 | 2.16E-2 | 1.16E-2 | 4.20E-3 | 3.75E-3 | 5.07E-3 |
| | SSIM | -1.37E-3 | 8.62E-1 | 9.23E-1 | 9.57E-1 | 9.83E-1 | 9.85E-1 | 9.79E-1 |

Table 33: BadDiffusion and VillanDiffusion on CIFAR10 Dataset with sampler: DDIM and trigger: Stop Sign

| Trigger | | Stop Sign | | | | | | | |
| --- | --- | --- | --- | --- | --- | --- | --- | --- | --- |
| Target | | Hat | | | | No Shift | | | |
| Poison Rate | | 20% | | 50% | | 20% | | 50% | |
| Correction Term DDIM $\eta$ | Metric | Bad | Villan | Bad | Villan | Bad | Villan | Bad | Villan |
| 0.0 | FID | 10.83 | 10.49 | 11.68 | 10.94 | 10.75 | 10.66 | 11.76 | 11.09 |
| | MSE | 2.36E-1 | 3.89E-5 | 2.35E-1 | 2.49E-06 | 1.28E-1 | 6.70E-4 | 1.26E-1 | 3.72E-6 |
| | SSIM | 5.81E-03 | 1.00E+0 | 7.52E-03 | 1.00E+0 | 5.06E-02 | 9.92E-1 | 6.04E-02 | 9.99E-1 |
| 0.2 | FID | 10.47 | 9.84 | 12.04 | 10.31 | 10.42 | 10.06 | 11.34 | 10.58 |
| | MSE | 2.36E-1 | 3.53E-6 | 2.35E-1 | 2.41E-06 | 1.32E-1 | 5.82E-6 | 1.29E-1 | 4.27E-6 |
| | SSIM | 5.71E-3 | 1.00E+0 | 7.53E-3 | 1.00E+0 | 4.11E-2 | 1.00E+0 | 4.96E-2 | 9.99E-1 |
| 0.4 | FID | 13.61 | 10.91 | 19.07 | 11.66 | 13.04 | 11.17 | 17.32 | 11.68 |
| | MSE | 2.37E-1 | 2.57E-3 | 2.37E-1 | 5.34E-4 | 1.39E-1 | 1.57E-4 | 1.37E-1 | 1.21E-4 |
| | SSIM | 4.33E-3 | 9.85E-1 | 5.94E-3 | 9.97E-1 | 1.97E-2 | 9.98E-1 | 2.44E-2 | 9.80E-1 |
| 0.6 | FID | 19.96 | 12.16 | 25.61 | 13.02 | 15.26 | 12.36 | 21.03 | 13.10 |
| | MSE | 2.39E-1 | 8.67E-2 | 2.39E-1 | 5.54E-2 | 1.46E-1 | 4.02E-2 | 1.45E-1 | 4.97E-2 |
| | SSIM | 1.44E-03 | 5.14E-1 | 2.00E-3 | 6.88E-1 | 3.93E-3 | 6.07E-1 | 4.70E-3 | 5.28E-1 |
| 0.8 | FID | 25.18 | 13.85 | 25.97 | 14.76 | 16.28 | 14.07 | 19.32 | 14.75 |
| | MSE | 2.40E-1 | 1.79E-1 | 2.40E-1 | 1.69E-1 | 1.48E-1 | 1.07E-1 | 1.48E-1 | 1.13E-1 |
| | SSIM | 2.29E-4 | 6.96E-2 | 1.15E-3 | 1.02E-1 | 7.87E-4 | 8.61E-2 | 1.12E-3 | 6.69E-2 |
| 1.0 | FID | 26.62 | 18.16 | 24.23 | 18.88 | 19.00 | 18.42 | 20.74 | 18.95 |
| | MSE | 1.86E-2 | 1.46E-1 | 7.74E-4 | 1.48E-1 | 2.14E-2 | 9.35E-2 | 2.22E-5 | 9.48E-2 |
| | SSIM | 9.17E-1 | 5.84E-2 | 9.96E-1 | 5.56E-2 | 8.10E-1 | 9.64E-2 | 1.00E+0 | 9.49E-2 |

Table 34: BadDiffusion and VillanDiffusion on CIFAR10 Dataset with ODE samplers and trigger: Stop Sign

| Trigger | | Stop Sign | | | | | | | |
| --- | --- | --- | --- | --- | --- | --- | --- | --- | --- |
| Target | | Hat | | | | No Shift | | | |
| Poison Rate | | 20% | | 50% | | 20% | | 50% | |
| Correction Term Samplers | Metric | Bad | Villan | Bad | Villan | Bad | Villan | Bad | Villan |
| UniPC | FID | 9.06 | 8.75 | 9.71 | 9.21 | 9.06 | 8.81 | 9.70 | 9.35 |
| | MSE | 2.35E-1 | 9.96E-5 | 2.34E-1 | 9.55E-5 | 1.31E-1 | 2.80E-4 | 1.28E-1 | 4.82E-5 |
| | SSIM | 5.98E-3 | 9.96E-1 | 7.92E-3 | 9.96E-1 | 4.26E-2 | 9.69E-1 | 5.16E-2 | 9.84E-1 |
| DPM-Solver | FID | 9.06 | 8.75 | 9.71 | 9.21 | 9.06 | 8.81 | 9.70 | 9.35 |
| | MSE | 2.35E-1 | 9.96E-5 | 2.34E-1 | 9.55E-5 | 1.31E-1 | 2.80E-4 | 1.28E-1 | 4.82E-5 |
| | SSIM | 5.98E-3 | 9.96E-1 | 7.92E-3 | 9.96E-1 | 4.26E-2 | 9.69E-1 | 5.16E-2 | 9.84E-1 |
| DDIM | FID | 10.83 | 10.49 | 11.68 | 10.94 | 10.75 | 10.66 | 11.76 | 11.09 |
| | MSE | 2.36E-1 | 3.89E-5 | 2.35E-1 | 2.49E-6 | 1.28E-1 | 6.70E-4 | 1.26E-1 | 3.72E-6 |
| | SSIM | 5.81E-3 | 1.00E+0 | 7.52E-3 | 1.00E+0 | 5.06E-2 | 9.92E-1 | 6.04E-2 | 9.99E-1 |
| PNDM | FID | 7.30 | 7.04 | 7.72 | 7.28 | 7.14 | 7.16 | 7.69 | 7.42 |
| | MSE | 2.36E-1 | 7.22E-5 | 2.35E-1 | 3.27E-5 | 1.31E-1 | 4.75E-4 | 1.28E-1 | 3.17E-5 |
| | SSIM | 5.65E-3 | 9.97E-1 | 7.38E-3 | 9.98E-1 | 4.38E-2 | 9.75E-1 | 5.30E-2 | 9.83E-1 |

