# OpenReview forum: "VillanDiffusion: A Unified Backdoor Attack Framework for Diffusion Models"
_NeurIPS.cc/2023/Conference — NeurIPS 2023 poster_

### Official Review · Reviewer_zhWX · 2023-07-01

**Soundness:** 3 good
**Presentation:** 3 good
**Contribution:** 3 good
**Rating:** 5
**Confidence:** 4

**Summary:**

This paper presents a unified backdoor attack framework (VillanDiffusion) to expand the current scope of backdoor analysis for DMs. The proposed framework covers mainstream unconditional and conditional DMs (denoising-based and score-based) and various training-free samplers for holistic evaluations.

**Strengths:**

1. Their experimental results not only analyzed DDPM but also score-based models. Besides, they also analyzed other acceleration sampling methods.
2. Their experiments included caption triggers.

**Weaknesses:**

1. To show that no modifications are needed to the sampling process, the article should include details of the sampling process.
2. The order of formulas 8-12 in the article is not clear enough. To describe in the order of forward process → backward process → sampling process may be more clear.
3. Please check some spelling errors in the article. For example, "Praobility" in the title on line 138.
4. Please check whether the last term in Eq. 4 is $L_0(x_1,x_0)$.
5. The article claims that BadDiffusion will fail when the coefficient is $\frac{1}{2}$ (line 55), but there is no further explanation.
6. The article claims that an attacker only needs to obtain the model parameters $\theta_{download}$. However, to execute a backdoor attack, some adjustments need to be made to the initial noise. This is difficult to achieve in reality, and the article needs to emphasize this point.
7. Although the article extends the attack to other models, such as score-based models, there is no essential difference from BadDiffusion[2]. On BadDiffusion, just changing the coefficient of the noise term $1-\bar\alpha_t \mathbf{I}$ in the formula $q\left(\mathbf{x}_{t}^{\prime} \mid \mathbf{x}_{0}^{\prime}\right):=\mathcal{N}\left(\mathbf{x}_{t}^{\prime} ; \sqrt{\bar{\alpha}_{t}} \mathbf{x}_{0}^{\prime}+\left(1-\sqrt{\bar{\alpha}_{t}}\right) \mathbf{r},\left(1-\bar{\alpha}_{t}\right) \mathbf{I}\right)$ (Eq. 6 in their article [2]) to $b_k \mathbf{I}$ can easily obtain the results in this article. In addition, Eq. 11 is only a change of sign of Eq. 38 in [1].


[1] Song, Y., Sohl-Dickstein, J., Kingma, D. P., Kumar, A., Ermon, S., & Poole, B. (2020). Score-based generative modeling through stochastic differential equations. arXiv preprint arXiv:2011.13456.

[2] Chou, S. Y., Chen, P. Y., & Ho, T. Y. (2023). How to backdoor diffusion models?. In Proceedings of the IEEE/CVF Conference on Computer Vision and Pattern Recognition (pp. 4015-4024).

**Questions:**

Please see the Weakness part.

**Limitations:**

The authors didn’t mention any potential limitations.
The authors can list limitations of the application. For example, to apply their method, the attacker needs to access the initial noise, which is not practical sometimes. Besides, the negative societal impact should also be included.

---

> ### Author Rebuttal · Authors · 2023-08-10
>
> Thank you for the valuable suggestions. We will reply to your comment one by one in the following.
>
> **[Including More Details of ODE Samplers]** Many thanks for your beneficial suggestions. Our paper uses genuine ODE samplers implemented by the library "diffusers." We will also introduce the samplers we used in the article.
>
> **[Reorder the Presentation of Processes]** Thank you for the excellent suggestion. We will introduce the forward process and reorder the formula presentation as your comments.
>
> **[Correct the Typos]** Thank you for your thorough review. We will correct the typos in line 138 and equation 4.
>
> **[Explain Why BadDiffusion Fails]** Many thanks for your meaningful and valuable question. We will provide further theoretical and empirical explanations. Firstly, in a theoretical view, we can derive a backdoor reversed transitional probability $q(x_{t-1}' | x_{t}')$ given a forward transitional probability $q(x_{t}' | x_{t-1}')$. Thus, the backdoor reversed transitional probability describes a backdoored reversed diffusion process from T to 0. We can convert backdoor reversed transitional probability into an SDE (equation 18 in appendix) with Taylor approximation as shown in appendix B.3.1. Also, to extend the results to ODE, we can introduce an additional parameter $\zeta$ and use the Fokker-Planck equation shown in appendix lemma 1. Finally, we can obtain equation 10 as a general form of a backdoor reversed process.
>
> On the other hand, different samplers might simulate the process deterministically (ODE) or stochastically (SDE), as in equation 11. Comparing equations 10 and 11, we found that the loss function of BadDiffusion can be derived with $\zeta = 1$, which means BadDiffusion is just a special SDE case of VillanDiffusion.
>
> Empirically, we also conduct an experiment described in the general response 2. We evaluated BadDiffusion on multiple ODE samplers, including DPM-Solver, PNDM, and UniPC, and found it performed badly. Furthermore, in general response 3, we also provide empirical evidence for our theory. We show that the randomness of samplers is the crucial factor that affects the performance of the backdoors because when the randomness of samplers drops, the MSE of VillanDiffusion trained for ODE goes down, but BadDiffusion goes up.
>
> **[The Difference from BadDiffusion]** Thank you for your valuable thoughts. We would like to emphasize that our main contribution is to provide a unified framework to explore advanced backdoor attacks on various diffusion models, especially for those that have not been studied, such as score-based models, DDIM, DEIS, and DPM-Solver. Firstly, with flexible correction schedulers obtained from VillanDiffusion, researchers can explore backdoor attacks based on their own configuration of the diffusion models. Secondly, with our unified view of continuous and discrete diffusion processes, researchers can incorporate the concepts of ODE and SDE to analyze the effectiveness of backdoor attacks under the same framework. For example, recent works have investigated the effect of the self-consistency property of ODE on diffusion models. Some works enhance the self-consistency of the sampling trajectory to improve the sample quality in one inference step. With our framework, researchers can also investigate the impact of the self-consistency of ODE on the backdoor to discover more advanced attacks. Thus, our method can facilitate more advanced backdoor attacks on diffusion models.
>
> **[Adjusting the Initial Noise is not Feasible]** We thank the reviewer for the suggestion. We would like to clarify that the attacker actually does not need to access the initial noise. The main idea of backdooring diffusion models is modifying the mean of diffusion processes. Thus, we only need to add a specific patch to the noise during training. At inference time, the attacker only needs to attach the trigger to the data input. Furthermore, we also take the inpainting task as a practical example in Appendix C6. Inpainting is a common application for diffusion models. We found that by inserting a trigger into the corrupted images, the diffusion models can produce target images easily. We also use LPIPS to evaluate the quality of recovered images and MSE to measure the backdoor success rate and found that our method can achieve both high utility and specificity.
>
> **[Potential Limitation]** I want to express my gratitude for your valuable recommendations. To address your comment on discussing the limitations, we will add a discussion on the limitation that although VillanDiffusion is a general framework that covers many existing configurations of diffusion models and sampling schemes, it is possible that VillanDiffusion cannot be applied in cases where the framework does not hold.
>
> **[Negative Societal Impact]** Your valuable guidance is warmly acknowledged. On negative societal impact, we will add that "Although cast as a general backdoor attack framework, we position our work as a red-teaming tool to explore and unveil hidden risks in diffusion models. We believe our framework can help accelerate the development of robust diffusion models."

---

> > ### Comment · Reviewer_zhWX · 2023-08-18
> > **Reply to the rebuttal**
> >
> > Thank you for the kind response. Most of the concerns were addressed and I decided to increase my score.

---

> > > ### Author Response · Authors · 2023-08-18
> > >
> > > We thank Reviewer zhWX for your reply and for increasing our rating! Again, we thank the reviewer for the valuable comments and suggestions.

---

### Official Review · Reviewer_mTjd · 2023-07-05

**Soundness:** 3 good
**Presentation:** 3 good
**Contribution:** 2 fair
**Rating:** 6
**Confidence:** 3

**Summary:**

This paper proposed a backdoor attack framework called VillanDiffusion, which extends the existing backdoor analysis capabilities for deep models (DMs). By encompassing both unconditional and conditional DMs, including denoising-based and score-based models, as well as incorporating training-free samplers, the proposed framework enables holistic evaluations of backdoor attacks. Experiments demonstrate that VillanDiffusion not only facilitates the analysis of diverse DM configurations but also offers valuable insights into caption-based backdoor attacks on DMs.

**Strengths:**

The paper is written in a clear and understandable manner, making it accessible to a wide range of readers. The authors effectively convey their ideas and concepts, ensuring that the content is comprehensible. Besides, the authors provide a thorough analysis of backdoor attacks on deep models and propose a unified approach, VillanDiffusion.

**Weaknesses:**

There are concerns regarding the effectiveness of the VillanDiffusion framework The framework proves to be effective when the poison ratio reaches up to 20%, which is significantly higher than what is typically observed in other tasks. This discrepancy raises doubts about the practicality of the framework, emphasizing the need for further investigation and evaluation across various tasks and poison ratios to ensure its broader applicability.

**Questions:**

See the above weakness.

**Limitations:**

Yes

---

> ### Author Rebuttal · Authors · 2023-08-10
>
> Thank you for your appreciation of our work. Here is our comment on the weakness.
>
> **[The Effective Poison Rate is Too High]** Thank you for your insightful comments. Firstly, in our threat model, the attackers would release the backdoor DMs to the public. As a result, once the utility is high enough to fool the users, the attack will succeed, no matter how high the poison rate is. The same threat model is also used in standard backdoor attacks BadNet [R1] and BadDiffusion. Secondly, we also train the backdoor DDPM on CelebA-HQ with a lower poison rate and more training epochs. However, due to limited time, we only have enough time to train a 10%-poisoned model with 330 epochs. We also attach the generated target and clean image in the author’s rebuttal Figure 1(c) and 1(d). Also, we can see that the target image emerges clearly in the figure. Note that in comparison to full training epochs: 2500, it’s a significant backdoor effect in a very early training period. Therefore, we might attack successfully with a 10% poison rate, which is much lower than 20%.
>
> Furthermore, we also take the inpainting task as a practical example in Appendix C6. Inpainting is a common application for diffusion models. We found that by inserting a trigger into the corrupted images, the diffusion models can generate target images easily. We also use LPIPS to evaluate the quality of recovered images and MSE to measure the backdoor success rate and found that our method can achieve both high utility and specificity.
>
> [R1] Tianyu Gu, Kang Liu, Brendan Dolan-Gavitt, Siddharth Garg BadNets: Evaluating Backdooring Attacks on Deep Neural Networks [C] IEEE Access

---

### Official Review · Reviewer_FRNp · 2023-07-06

**Soundness:** 3 good
**Presentation:** 2 fair
**Contribution:** 2 fair
**Rating:** 5
**Confidence:** 3

**Summary:**

This paper proposes a universal backdoor attack framework on diffusion models facing different kinds of content schedulers, different kinds of samplers, and conditional and unconditional tasks.

**Strengths:**

1. This paper proposes a universal backdoor attack framework on diffusion models, which are important.
2. The experiments are sufficient.
3. This paper is well written and technically sound.

**Weaknesses:**

1. From my point of view, backdoor in diffusion models is an end-to-end process, can you explain the main difference from some prior works in diffusion models such as [1]? If the only difference is to test on different diffusion models, the contribution is limited.
2. There is no comparison with the former methods, and thus I cannot find out whether there is improvement in backdoor attack.
3.There are some flaws such as line 191 learns should.



[1] Chou S Y, Chen P Y, Ho T Y. How to backdoor diffusion models?[C]//Proceedings of the IEEE/CVF Conference on Computer Vision and Pattern Recognition. 2023: 4015-4024.

**Questions:**

Can you explain the main difference from some prior works in diffusion models such as [1]?

**Limitations:**

Yes

---

> ### Author Rebuttal · Authors · 2023-08-10
>
> Thank you for giving me such valuable advice. We will elaborate on the following points.
>
> **[Main Difference from BadDiffusion]** Thank you for sharing your valuable thoughts. Firstly, from a theoretical perspective, our work is not just an extension of BadDiffusion but a general framework to deal with various configurations of diffusion models. With our theory in line 198, we can derive the correct backdoor correction terms of diffusion models described within lines 110 ~ 116, which can cover not only DDPM and score-based models but much more than these. To the best of our knowledge, no backdoor attacks can be applied to this wide range of diffusion models. We also provide proof that the loss of BadDiffusion describes a backdoor SDE, which is also a special case of a general backdoor process controlled by a hyperparameter $\zeta$. Our contribution is not only the simple and elegant adaptive correction term derived from VillanDiffusion with the specification of diffusion settings but also the general framework that is beneficial to different configurations of diffusion models, including SDE samplers, ODE samplers, flexible noise schedulers, and conditional generation. Due to the generality of VillanDiffusion, it also serves as a powerful tool for exploring backdoor attacks of future diffusion models. Secondly, from an empirical point of view, we also show the limitations of BadDiffusion in the general responses 2 and 3. In general response 2, we offer that the BadDiffusion would work poorly for ODE samplers and only works on DDPM, while VillanDiffusion performs well. In general response 3, we also show that the randomness of samplers is the critical factor that affects the performance of the backdoors because when the randomness of samplers decreases, the MSE of VillanDiffusion trained for ODE goes down, but BadDiffusion goes up. It also offers empirical evidence for our theory.
>
> **[Compare to Other Baselines]** Thank you for your advice. BadDiffusion could not work with ODE samplers because it actually describes an SDE, which is proved in our papers theoretically and in general responses 2 and 3 empirically. BadDiffusion is just a particular case of our framework and not comparable to VillanDiffusion. However, we still conduct an experiment to evaluate BadDiffusion on some ODE samplers and present the results in general response 2. We can see that BadDiffusion performs much more poorly than VillanDiffusion. Also, in general response 3, we point out that the leading cause of this phenomenon is the level of stochasticity and provide empirical evidence of our theory. As for TrojDiff and RickRolling the Artist, the authors modify the samplers and the text encoder to achieve backdoor attack respectively, so it has different threat models than us. Both methods are not comparable to our approach. This study is the first to explore backdoor attacks in many diffusion model configurations, such as ODE samplers and flexible noise schedulers. Our study is also the first line of works exploring conditional generation.
>
> **[Line 191 Flaws]** Your valuable advice is much appreciated. We will revise the sentence as the following: When we compare it to the learned reversed process of SDE Eq. (11), we can see that the diffusion model $\epsilon_{\theta}$ should learn the backdoor score function to generate the backdoor target distribution $q(\mathbf{x}_{0}’)$.

---

> > ### Comment · Reviewer_FRNp · 2023-08-21
> >
> > Most of concerns are solved and I raised my scores.

---

> > > ### Author Response · Authors · 2023-08-21
> > >
> > > We appreciate the reviewer's response and for increasing your rating score!

---

### Official Review · Reviewer_FG5D · 2023-07-08

**Soundness:** 3 good
**Presentation:** 2 fair
**Contribution:** 2 fair
**Rating:** 5
**Confidence:** 4

**Summary:**

The paper presents VillanDiffusion, a framework for analyzing backdoor attacks on different types of diffusion models (DMs). VillanDiffusion covers various DM configurations such as unconditional and conditional DMs or training-free samplers and provides new insights into caption-based backdoor attacks.

**Strengths:**

+ originality, the paper presents a unified framework for analyzing backdoor attacks on DMs, covering various configurations and training-free samplers. the soundness of this paper is also noteworthy, with detailed proof in Appendix.

+ The experiments are comprehensive and demonstrate the effectiveness of the VillanDiffusion framework in detecting backdoor attacks on DMs.

 + The paper is well-structured, with each section building on the previous one, making it easy to follow.

**Weaknesses:**

- the effectiveness of their backdoor attack on Celeba is limited: the increased FID score is huge in this scenario, which does not show the advantage of their method against other baselines.

**Questions:**

- how to read the figure 2a and 2e? on line 278, this paper says that "From Fig. 2a and Fig. 2e, we can see the FID score of the backdoored DM on CelebA-HQ-Dialog is slightly better than the clean one". However, it seems that data in Fig. 2a and Fig. 2e are the generated clean samples by the backdoored model.

- lack of further analysis about the robustness of various configurations of DMs against backdoor attacks: since VillanDiffusion can generalize to different mechanisms, samplers, and schedulers, it is worthwhile to analyze how the different modules affects the robustness of DMs.

- how DMs pretrained on cifar-10 become stronger in terms of utility after being fine-tuned on cifar-10? on line 298, this paper says that "We can see all samplers reach lower FID scores than the clean models under 70% poison rate for the image trigger Hat." Since the data used for pretraining and fine-tuning is the same, why does the generation ability of DMs still increase?

**Limitations:**

yes

---

> ### Author Rebuttal · Authors · 2023-08-10
>
> Thank you for the valuable suggestions. We will reply to your questions in the following.
>
> **[FID Score Increase]** Thank you for the constructive advice. To fully evaluate the threat of VillanDiffusion, we train the backdoor DDPM on CelebA-HQ with 20% poison rate and more training epochs (the original training epochs are 1500). We found that with 2000 training epochs and the UniPC sampler, the FID score will become 19.26, which is much lower than we report: 20.67. We believe that with sufficient training, the utility and specificity of the backdoor can get much better. We will update our better results in the future.
>
> **[Elaborate Figure 2(a) and 2(e)]** I am thankful for your valuable feedback. In Figure 2, we use empty quotation “” and green dots as the results of clean (backdoor-free) models. In the attachment, Figure 1, we mark the results with red boxes. In the Figure 1(a), we can see the FID scores of clean models trained on the CelebA-HQ-Dialog dataset are about 25, which are slightly higher than backdoored models.
>
> **[Further Analysis of the Robustness of Various Configurations]** We appreciate your precious comments. According to general responses 2 and 3, we’ve conducted experiments on BadDiffusion and VillanDiffusion with different samplers. We found that BadDiffusion is only effective in SDE samplers. When DDIM $\eta$ goes down, which means the sampler becomes more likely an ODE, the MSE of VillanDiffusion trained for ODE samplers would decrease, but BadDiffusion would increase. Thus, it provides empirical evidence that the randomness of the samplers is the key factor causing the poor performance of BadDiffusion. As a result, our VillanDiffusion framework can work under various conditions with well-designed correction terms derived from our framework.
>
> **[Poisoned DDPM Becomes Stronger]** I am thankful for the valuable input you've provided. We use the same pre-trained CIFAR10 DDPM model as BadDiffusion. According to Figure 2(b) in the BadDiffusion paper, they also present a similar phenomenon that many poisoned models achieved better FID scores than clean models. It might be caused by the non-optimal training of the models.

---

### Author Rebuttal · Authors · 2023-08-09

## General Response

Thanks for the insightful comments. We appreciate your precious reviews. Here, we will give a general response to common suggestions.

**[Unlike standard backdoor attacks, backdoor diffusion models require modifying the diffusion process]** Based on the review comments regarding the difference of VillanDiffusion to existing backdoor attacks, there are two major points that we would like to emphasize: (1) regular data poisoning attacks for backdoor injection (e.g., only changing the training data and labels) are not effective on diffusion models. Diffusion models aim to learn cascading denoising processes, and the models would learn to remove specific levels of noise. We can see the loss function of DDPM in equation 7. It would add a specific amount of Gaussian noise $\hat{\beta}(t) \epsilon$ to images and remove $\epsilon$ noise from noisy images. With hundreds of steps, a meaningful pattern would finally emerge. Thus, simply poisoning the dataset with a fixed backdoor trigger and target without modifying the diffusion process would not inject trojans into diffusion models successfully.

As a result, refer to villanDiffusion loss in line 200. Backdoored diffusion models need to learn how to remove specific levels of triggers $\frac{2 H(t)}{(1+\zeta) G^2(t)} r$. In addition, the removed levels of triggers vary over time $t$ based on the content $\hat{\alpha}(t)$ and noise scheduler $\hat{\beta}(t)$. In contrast, regular data poisoning would make the diffusion models learn to remove the triggers at once and cause wrong pattern accumulation. That is also why we need an additional correction term for backdooring diffusion models. (2) VillanDiffusion is a universal framework for any diffusion model following the diffusion process, and BadDiffusion is just a particular case under our framework. In addition, BadDiffusion only works on DDPM and ancestral sampling (the original sampler of DDPM) with well-designed and specific correction terms. Secondly, our framework incorporates the continuous view of diffusion models, like SDE, which has not been explored. It also provides a tool to analyze backdoor attacks for different configurations. Researchers can also investigate the risks of backdoor attacks on their own diffusion models by designing different correction schedulers following our framework.

**[BadDiffusion fails to generalize to different configurations, while VillanDiffusion does not]** To further demonstrate the generality of VillanDiffusion and the limitation of BadDiffusion, we conduct experiments to show that backdooring DDPM with CIFAR10 and various ODE samplers (including UniPC, DPM-Solver, and DEIS, etc.) will not be successful with BadDiffusion loss, while using the correct loss derived from VillanDiffusion will be successful. Please refer to Table 2 in the author's rebuttal. In Table 2, we can find the MSE of BadDiffusion remains high and SSIM keeps low. That means BadDiffusion performs badly on ODE.

**[Further Analysis of the Robustness of BadDiffusion and VillanDiffusion]** Here, we conduct an additional experiment. We evaluate the robustness of BadDiffusion and VillanDiffusion with different randomness ($\eta$) hyperparameters of DDIM samplers. We attach the numerical results of the experiment in Table 1 of the author's rebuttal. Also, we control the randomness of the DDIM sampler with hyperparameters $\eta$. When eta is 0, the DDIM sampler has no randomness and reduces to an ODE sampler. In contrast, when eta is 1, the DDIM sampler would become an SDE sampler. As a result, we can evaluate the effects of the randomness of samplers on the correction terms derived from ODE and SDE. As the figures show, when the $\eta$ goes down, and DDIM gets closer to an ODE sampler, the terms derived from ODE (VillanDiffusion) would become more effective, and ones of SDE (BadDiffusion) would worsen. Thus, we can see that the randomness of the samplers is the key factor causing the failure of BadDiffusion, as our theory presents. The results also show the necessity and novelty of our framework to implement successful backdoor attacks.

---

### Decision · Program_Chairs · 2023-09-21

**Decision:**

Accept (poster)

**Comment:**

All reviewers agree with the contribution of this paper to the community. AC hopes the authors can improve the paper based on reviewers' comments in the final version.